# Convergence of Steepest Descent and Adam under Non-Uniform Smoothness

**Sharan Vaswani** [1]  **Yifan Sun** [2]  **Reza Babanezhad** [3]

## Abstract

Recent work has analyzed the convergence of first-order methods under non-uniform smoothness assumptions that better model the loss landscape in machine learning tasks. We generalize this assumption to objectives whose curvature is an affine function of the objective value. This property is satisfied by a broad class of problems, including logistic regression, generalized linear models with a logistic link function, softmax policy gradient in reinforcement learning, and a class of neural networks. Under this assumption and gradient domination conditions, we establish a general convergence rate for the steepest descent method, and deterministic, diagonal variants of RMSProp and Adam. Our results imply that for logistic regression on separable data and the softmax policy gradient objective, sign GD converges linearly and is provably faster than GD. Furthermore, we show that for a class of two-layer neural networks on separable data, RMSProp and Adam can converge at a linear rate with a constant step-size and momentum parameter. Finally, we present a lower bound demonstrating that, under our assumption, RMSProp and Adam are provably faster than AdaGrad, AMSGrad, gradient descent, and heavy-ball momentum.

## 1. Introduction

Recent works (Zhang et al., 2020b;a; Chen et al., 2023b; Vankov et al., 2025; Gorbunov et al., 2025; Vaswani & Harikandeh, 2025; Alimisis et al., 2025) have studied the convergence of first-order methods under *non-uniform smoothness*. Unlike the standard smoothness assumption, which imposes a global uniform upper-bound on the Hes-

sian norm, non-uniform smoothness (NS) upper-bounds the Hessian locally as a function of the parameter value. For example, Zhang et al. (2020b) proposed that the Hessian norm at any point can be bounded as an affine function of the gradient norm at that point. They empirically showed that this property holds during neural network training under standard loss functions, and leveraged this assumption to justify the role of gradient clipping.

Following this, there have been numerous works (Chen et al., 2023b; Li et al., 2023a; Vaswani & Harikandeh, 2025; Alimisis et al., 2025) relaxing this NS assumption, with the goal of better modeling the loss landscapes arising in machine learning problems. Under such NS assumptions, recent works have revisited the theoretical convergence rates of common first-order methods including gradient descent (GD) with adaptive step-sizes, heavy-ball momentum (Vankov et al., 2025; Gorbunov et al., 2025; Vaswani & Harikandeh, 2025; Hübler et al., 2024), and adaptive gradient methods such as Adam and RMSProp (Li et al., 2023b; Wang et al., 2024a;b) (see App. A for a detailed review).

We focus on a particular form of NS where the spectral norm of the Hessian is upper-bounded by an affine function of the objective value (Vaswani & Harikandeh, 2025; Alimisis et al., 2025). For twice-differentiable functions, this property strictly generalizes the NS assumption in Zhang et al. (2020b). Furthermore, this property is satisfied by a broad class of problems, including logistic regression, generalized linear models with a logistic link (Mei et al., 2021), softmax policy gradient in reinforcement learning (Mei et al., 2020), and certain two-layer neural networks (Taheri & Thrampoulidis, 2023; Alimisis et al., 2025).

Moreover, unlike the assumption in Zhang et al. (2020b), the class of functions satisfying this alternative NS assumption is closed under finite-sums and affine transformations (Alimisis et al., 2025). This enables an easier analysis of finite-sum losses prevalent in machine learning. We note that under this assumption, Vaswani & Harikandeh (2025) analyzed GD and demonstrated the advantage of using an Armijo line-search over constant step-sizes. On the other hand, Alimisis et al. (2025) have used this assumption to theoretically characterize the importance of learning rate warm-up. In this paper, we generalize this assumption to

[1]Simon Fraser University [2]Stony Brook University [3]Samsung AI, Montreal. Correspondence to: Sharan Vaswani <vaswani.sharan@gmail.com>.

*Proceedings of the 43rd International Conference on Machine Learning*, Seoul, South Korea. PMLR 306, 2026. Copyright 2026 by the author(s).

handle induced $(p, q)$ operator norms of the Hessian, and refer to the resulting property as $(H_0, H_1)$-NS.

In addition to this assumption, we consider objectives that satisfy a (possibly non-uniform) Łojasiewicz (NL) assumption. The NL condition is a gradient domination property implying that the gradient norm is lower-bounded in terms of the function sub-optimality. Hence, minimizing the gradient norm results in minimizing the function value. Consequently, the NL condition is widely used to analyze the global convergence of algorithms (Karimi et al., 2016). It is satisfied for convex objectives such as quadratics, the exponential loss on separable data, and structured non-convex losses including the softmax policy gradient objective (Mei et al., 2020), matrix factorization (Ward & Kolda, 2023) and sufficiently over-parameterized neural networks (Liu et al., 2022; Soltanolkotabi et al., 2018; Zou & Gu, 2019; Li & Liang, 2018).

Working under the $(H_0, H_1)$-NS and NL assumptions, we derive convergence guarantees for steepest descent, and deterministic, diagonal variants of RMSProp and Adam. To our knowledge, these are the first such results. Our contributions are summarized as follows.

**Contribution 1:** In Sec. 3, we derive structural properties of functions satisfying $(H_0, H_1)$-NS. In particular, we first generalize the results in Vaswani & Harikandeh (2025); Alimisis et al. (2025) beyond Euclidean norms. This subsequently allows us to derive convergence rates for steepest descent algorithms without any dimension dependence. Furthermore, we prove that $(H_0, H_1)$-NS implies *multiplicative Lipschitz* bounds for both the function and the gradient norm. This property enables a new framework for analyzing RMSProp (Tieleman, 2012) and Adam (Kingma & Ba, 2015).

**Contribution 2:** In Sec. 4, under our NS and NL assumptions, we analyze the convergence of normalized steepest descent (NSD) methods including Sign GD and Normalized GD. Our general theorem implies that for logistic regression on separable data and the softmax policy gradient objective, using Sign GD with a constant step-size results in a dimension-free linear convergence rate matching GD with line-search (Vaswani & Harikandeh, 2025). For these applications, Sign GD is provably faster than constant step-size GD (Mei et al., 2020; Wu et al., 2024). Furthermore, our results demonstrate the effectiveness of Sign CD-GS, a specific coordinate descent instantiation of NSD. Sign CD-GS uses the Gauss–Southwell rule (Nutini et al., 2015) to select the coordinate, and updates it using the sign of its gradient. When minimizing the exponential or logistic loss on separable data, Sign CD-GS converges linearly, matching the rate for a normalized variant of coordinate descent (Axiotis & Sviridenko, 2023).

**Contribution 3:** In Sec. 5, we derive convergence rates for the deterministic, diagonal variants of RMSProp and Adam. Our results imply that for a class of two-layer neural networks on separable data, RMSProp and Adam can converge with a constant step-size, and do so at a dimension-free linear rate. Importantly, our analysis does not require any convexity or bounded gradient assumption. In contrast to our setting, previous works (Li et al., 2023b; Wang et al., 2024a;b) consider non-convex functions satisfying the NS assumption in Zhang et al. (2020b), and analyze both deterministic and stochastic variants of Adam. For this class of functions, we show that our proof techniques can also be used to attain faster rates in the deterministic setting.

**Contribution 4:** In Sec. 6, we consider the one-dimensional logistic loss which satisfies our NL and NS assumptions, and for which RMSProp and Adam achieve a linear convergence rate. For this example, we prove a sub-linear lower-bound for GD, heavy-ball momentum and deterministic variants of AdaGrad (Duchi et al., 2011) and AMSGrad (Reddi et al., 2018). Consequently, our results show that these methods are provably slower on this class of functions. This is the first such separation for adaptive gradient methods, and provides theoretical justification for the practical dominance of RMSProp and Adam over AdaGrad and AMSGrad.

## 2. Problem Formulation

We aim to solve the unconstrained minimization problem: $\min_{\theta \in \mathbb{R}^D} f(\theta)$. We define $\theta^* \in \arg\inf f(\theta)$ as an optimal solution and $f^* := \inf f(\theta)$ as the minimum function value. We make the following assumptions.

**Assumption 1.** *$f$ is twice-differentiable and non-negative i.e. for all $\theta$, $f(\theta) \geq 0$.*

**Assumption 2.** *($(H_0, H_1)$-NS) $f$ is $(H_0, H_1)$ non-uniform smooth if for constants $H_0 \geq 0, H_1 \geq 0$, and $p, q \geq 1$ s.t. $\frac{1}{p} + \frac{1}{q} = 1$, for all $\theta$,*

$$\left\| \nabla^2 f(\theta) \right\|_{p \to q} \leq H_0 + H_1 f(\theta), \qquad (1)$$

*where, for a matrix $A$, $\|A\|_{p \to q} = \max_{\|x\|_p \leq 1} \|A x\|_q$.*

For twice-differentiable functions, recent work (Vaswani & Harikandeh, 2025; Alimisis et al., 2025) has considered Assn. 2 with $(p, q) = (2, 2)$, and proved that the resulting condition generalizes the non-uniform smoothness conditions in prior works (Zhang et al., 2020b;a).

In particular, if a function is twice-differentiable and $(L_0, L_1)$-NS meaning that,

$$\left\| \nabla^2 f(\theta) \right\|_2 \leq L_0 + L_1 \left\| \nabla f(\theta) \right\|_2, \qquad (2)$$

then, it also satisfies Assn. 2 (Vaswani & Harikandeh, 2025, Prop. 3).

Furthermore, note that if $H_1 = 0$ and $(p,q) = (2,2)$, Assn. 2 recovers the standard uniform smoothness condition as a special case. Consequently, common smooth objectives such as linear regression or logistic regression satisfy Assn. 2. For example, if $X \in \mathbb{R}^{n \times D}$ is the feature matrix, and $y \in \mathbb{R}^n$ is the vector of measurements, then, the linear regression objective, $f(\theta) = \frac{1}{2n} \|X\theta - y\|^2$ is $(\frac{1}{n}\lambda_{\max}[X^T X], 0)$-NS where $\lambda_{\max}[A]$ is the maximum eigenvalue of the positive semi-definite matrix $A$. In this paper, we will be particularly interested in functions for which $H_1 > 0$.

In addition to Assn. 2, we consider functions that also satisfy a Łojasiewicz or gradient domination condition.

**Assumption 3.** *(NL) $f$ satisfies a non-uniform Łojasiewicz condition if for $\tau \in (0,1]$, for all $\theta$, there exists $\mu(\theta) > 0$, such that,*

$$\|\nabla f(\theta)\|_q \geq \mu(\theta) \left[ f(\theta) - f^* \right]^\tau. \tag{3}$$

First, we note that Assn. 3 with $\tau = \frac{1}{2}$ and a uniform $\mu$ (for all $\theta$) is known as the Polyak Łojasiewicz condition and generalizes the notion of strong-convexity. For $\tau = \frac{1}{2}$, Assn. 3 implies curvature near the optimum, and is related to the restricted secant inequality (Zhang & Yin, 2013) and error bound conditions (Luo & Tseng, 1993) used to analyze the global convergence of algorithms (Karimi et al., 2016) despite non-convexity.

### 2.1. Examples

To motivate Assn. 2 and 3, in App. B, we prove that common convex objectives for supervised learning such as binary classification with the exponential loss, linear logistic regression and linear multi-class classification can satisfy these assumptions. While such examples have been studied in prior works, we generalize these results and present them to highlight the prevalence of our assumptions in machine learning problems.

**Proposition 1.** *Consider $n$ points where $x_i \in \mathbb{R}^d$ are the features and $y_i \in \{-1, 1\}$ are the corresponding labels. Binary classification with an exponential loss,*

$$f(\theta) := \frac{1}{n} \sum_{i=1}^n \exp(-y_i \langle x_i, \theta \rangle), \tag{4}$$

*satisfies Assn. 1 and Assn. 2 with $H_0 = 0$ and $H_1 = \max_i \|x_i\|_q^2$. Furthermore, if the data is separable with a normalized margin $\gamma_p := \max_\theta \min_i \frac{y_i \langle x_i, \theta \rangle}{\|\theta\|_p} > 0$, then, $f(\theta)$ satisfies Assn. 3 with $\tau = 1$, $\mu = \gamma_p$ and $f^* = 0$.*

Note that the above example is not uniform smooth on an unbounded domain, but satisfies Assn. 2.

**Proposition 2.** *Consider $n$ points where $x_i \in \mathbb{R}^D$ are the features and $y_i \in \{-1, 1\}$ are the corresponding labels.*

*Logistic regression with the objective,*

$$f(\theta) := \frac{1}{n} \sum_{i=1}^n \ln(1 + \exp(-y_i \langle x_i, \theta \rangle)) \tag{5}$$

*satisfies Assn. 1 and Assn. 2 with $H_0 = 0$ and $H_1 = \max_i \|x_i\|_q^2$. Furthermore, if the data is separable with a normalized margin $\gamma_p := \max_\theta \min_i \frac{y_i \langle x_i, \theta \rangle}{\|\theta\|_p} > 0$ then,*

- *For $f(\theta) \leq \frac{\ln(2)}{n}$, $f$ satisfies Assn. 3 with $\tau = 1$, $\mu = \frac{\gamma_p}{2}$ and $f^* = 0$*
- *Else, if $f(\theta) > \frac{\ln(2)}{n}$, then, $\|\nabla f(\theta)\|_q \geq \frac{\gamma_p}{3n}$.*

Note that the logistic regression objective is also uniform smooth, meaning that it simultaneously satisfies Assn. 2 with $H_0 = \frac{1}{4n}\lambda_{\max}[X^T X]$ and $H_1 = 0$, where $X \in \mathbb{R}^{n \times D}$ is the corresponding feature matrix. The above results generalize those in Vaswani & Harikandeh (2025) from $(p,q) = (2,2)$ to general pairs of dual norms. We defer the result for multi-class classification to Prop. 5 in App. B.

Assn. 2 and 3 are also satisfied by certain non-convex functions such as the softmax policy gradient objective in reinforcement learning. In particular, we prove the following result for the multi-armed bandit problem with known deterministic rewards. This setting is often used as a testbed to analyze policy gradient methods (Mei et al., 2020; Lu et al.).

**Proposition 3.** *Given a multi-armed bandit problem with $K$ arms and known deterministic rewards $r \in [0,1]^K$, consider softmax policies $\pi_\theta \in \Delta_K$ parameterized by $\theta \in \mathbb{R}^K$ s.t. $\pi_\theta(a) = \exp(\theta(a))/\sum_{a'} \exp(\theta(a'))$. The softmax policy gradient objective is given by*

$$f(\theta) := r(a^*) - \langle \pi_\theta, r \rangle, \tag{6}$$

*where $a^* := \arg\max_{a \in [K]} r(a)$ is the optimal arm. $f(\theta)$ satisfies Assn. 1 and Assn. 2 with (i) $H_0 = 0$, $H_1 = 24$ for $p = q = 2$ and (ii) $H_0 = 0$, $H_1 = 6$ for $p = \infty, q = 1$ and $p = 1, q = \infty$. Furthermore, $f(\theta)$ satisfies Assn. 3 for all $q \geq 1$ with $\tau = 1$, $\mu(\theta) = \pi_\theta(a^*)$ and $f^* = 0$.*

The above result generalizes that in Mei et al. (2021) beyond Euclidean norms. By following a similar analysis as Vaswani & Harikandeh (2025, Proposition 7), we can extend this result from bandits to Markov decision processes. The proposition below shows that certain two-layer networks (with restrictions on the activation function) satisfy Assn. 2 and 3.

**Proposition 4.** *Consider $n$ points where $x_i \in \mathbb{R}^D$ are the features and $y_i \in \{-1, 1\}$ are the corresponding labels, and a neural network,*

$$\Phi(\theta, x) := \sum_{j=1}^m a_j \, \sigma(\langle \theta_j, x \rangle), \tag{7}$$

*where $a_j$ are fixed, $m$ is the width of the layer and $\sigma$ is the activation function. Consider the case when $\sigma$ is smooth s.t. for all $t$, $|\sigma''(t)| \leq M$ and has bounded derivatives i.e. there exists positive constants $\alpha_1, \alpha_2$ such that $\alpha_1 \leq |\sigma'(t)| \leq \alpha_2$. Consider the loss*

$$f(\theta) := \frac{1}{n} \sum_{i=1}^{n} g\left(y_i \, \Phi(\theta, x_i)\right), \tag{8}$$

*where, $g : \mathbb{R} \to \mathbb{R}$ is differentiable everywhere and for all $s$, $g(s) \geq 0$, $g'(s) \leq 0$, $\frac{|g'(s)|}{g(s)} \in [c_1, c_2]$ and $|g''(s)| \leq c_2' \, g(s)$. $f$ satisfies Assn. 1 and Assn. 2 with $H_0 = 0$ and $H_1 = \left[c_2 \, M \, \|a\|_1 + c_2' \, \alpha_2^2 \, \|a\|_q^2\right] \max_i \|x_i\|_q^2$. Furthermore, if the data is linearly separable with a normalized margin $\gamma_p := \max_\theta \min_i \frac{y_i \langle x_i, \theta \rangle}{\|\theta\|_p} > 0$ then, $f(\theta)$ satisfies Assn. 3 with $\tau = 1$, $\mu = \frac{c_1 \, \alpha_1 \, \gamma \, \|a\|_2^2}{\|a\|_p}$ and $f^* = 0$.*

The conditions on $\sigma(t)$ are satisfied for a smoothed variant of the leaky ReLU function, whereas the condition on $g$ is satisfied by the exponential loss. This result generalizes Taheri & Thrampoulidis (2023, Lemmas 3 & 5) beyond $\ell_2$ norms and the exponential loss.

Furthermore, recent work (Alimisis et al., 2025, Proposition 3.3) shows that two-layer neural networks with $\ell_2$-regularization and weaker assumptions on the activation function satisfy Assn. 2 for $p = q = 2$, and $H_0 \neq 0$ and $H_1 \neq 0$. In addition, this work provides some empirical evidence verifying that Assn. 2 holds when training language models. On the other hand, we note that sufficiently over-parameterized neural networks (Liu et al., 2022; Soltanolkotabi et al., 2018; Zou & Gu, 2019; Li & Liang, 2018) are known to satisfy Assn. 3 with $\tau = \frac{1}{2}$ and $(p, q) = (2, 2)$.

Finally, in Prop. 6 in App. B, we show that generalized linear models with the logistic link function also satisfy Assn. 2 with non-zero $H_0$ and $H_1$, and satisfy Assn. 3 with $\tau = \frac{1}{2}$.

## 3. Properties of $(H_0, H_1)$-NS Functions

In this section, we develop properties of $(H_0, H_1)$-NS functions that will be crucial in the subsequent analyses. We defer all the proofs to App. C.

In Lemma. 3, we first prove that for functions satisfying Assn. 1 and 2, the gradient can be bounded in terms of the function value, i.e., for all $\theta$,

$$\|\nabla f(\theta)\|_q \leq \sqrt{2H_0 \, f(\theta) + H_1 \, [f(\theta)]^2} \tag{9}$$

If $H_1 = 0$ and $(p, q) = (2, 2)$, the above inequality implies that the squared Euclidean norm of the gradient is bounded by the function value, a standard result for uniformly smooth functions. On the other hand, if $H_0 = 0$, then Eq. (9) simplifies to $\|\nabla f(\theta)\|_q \leq \sqrt{H_1} \, f(\theta)$, implying that $\ln(f(\theta))$ is $\sqrt{H_1}$-Lipschitz.

In Lemma. 5, we generalize this property and prove that an appropriately shifted $f$ is uniformly Lipschitz in a *multiplicative sense*, i.e. $\forall y, x$ and $H_1 > 0$,

$$f(y) + \frac{H_0}{H_1} \leq \left(f(x) + \frac{H_0}{H_1}\right) \exp\left(\sqrt{H_1} \, \|y - x\|_p\right) \tag{10}$$

If $H_0 = 0$, Eq. (10) implies that as $y \to x$, the ratio $f(y)/f(x) \to 1$. This multiplicative Lipschitzness of the function will be helpful in the subsequent analysis. Moreover, it enables us to prove that the gradients are Lipschitz in the usual additive sense, and the gradient norms are Lipschitz in a multiplicative sense similar to Eq. (10). In particular, in Lemmas. 6 and 10, we prove that $\forall y, x$ such that $\|y - x\|_p \leq \frac{1}{\sqrt{H_1}}$ and $c > 0$,

$$\|\nabla f(y) - \nabla f(x)\|_q \leq e \, [H_0 + H_1 \, f(x)] \, \|y - x\|_p \tag{11}$$

$$\left(\|\nabla f(y)\|_q + c\right) \leq \left(\|\nabla f(x)\|_q + c\right) \tag{12}$$
$$\times \exp\left((H_0 + H_1 f(x)) \frac{e \, \|y - x\|_p}{c}\right)$$

Hence, Eqs. (10) and (12) enable bounding both the (appropriately shifted) function and gradient norm at $y$ in terms of $x$, a nearby point in the $\ell_p$ norm. These properties will be particularly important when we analyze algorithms. Finally, we prove the *descent lemma* for NS functions, implying that they can be upper-bounded in terms of a quadratic. In particular, if $\|y - x\|_p \leq \frac{1}{\sqrt{H_1}}$,

$$f(y) \leq f(x) + \langle \nabla f(x), y - x \rangle + (H_0 + H_1 \, f(x)) \, \|y - x\|_p^2 \tag{13}$$

Eq. (13) will serve as the starting point of all our analyses in Secs. 4 and 5. Finally, we emphasize that unlike Eq. (10), Eqs. (11) to (13) are non-uniform in that the constant depends on $f(x)$, and hold for $y, x$ that are sufficiently close in the $\ell_p$ norm.

## 4. Convergence of Steepest Descent

In this section, we characterize the convergence of steepest descent methods on functions satisfying Assn. 1 to 3. We then highlight some important practical consequences of our result to softmax policy gradient and logistic regression for separable data.

We focus on normalized Steepest Descent (NSD) (Boyd & Vandenberghe, 2004) which has the following update:

$$\theta_{t+1} = \theta_t - \eta_t \, d_t \tag{NSD}$$
$$\text{where, } d_t := \arg\max_{\|d\|_p \leq 1} \langle d, \nabla f(\theta_t) \rangle \tag{14}$$

We will be particularly interested in $(p, q) = (\infty, 1)$, $(2, 2)$ and $(1, \infty)$. In these special cases, the update in Eq. (14) can be simplified and recovers sign gradient descent (Sign GD), normalized gradient descent (Norm. GD) and sign coordinate descent with the Gauss-Southwell rule (Sign CD-GS) respectively (see Prop. 7 in App. D for a proof). In particular, we define $\nabla_t := \nabla f(\theta_t)$ with $\nabla_{t,i}$ denoting coordinate $i$ of this vector, use $\text{sign}(\nabla_t) \in \{-1, 0, 1\}^D$ to denote the element-wise sign operation with $\text{sign}(0) := 0$ and let $e_i$ denote the $i$-th standard basis vector. For,

- $(p, q) = (\infty, 1)$, $\theta_{t+1} = \theta_t - \eta_t \, \text{sign}(\nabla_t)$ (Sign GD)

- $(p, q) = (2, 2)$, $\theta_{t+1} = \theta_t - \eta_t \frac{\nabla_t}{\|\nabla_t\|_2}$ (Norm.GD)

- $(p, q) = (1, \infty)$, $\theta_{t+1} = \theta_t - \eta_t \, \text{sign}(\nabla_{t,i_t}) \, e_{i_t}$, where, $i_t \in \arg\max_{i \in [D]} |\nabla_{t,i}|$ (Sign CD-GS)

We will subsequently use these special cases while discussing the practical implications of NSD. Next, we present the theorem analyzing the convergence of NSD on functions satisfying Assn. 1 and 2 and Assn. 3 with $\mu(\theta) = \mu$ and $f^* = 0$.

**Theorem 1.** *Under Assn. 1, Assn. 2 and Assn. 3 with $\mu(\theta) = \mu$ for all $\theta$, $f^* = 0$, NSD converges as:*

- *if $\epsilon > \frac{H_0}{H_1}$, using $\eta_t = \eta = O(1)$ guarantees that after $T = O\left(\ln\left(\frac{1}{\epsilon}\right)\right)$ iterations, $f(\theta_{T+1}) \leq \epsilon$;*

- *else, using $\eta_t = \eta = O(\epsilon^\tau)$ guarantees that after $T = O\left(\frac{1}{\epsilon^\tau}\right)$ iterations, $f(\theta_{T+1}) \leq \epsilon$.*

In order to interpret the above theorem, let us first consider the setting corresponding to $H_0 > 0$ and $H_1 = 0$. This corresponds to uniform smoothness and implies that NSD attains an $O\left(\frac{1}{\epsilon^\tau}\right)$ rate. For strongly-convex quadratics, $\tau = \frac{1}{2}$, and in this case, the $\Theta\left(\frac{1}{\sqrt{\epsilon}}\right)$ convergence for NSD (see Prop. 8 for the corresponding lower-bound) is slower than the linear convergence rate of GD. On the other hand, if we consider the other extreme – when $H_0 = 0$ and $H_1 > 0$, NSD converges at a linear rate for all values of $\tau$ and $\epsilon$. As mentioned in Sec. 2.1, such a property is satisfied by binary classification with the exponential loss on separable data. In this case, GD attains a sublinear $\Theta\left(\frac{1}{\epsilon}\right)$ rate (Soudry et al., 2018), and is provably slower than NSD.

In general, for non-zero values of $H_0, H_1$, NSD results in a faster linear convergence rate when an $O\left(\frac{H_0}{H_1}\right)$ sub-optimality is acceptable. On the other hand, for $\epsilon < \frac{H_0}{H_1}$, NSD converges in two phases – a fast, first phase when the method converges to an $O\left(\frac{H_0}{H_1}\right)$ neighbourhood, followed by a slower second phase. Intuitively, in the second phase, as the iterates get closer to the optimum, guaranteeing convergence requires using a smaller step-size so as not to "overshoot" the minimizer. The convergence in this second phase depends on the value of $\tau$ and is slower as $\tau \to 1$.

On the other hand, for losses such as the exponential loss on separable data, the minimum is achieved as $\|\theta\| \to \infty$. In this case, since there is no finite minimizer, NSD can use a constant step-size throughout, resulting in a faster linear rate.

**Proof Sketch:** Using Eq. (13) for the NSD update at iteration $t$ with $\eta \leq \frac{1}{\sqrt{H_1}}$ and noting that $\|d_t\|_p^2 \leq 1$,

$$f(\theta_{t+1}) \leq f(\theta_t) - \eta \langle \nabla f(\theta_t), d_t \rangle + (H_0 + H_1 \, f(\theta_t)) \, \eta^2$$

Using the definition of the dual norm to simplify $\langle \nabla_t, d_t \rangle = \|\nabla_t\|_q$ and Assn. 3 with $f^* = 0$ to bound the gradient in terms of the function value, we get that,

$$f(\theta_{t+1}) \leq f(\theta_t) - \eta \, \mu \, [f(\theta_t)]^\tau + (H_0 + H_1 \, f(\theta_t)) \, \eta^2.$$

The subsequent proof proceeds in two phases. We define $T_0$ as the first iteration s.t. $f(\theta_{T_0}) < \max\left\{\epsilon, \frac{H_0}{H_1}\right\}$.

*Phase 1:* For $t < T_0$, $f(\theta_t) \geq \max\left\{\epsilon, \frac{H_0}{H_1}\right\}$ and consequently, $H_0 + H_1 \, f(\theta_t) \leq 2 \, H_1 \, f(\theta_t)$. Using this relation to simplify the above inequality,

$$f(\theta_{t+1}) \leq f(\theta_t) - \eta \, \mu \, [f(\theta_t)]^\tau + 2 \, H_1 \, f(\theta_t) \, \eta^2. \quad (15)$$

By setting an appropriate $\eta = O(1)$, we inductively prove that $f(\theta_t) < f(\theta_1)$ for all $t < T_0$, and bound $[f(\theta_t)]^\tau$ in Eq. (15) by $f(\theta_t)/[f(\theta_1)]^{1-\tau}$. Solving the resulting recursion using Lemma 13 immediately implies a linear convergence rate.

*Phase 2:* When $\epsilon < \frac{H_0}{H_1}$, consider $t > T_0$ s.t. $f(\theta_t) < \frac{H_0}{H_1}$. In this case, we bound $H_0 + H_1 \, f(\theta_t) \leq 2 \, H_0$, and show that setting an appropriate $\eta = O(\epsilon^\tau)$ implies descent meaning that $f(\theta_{t+1}) \leq f(\theta_t) \leq \frac{H_0}{H_1}$ for all $t > T_0$. Solving the resulting recursion using Lemma 14 with $r = \tau$ implies an $O(1/\epsilon^\tau)$ rate. $\square$

**Remark 1.** *The $f^* = 0$ assumption is made w.l.o.g. In particular, if $f^* \neq 0$, we can analyze the shifted function $h(\theta) = f(\theta) - f^*$ s.t. $h^* = 0$ and $\nabla h(\theta) = \nabla f(\theta)$. Since NSD and all subsequent algorithms only depend on the gradients, they produce the same iterates on both $h$ and $f$. In Lemma 11, we prove that if $f$ is $(H_0, H_1)$-NS, then, $h$ is $(H_0 + H_1 \, f^*, H_1)$-NS.*

### 4.1. Implications

Thm. 1 applies to all objectives in Sec. 2.1 and implies the efficacy of Sign GD, Sign CD-GS and Norm.GD. Below, we highlight some practical consequences.

*Implication 1:* Recall from Sec. 2.1 that the softmax policy gradient objective satisfies Assn. 2 with $H_0 = 0$ and Assn. 3 with $\tau = 1$ and $f^* = 0$, though with a non-uniform $\mu(\theta)$. In App. D, we handle this non-uniformity and prove the following corollary.

**Corollary 1.** *For the multi-armed bandit problem in Prop. 3, NSD with a uniform initialization i.e. $\forall a, \pi_{\theta_1}(a) = 1/K$ and $\eta = O(1)$ requires $T = O\left(\ln\left(1/\epsilon\right)\right)$ iterations to guarantee $\langle \pi_{\theta_{T+1}}, r \rangle \geq r(a^*) - \epsilon$.*

The above result is in contrast to constant step-size GD which can only attain an $\Omega\left(\frac{1}{\epsilon}\right)$ convergence rate for this problem (Mei et al., 2020, Theorem 9). Since NSD includes Norm.GD, Cor. 1 recovers the result in Mei et al. (2021). Moreover, since NSD also includes Sign GD and Sign CD-GS, the above result implies that these methods can match the convergence rate of algorithms designed for this specific problem, including GD with a line-search (Lu et al.; Vaswani & Harikandeh, 2025), GD with specific increasing step-sizes (Liu et al., 2024), natural policy gradient (Kakade & Langford, 2002; Xiao, 2022) and mirror descent with a log-sum-exp mirror map (Asad et al., 2025).

*Implication 2:* Logistic regression on separable data is a canonical example in machine learning, and has been the focus of recent works (Wu et al., 2024; Zhang et al., 2025) analyzing the impact of large constant and adaptive step-sizes. More recently, Vaswani & Harikandeh (2025) prove that GD with an Armijo line-search can attain a linear convergence rate for this problem.

We now prove that NSD and consequently, Sign GD and Norm.GD can match this linear rate while using a constant step-size. We first note that the objective in Prop. 2 satisfies Assn. 3 with $\tau = 1$ only when the loss is below $\ln(2)/n$. Consequently, Thm. 1 does not directly apply to this objective, and we handle this in App. D, proving the following theorem.

**Theorem 2.** *For logistic regression on separable data with the margin $\gamma_p$ (see Prop. 2), NSD with $\theta_1 = 0$, $\eta_t = \eta = O(1)$ and $T = O\left(\frac{H_1}{\gamma_p^2}\left[n^2 + \ln\left(\frac{1}{n\epsilon}\right)\right]\right)$ iterations guarantees that $f(\theta_{T+1}) \leq \epsilon$.*

We note that after $O\left(n^2/\gamma_p^2\right)$ burn-in iterations, the loss is below the $\frac{\ln(2)}{n}$ threshold, the objective satisfies Assn. 3 and consequently, NSD converges at a linear rate. Finally, we note that for this objective, Axiotis & Sviridenko (2023) prove that a normalized variant of coordinate descent with greedy Gauss-Southwell selection achieves a linear convergence rate. Thm. 2 implies that Sign CD-GS, which uses the Gauss–Southwell rule to select the coordinate, and updates it using the sign of its gradient, is another example of a coordinate descent method that can achieve linear convergence.

## 5. Convergence of **RMSProp** & **Adam**

In this section, we provide general convergence theorems for RMSProp and Adam on functions satisfying Assn. 1 to 3. In Sec. 5.1, we first analyze RMSProp, highlighting

the additional challenges compared to NSD. In Sec. 5.2, we turn to Adam and explain how to handle the additional momentum term. Subsequently, in Sec. 5.3, we highlight some practical implications of our results. Finally, in Sec. 5.4, we provide an additional result characterizing the stationary point convergence of Adam for non-convex, $(L_0, L_1)$ NS functions (Zhang et al., 2020b) that do not not necessarily satisfy Assn. 3.

### 5.1. **RMSProp**

We analyze RMSProp (Tieleman, 2012) whose update is given as: if $g_{t,i} := \nabla_{t,i}$, $v_{0,i} = 0$ for all $i \in [D]$ and for $\delta \geq 0$ and $t \geq 1$,

$$\theta_{t+1} = \theta_t - \eta_t \, d_t \;\; \text{s.t.,} \, \forall i \in [D] \,, \; d_{t,i} = \frac{g_{t,i}}{\sqrt{v_{t,i}} + \delta} \quad (16)$$

$$v_{t,i} = (1-\beta)\sum_{s=1}^{t} \beta^{t-s} g_{s,i}^2 = \beta \, v_{t-1,i} + (1-\beta) g_{t,i}^2$$

We now analyze the convergence of RMSProp.

**Theorem 3.** *Under Assn. 1, Assn. 2 and Assn. 3 with $f^* = 0$, $\mu(\theta) = \mu$ for all $\theta$, RMSProp with $\delta = 0$ converges as:*

• *If $\epsilon > \frac{H_0}{H_1}$, using $\eta_t = \eta = O(1)$ guarantees that after $T = O\left(\ln\left(\frac{1}{\epsilon}\right)\right)$ iterations, $f(\theta_{T+1}) \leq \epsilon$.*

• *Else, using $\eta_t = \eta = O(\epsilon^{2\tau})$ guarantees that $f(\theta_{T+1}) \leq \epsilon$ after $T$ iterations, where,*

   •*If $\tau \leq \frac{1}{2}$, $T = O\left(\frac{1}{\epsilon^{2\tau}}\right)$.*

   •*Else, if $\tau > \frac{1}{2}$, $T = O\left(\frac{1}{\epsilon^{4\tau - 1}}\right)$.*

**Proof Sketch:** We use Eq. (13) with $p = \infty$ and $q = 1$ for the RMSProp update at iteration $t$. Noting that $\|d_t\|_\infty \leq \frac{1}{\sqrt{1-\beta}}$ (see Lemma. 16) and using an appropriate $\eta$ gives us the following inequality,

$$f(\theta_{t+1}) \leq f(\theta_t) - \eta \langle \nabla_t, d_t \rangle + \bar{L}_t \frac{\eta^2}{1-\beta}\,,$$

where $\bar{L}_t := H_0 + H_1 f(\theta_t)$. To lower-bound $\langle \nabla_t, d_t \rangle = \sum_i \frac{g_{t,i}^2}{\sqrt{v_{t,i}}}$, we prove Lemma. 17, which uses the Cauchy-Schwarz inequality with the $v_{t,i}$ update to get that,

$$\langle \nabla_t, d_t \rangle \geq \|\nabla_t\|_1 \underbrace{\frac{\|\nabla_t\|_1}{\sqrt{1-\beta}\sum_{j=0}^{t-1}(\sqrt{\beta})^j \|\nabla_{t-j}\|_1}}_{:=(*)}$$

Combining the above inequalities, we get that,

$$f(\theta_{t+1}) \leq f(\theta_t) - \eta \, \|\nabla_t\|_1 \, (*) + \bar{L}_t \frac{\eta^2}{1-\beta}\,. \quad (17)$$

The (*) term quantifies the effect of the RMSProp preconditioner, and involves the ratio of gradients evaluated at

different points. To complete the proof, we split the analysis in two phases similar to Thm. 1, and define $T_0$ as the first iteration s.t. $f(\theta_{T_0}) < \max\left\{\epsilon, \frac{H_0}{H_1}\right\}$.

*Phase 1:* Consider $t < T_0$. In this phase, we inductively prove that RMSProp results in descent and consequently, $f(\theta_t) \leq f(\theta_1)$ for all $t < T_0$. In particular, given the inductive hypothesis at iteration $t$, we use Eq. (12) for $x = \theta_t, y = \theta_{t-1}, c > 0$ to relate $\|\nabla_t\|_1$ and $\|\nabla_{t-1}\|_1$,

$$\|\nabla_{t-1}\|_1 + c \leq (\|\nabla_t\|_1 + c)$$
$$\times \exp\left((H_0 + H_1 f(\theta_1)) \frac{e \|\theta_{t-1} - \theta_t\|_\infty}{c}\right).$$

Using a sufficiently small $\eta$ and successively using the above inequality for $j \in \{0, 1, \ldots, t-1\}$ enables us to obtain the following bound for a constant $a > 0$ that depends on $\beta$,

$$\|\nabla_{t-j}\|_1 \leq (\|\nabla_t\|_1 + c) \exp(a j)$$

$$\implies \sum_{j=0}^{t-1} (\sqrt{\beta})^j \|\nabla_{t-j}\|_1 \leq (\|\nabla_t\|_1 + c) \sum_{j=0}^{t-1} (\sqrt{\beta} e^a)^j$$

Ensuring that $\sqrt{\beta} \exp(a) < 1$ enables us to bound the geometric series. Simplifying and using Assn. 3 with the appropriate value of $c$ gives the final bound on (*),

$$(*) \geq \Omega\left(\frac{\|\nabla_t\|_1}{\|\nabla_t\|_1 + c}\right) \geq \Omega\left(\left[1 + \left(\frac{H_0}{H_1} \frac{1}{f(\theta_t)}\right)^\tau\right]^{-1}\right).$$

Since $f(\theta_t) \geq H_0/H_1$ for all $t$ in Phase 1, $(*) = \Omega(1)$ and $\bar{L}_t = O(f(\theta_t))$. Simplifying Eq. (17) for Phase 1, and using Assn. 3 to simplify the $\|\nabla_t\|_1$ term, we get that,

$$f(\theta_{t+1}) \leq f(\theta_t) - \|\nabla_t\|_1 O(\eta) + O(\eta^2) f(\theta_t)$$
$$\implies f(\theta_{t+1}) \leq f(\theta_t) - O(\eta) [f(\theta_t)]^\tau + O(\eta^2) f(\theta_t)$$

Setting an appropriate $\eta$ ensures descent and completes the induction. The rest of the proof for Phase 1 is the same for Thm. 1, and results in linear convergence.

*Phase 2:* Similar to Phase 1, we inductively prove that for all $t > T_0$, $f(\theta_{t+1}) \leq f(\theta_t) \leq f(\theta_{T_0}) \leq \frac{H_0}{H_1} \leq f(\theta_1)$. For this phase, since $f(\theta_t) \leq H_0/H_1$ by the inductive hypothesis, we get that $(*) = \Omega([f(\theta_t)]^\tau)$ and $\bar{L}_t = O(1)$. Simplifying Eq. (17) analogous to Thm. 1,

$$f(\theta_{t+1}) \leq f(\theta_t) - O(\eta) [f(\theta_t)]^{2\tau} + O(\eta^2)$$

The rest of the proof for Phase 2 proceeds as that for Thm. 1, with the only change being the $2\tau$ exponent on $f(\theta_t)$ (rather than $\tau$ for the NSD proof). Solving the above recursion using Lemma. 14 finishes the proof. $\square$

Before we interpret the above result, we first show how to handle the momentum term in Adam, and prove that it attains the same convergence rate as RMSProp.

## 5.2. Adam

We are now ready to analyze Adam (Kingma & Ba, 2015) without bias correction[1]. The resulting update is given as follows: if $g_{t,i} := \nabla_{t,i}$, $v_{0,i} = 0$ and $m_{0,i} = g_{1,i}$ for all $i \in [D]$, then, for $\delta \geq 0$ and $t \geq 1$,

$$\theta_{t+1} = \theta_t - \eta_t d_t \text{ s.t., } \forall i \in [D], d_{t,i} = \frac{m_{t,i}}{\sqrt{v_{t,i}} + \delta} \quad (18)$$

$$m_{t,i} = (1 - \beta_1) \sum_{s=1}^{t} \beta_1^{t-s} g_{s,i} = \beta_1 m_{t-1,i} + (1 - \beta_1) g_{t,i}$$

$$v_{t,i} = (1 - \beta_2) \sum_{s=1}^{t} \beta_2^{t-s} g_{s,i}^2 = \beta_2 v_{t-1,i} + (1 - \beta_2) g_{t,i}^2$$

In Sec. 5.2, we characterize the convergence of Adam and prove the following theorem.

**Theorem 4.** *Under Assn. 1, Assn. 2 with $H_0 \geq 0$ and $H_1 > 0$, and Assn. 3 with $f^* = 0$ and $\mu(\theta) = \mu$ for all $\theta$, Adam with the update in Eq. (18) with $\beta_1 \leq \beta_2$ has the following convergence rate:*

• *If $\epsilon > \frac{H_0}{H_1}$, using $\eta_t = \eta = O(1)$ guarantees that after $T = O\left(\ln\left(\frac{1}{\epsilon}\right)\right)$ iterations, $f(\theta_{T+1}) \leq \epsilon$.*

• *Else, using $\eta_t = \eta = O(\epsilon^{2\tau})$ guarantees that $f(\theta_{T+1}) \leq \epsilon$ after $T$ iterations, where,*

• *If $\tau \leq \frac{1}{2}$, $T = O\left(\frac{1}{\epsilon^{2\tau}}\right)$.*

• *Else, if $\tau > \frac{1}{2}$, $T = O\left(\frac{1}{\epsilon^{4\tau-1}}\right)$.*

**Proof Sketch:** As with Thm. 3, we begin with Eq. (13) with $p = \infty$ and $q = 1$. In Lemma. 19, we show that if $\beta_1 \leq \beta_2$, $\|d_t\|_\infty = O(1)$. We define $\bar{L}_t := H_0 + H_1 f(\theta_t)$ and obtain the following inequality,

$$f(\theta_{t+1}) \leq f(\theta_t) - \eta \langle \nabla_t, d_t \rangle + \bar{L}_t O(\eta^2) \quad (19)$$

Bounding $-\langle \nabla_t, d_t \rangle$,

$$\langle \nabla_t, d_t \rangle = \sum_i \frac{g_{t,i} m_{t,i}}{\sqrt{v_{t,i}}} = \sum_i \frac{g_{t,i} (m_{t,i} - g_{t,i})}{\sqrt{v_{t,i}}} + \frac{g_{t,i}^2}{\sqrt{v_{t,i}}}$$

$$\implies -\langle \nabla_t, d_t \rangle \leq \underbrace{\sum_i \frac{|g_{t,i}| |m_{t,i} - g_{t,i}|}{\sqrt{v_{t,i}}}}_{:= \text{Term (i)}} - \underbrace{\sum_i \frac{g_{t,i}^2}{\sqrt{v_{t,i}}}}_{:= \text{Term (ii)}}$$

Term (i) depends on the momentum, and we bound it using Lemma. 20. In particular, by lower-bounding $v_{t,i}$ in terms of $g_{t,i}$, we obtain that,

$$\text{Term (i)} \leq \frac{\|m_t - g_t\|_1}{\sqrt{1 - \beta_2}}. \quad (20)$$

---

[1]The effect of bias correction decays exponentially fast.

By using the momentum update, we get that,

$$\|m_t - g_t\|_1 \leq \beta_1 \|m_{t-1} - g_{t-1}\|_1 + \beta_1 \|\nabla_t - \nabla_{t-1}\|_1$$

Using Eq. (11) to bound the difference between consecutive gradients with an appropriate step-size,

$$\|\nabla_t - \nabla_{t-1}\|_1 \leq O(\eta)\,\bar{L}_t$$

Combining the above two inequalities, we get that,

$$\frac{\|m_t - g_t\|_1}{\bar{L}_t} \leq \beta_1 \frac{\|m_{t-1} - g_{t-1}\|_1}{\bar{L}_t} + \beta_1\, O(\eta)$$

We now use Eq. (10) to write $\bar{L}_t$ in terms of $\bar{L}_{t-1}$,

$$\bar{L}_t = H_0 + H_1\, f(\theta_t) \geq (H_0 + H_1\, f(\theta_{t-1}))$$
$$\times \exp\left(-\sqrt{H_1}\,\|\theta_t - \theta_{t-1}\|_\infty\right)$$

With an appropriate step-size, $\frac{1}{\bar{L}_t} \leq \frac{1}{\bar{L}_{t-1}} \frac{1}{\sqrt{\beta_1}}$,

$$\implies \frac{\|m_t - g_t\|_1}{\bar{L}_t} \leq \sqrt{\beta_1}\frac{\|m_{t-1} - g_{t-1}\|_1}{\bar{L}_{t-1}} + \beta_1\, O(\eta)$$

Solving the above recursion and using Eq. (20), we get that Term (i) $\leq \bar{L}_t\, O(\eta)$.

$$\implies -\eta\,\langle \nabla_t, d_t\rangle \leq \bar{L}_t\, O(\eta^2) - \eta\,(\text{Term (ii)})$$

Combining with Eq. (19),

$$f(\theta_{t+1}) \leq f(\theta_t) - \eta\,(\text{Term (ii)}) + \bar{L}_t\, O(\eta^2)$$

The term that depends on the momentum is of the same order as the third term in Eq. (19), and is absorbed into it. Term (ii) does not depend on momentum, and is the same as in the RMSProp proof. Up to constants, the remaining proof is exactly the same as that for RMSProp, and results in the same rate. $\square$

Importantly, our analysis does not require convexity or bounded gradients, and the resulting rate is dimension-independent. The above results show that, similar to NSD, RMSProp and Adam result in linear convergence rates when $\epsilon = O\left(H_0/H_1\right)$. In this regime, these methods can converge to the desired sub-optimality with an $O(1)$ step-size and $\delta = 0$. For $\epsilon < H_0/H_1$, RMSProp and Adam can still converge with $\delta = 0$, provided that the step-size is sufficiently small (of the order $\epsilon^{2\tau}$). In this case, these methods attain a slower sub-linear rate.

### 5.3. Implications

The above results apply to all objectives in Sec. 2.1. Below, we highlight some practical consequences.

*Implication 1:* For neural network objectives satisfying Assn. 2 and the PL condition (Assn. 3 with $\tau = 1/2$),

Adam and RMSProp (corresponding to Adam with $\beta_1 = 0$) can converge at a linear rate for $\epsilon = O\left(H_0/H_1\right)$, and at an $O\left(1/\epsilon\right)$ rate for smaller $\epsilon$. As mentioned in Sec. 2.1, this includes over-parameterized neural networks.

*Implication 2:* For binary classification using either a linear model (Prop. 1) or a two-layer neural network (Prop. 4), when minimizing the exponential loss on separable data, RMSProp and Adam can use an $O(1)$ step-size and result in a linear convergence rate.

*Implication 3:* In Thm. 7, we prove that both RMSProp and Adam can achieve linear convergence for logistic regression on separable data. Given the sub-linear lower-bounds for GD, this provides a concrete setting in which RMSProp and Adam are provably faster. Furthermore, by using a similar argument as in Cor. 1, we conjecture that RMSProp and Adam can attain a linear rate for the softmax policy gradient objective.

### 5.4. Handling $(L_0, L_1)$-NS Functions

In App. H, we analyze the convergence of Adam for non-convex functions satisfying the $(L_0, L_1)$-NS condition in Eq. (2). Unlike the above result, the next theorem does not rely on the NL-assumption in Assn. 3.

**Theorem 5.** *Under Assn. 1 and Assn. 4 with $L_0 \geq 0$ and $L_1 > 0$, and $f^* = 0$, Adam with the update in Eq. (18) has the following convergence rate:*

- *If $\epsilon \geq \frac{L_0}{L_1}$, using $\eta_t = \eta = O(1)$ guarantees that after $T = O\left(\frac{1}{\epsilon}\right)$ iterations, $\|\nabla f(\theta_T)\|_1 \leq \epsilon$.*

- *Else, using $\eta_t = \eta = O(\epsilon^2)$ guarantees that after $T = O\left(\frac{1}{\epsilon^2}\right)$ iterations, $\min_{t \leq T} \|\nabla f(\theta_t)\|_1 \leq \epsilon$.*

In contrast to Wang et al. (2024a) that only considers the scalar or norm version of the update, we consider the standard, diagonal variant of Adam. The above result requires Eq. (2), the generalization of the assumption in Zhang et al. (2020b) to $p, q$ norms. In particular, for large $\epsilon \geq \frac{L_0}{L_1}$, the above rate is faster than the $O\left(\frac{1}{\epsilon^2}\right)$ rate in Wang et al. (2024a, Theorem 1). For small $\epsilon < \frac{L_0}{L_1}$, the resulting $O\left(\frac{1}{\epsilon^2}\right)$ matches that of Wang et al. (2024a) and the standard non-convex convergence rate for GD.

## 6. Inefficiency of AdaGrad & AMSGrad

In Sec. 5, we have identified $(0, H_1)$-NS functions satisfying Assn. 3 as examples where RMSProp and Adam achieve linear convergence rates. The one-dimensional logistic loss function, $f(\theta) = \ln(1 + \exp(-\theta))$ is such an example, and satisfies Assn. 1, Assn. 2 with $H_0 = 0$, $H_1 = 1$ and Assn. 3 with $\tau = 1$, $\mu = 1$, $f^* = 0$. For this loss function, we now prove that other adaptive gradient methods such as AdaGrad (Duchi et al., 2011) and

AMSGrad (Reddi et al., 2018) cannot achieve this faster linear convergence. In particular, we present the following $\Omega\left(\frac{1}{T}\right)$ lower-bound (proved in App. G).

**Theorem 6.** *Starting from $\theta_1 = 0$, consider $T$ iterations of the form $\theta_{t+1} = \theta_t - \eta_t\, m_t$ s.t. (i) the effective step-size $\eta_t$ is bounded, non-increasing and independent of $T$, (ii) for $\beta \in [0, 1)$, $m_t = (1 - \beta)\sum_{s=1}^{t}\beta^{t-s}\,\nabla f(\theta_s)$ is the momentum vector and (iii) $\eta_1 \le \ln\left(1/\beta\right)$. When minimizing the logistic loss with these update restrictions, the convergence rate is lower-bounded by $\Omega(1/T)$.*

In App. G.1, we show that constant step-size GD and heavy-ball momentum with an appropriate constant step-size satisfy the conditions in Thm. 6. Our result complements the lower-bound for GD in Wu et al. (2024), and implies that for this class of functions, heavy-ball momentum cannot improve the convergence.

Furthermore, we show that AdaGrad (Duchi et al., 2011) with any $O(1)$ step-size as well as AMSGrad (Reddi et al., 2018) with an appropriate constant step-size also satisfy the conditions in Thm. 6, and consequently have an $\Omega(1/T)$ lower-bound. Unlike these methods, RMSProp and Adam do not ensure that the effective step-size is non-increasing and hence do not satisfy condition (i) in Thm. 6. Consequently, they do not suffer from this slower convergence. In fact, in Thm. 7, we have shown that RMSProp and Adam (with similar restrictions on the step-size as in Thm. 6) achieve an $O(\exp(-T))$ convergence rate. Hence, we have identified a natural class of functions where RMSProp and Adam are provably faster than AdaGrad and AMSGrad.

## 7. Conclusion

We analyzed the convergence of steepest descent, and deterministic RMSProp and Adam on non-uniform smooth functions satisfying gradient domination. Our results imply the fast convergence of these methods for certain neural networks and policy gradient objectives. Furthermore, we proved a separation, identifying practical objectives where RMSProp and Adam are provably faster than GD, AdaGrad and AMSGrad.

We believe that the same techniques can be used to analyze the convergence of related methods such as steepest descent with momentum (Bernstein et al., 2018) and deterministic variants of generalized sign GD (Crawshaw et al., 2022) and Lion (Chen et al., 2023a). In the future, we plan to generalize our results to the stochastic setting, and study a broader class of non-convex functions.

## Impact Statement

This paper presents work whose goal is to advance the field of Machine Learning. There are many potential societal consequences of our work, none which we feel must be specifically highlighted here.

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

# Appendix

## A. Related Work

In this section, we present a more detailed review of the related work. Previous work can be characterized into two broad categories:

- **Convergence under** $(H_0, H_1)$ **non-uniform smoothness**: The most relevant papers are Vaswani & Harikandeh (2025); Alimisis et al. (2025) that consider the same non-uniform assumption as we do. Vaswani & Harikandeh (2025) use this assumption to justify the use of Armijo line-search, while Alimisis et al. (2025) use this assumption to justify the importance of learning-rate warm-up. In terms of convergence guarantees, Vaswani & Harikandeh (2025) analyze GD with Armijo line-search and Alimisis et al. (2025) analyze normalized GD. Both papers derive similar convergence guarantees as Thm. 1 in our paper. However, both these papers are inherently in the Euclidean setting, and do not derive dimension-free guarantees for general (normalized) steepest descent, nor do they consider RMSProp or Adam. In App. I, we generalize the result in Vaswani & Harikandeh (2025) to steepest descent with Armijo line-search, while substantially simplifying their proof.

- **Convergence under** $(L_c, L_g)$ **non-uniform smoothness**: (Zhang et al., 2020b; Gorbunov et al., 2025; Vankov et al., 2025; Li et al., 2023b; Wang et al., 2024a;b) consider a different non-uniform smoothness assumption in Eq. (2). This assumption only considers Euclidean norms, and is in general, stronger than our $(H_0, H_1)$ assumption (Alimisis et al., 2025). Importantly, this assumption cannot model even the exponential or logistic regression loss (Vaswani & Harikandeh, 2025).

  Under this assumption (Zhang et al., 2020b; Gorbunov et al., 2025; Vankov et al., 2025) have analyzed the convergence of variants of normalized gradient descent for general non-convex functions (that do not necessarily satisfy Assumption 3) and convex functions. However, unlike Thm. 1, these papers do not derive dimension-free guarantees for general normalized steepest descent.

  Furthermore, under this assumption, Li et al. (2023b); Wang et al. (2024a;b) analyze the scalar or norm version of Adam and derive a $O(1/\epsilon^2)$ stationary-point convergence for general non-convex functions. In Thm. 5, we specialize our proof technique to the $(L_0, L_1)$ assumption, and derive faster convergence rates.

**Other work:** Apart from these main bodies of work, there are papers that focus on specific examples of $(H_0, H_1)$ functions and analyze the convergence of specific normalized steepest descent methods and derive similar rates as in our Theorem 1. For example, Mei et al. (2021) use normalized gradient descent for the softmax policy gradient objective, Taheri & Thrampoulidis (2023) use it for 2 layer neural networks and Axiotis & Sviridenko (2023) use greedy coordinate descent (corresponding to $p = 1, q = \infty$ in our Theorem 1) and analyze its convergence on logistic regression.

## B. Examples

**Proposition 1.** *Consider $n$ points where $x_i \in \mathbb{R}^d$ are the features and $y_i \in \{-1, 1\}$ are the corresponding labels. Binary classification with an exponential loss,*

$$f(\theta) := \frac{1}{n} \sum_{i=1}^{n} \exp(-y_i \langle x_i, \theta \rangle), \tag{4}$$

*satisfies Assn. 1 and Assn. 2 with $H_0 = 0$ and $H_1 = \max_i \|x_i\|_q^2$. Furthermore, if the data is separable with a normalized margin $\gamma_p := \max_\theta \min_i \frac{y_i \langle x_i, \theta \rangle}{\|\theta\|_p} > 0$, then, $f(\theta)$ satisfies Assn. 3 with $\tau = 1$, $\mu = \gamma_p$ and $f^* = 0$.*

*Proof.* Clearly, $f_i(\theta) \geq 0$ and hence $f(\theta) \geq 0$ for all $\theta$. Calculating the gradient and hessian for $f_i(\theta) := \exp(-y_i \langle x_i, \theta \rangle)$,

$$\nabla f_i(\theta) = -\exp(-y_i \langle x_i, \theta \rangle) y_i \, x_i$$

$$\nabla^2 f_i(\theta) = \exp(-y_i \langle x_i, \theta \rangle) \, y_i^2 \, x_i x_i^T = \exp(-y_i \langle x_i, \theta \rangle) \, x_i x_i^T \qquad (y_i^2 = 1)$$

$$\implies \left\| \nabla^2 f_i(\theta) \right\|_{p \to q} \leq \exp(-y_i \langle x_i, \theta \rangle) \left\| x_i x_i^T \right\|_{p \to q} = f_i(\theta) \left\| x_i \right\|_q^2 \quad \text{(Since for a rank one matrix } \|uu^T\|_{p \to q} = \|u\|_q^2\text{)}$$

$$\left\| \nabla^2 f(\theta) \right\|_{p \to q} \leq \frac{1}{n} \sum_{i=1}^{n} \left\| \nabla^2 f_i(\theta) \right\|_{p \to q} \leq f(\theta) \max_i \|x_i\|_q^2 \qquad \text{(Using triangle inequality and above relation)}$$

Verifying Assn. 3 for separable data,

$$\|\nabla f(\theta)\|_q = \sup_{\|u\|_p \leq 1} \langle u, \nabla f(\theta) \rangle$$

$$= \sup_{\|u\|_p \leq 1} \left[ \frac{1}{n} \left\langle -\sum_{i=1}^{n} \exp(-y_i \langle x_i, \theta \rangle) y_i \, x_i, u \right\rangle \right]$$

$$\geq \left[ \frac{1}{n} \left\langle -\sum_{i=1}^{n} \exp(-y_i \langle x_i, \theta \rangle) y_i \, x_i, u \right\rangle \right] \qquad \text{(for all } u \text{ s.t. } \|u\|_p \leq 1)$$

$$\implies \|\nabla f(\theta)\|_q \geq \left[ \frac{1}{n} \left\langle \sum_{i=1}^{n} \exp(-y_i \langle x_i, \theta \rangle) y_i \, x_i, \frac{\theta^*}{\|\theta^*\|_p} \right\rangle \right] \qquad \text{(Setting } u = -\frac{\theta^*}{\|\theta^*\|_p})$$

$$\geq \min_j \frac{y_j \langle x_j, \theta^* \rangle}{\|\theta^*\|_p} \left[ \frac{1}{n} \sum_{i=1}^{n} \exp\left( -y_i \langle x_i, \theta \rangle \right) \right]$$

$$\implies \|\nabla f(\theta)\|_q \geq \gamma_p \, f(\theta) \qquad \text{(By definition of } \gamma_p \text{ and } f(\theta))$$

$$\square$$

**Proposition 2.** *Consider $n$ points where $x_i \in \mathbb{R}^D$ are the features and $y_i \in \{-1, 1\}$ are the corresponding labels. Logistic regression with the objective,*

$$f(\theta) := \frac{1}{n} \sum_{i=1}^{n} \ln(1 + \exp(-y_i \langle x_i, \theta \rangle)) \qquad (5)$$

*satisfies Assn. 1 and Assn. 2 with $H_0 = 0$ and $H_1 = \max_i \|x_i\|_q^2$. Furthermore, if the data is separable with a normalized margin $\gamma_p := \max_\theta \min_i \frac{y_i \langle x_i, \theta \rangle}{\|\theta\|_p} > 0$ then,*

• *For $f(\theta) \leq \frac{\ln(2)}{n}$, $f$ satisfies Assn. 3 with $\tau = 1$, $\mu = \frac{\gamma_p}{2}$ and $f^* = 0$*

• *Else, if $f(\theta) > \frac{\ln(2)}{n}$, then, $\|\nabla f(\theta)\|_q \geq \frac{\gamma_p}{3n}$.*

*Proof.* Clearly, $f_i(\theta) \geq 0$ and hence $f(\theta) \geq 0$ for all $\theta$. Calculating the gradient and hessian for $f_i(\theta) := \ln(1 + \exp(-y_i \langle x_i, \theta \rangle))$,

$$\nabla f_i(\theta) = \frac{-\exp(-y_i \langle x_i, \theta \rangle)}{1 + \exp(-y_i \langle x_i, \theta \rangle)} y_i \, x_i \quad ; \quad \nabla^2 f_i(\theta) = \frac{1}{1 + \exp(-y_i \langle x_i, \theta \rangle)} \frac{\exp(-y_i \langle x_i, \theta \rangle)}{1 + \exp(-y_i \langle x_i, \theta \rangle)} y_i^2 \, x_i \, x_i^T$$

Bounding the Hessian,

$$\left\| \nabla^2 f_i(\theta) \right\|_{p \to q} \leq \left\| \frac{1}{1 + \exp(-y_i \langle x_i, \theta \rangle)} \frac{\exp(-y_i \langle x_i, \theta \rangle)}{1 + \exp(-y_i \langle x_i, \theta \rangle)} y_i^2 \, x_i \, x_i^T \right\|_{p \to q}$$

$$= \frac{1}{1 + \exp(-y_i \langle x_i, \theta \rangle)} \frac{\exp(-y_i \langle x_i, \theta \rangle)}{1 + \exp(-y_i \langle x_i, \theta \rangle)} \left\| x_i \, x_i^T \right\|_{p \to q} \qquad \text{(Since } y_i^2 = 1)$$

$$\leq \frac{1}{1 + \exp(-y_i \langle x_i, \theta \rangle)} \frac{\exp(-y_i \langle x_i, \theta \rangle)}{1 + \exp(-y_i \langle x_i, \theta \rangle)} \|x_i\|_q^2$$

$$\text{(Since for a rank one matrix } \left\| uu^T \right\|_{p \to q} = \|u\|_q^2)$$

$$\leq \frac{\exp(-y_i \langle x_i, \theta \rangle)}{1 + \exp(-y_i \langle x_i, \theta \rangle)} \|x_i\|_q^2 \qquad \text{(For all } x, \frac{1}{1 + e^x} \leq 1)$$

$$\leq \ln(1 + \exp(-y_i \langle x_i, \theta \rangle)) \|x_i\|_q^2 \qquad \text{(For all } x \geq 0, \frac{x}{1+x} \leq \ln(1 + x))$$

$$\implies \left\| \nabla^2 f_i(\theta) \right\|_{p \to q} \leq f_i(\theta) \|x_i\|_q^2$$

$$\implies \left\|\nabla^2 f(\theta)\right\|_{p\to q} \leq \frac{1}{n}\sum_{i=1}^{n}\left\|\nabla^2 f_i(\theta)\right\|_{p\to q} \leq f(\theta)\max_i\|x_i\|_q^2 \qquad \text{(Using triangle inequality and above relation)}$$

Verifying Assn. 3 for separable data,

$$\|\nabla f(\theta)\|_q = \sup_{\|u\|_p\leq 1}\langle u, \nabla f(\theta)\rangle$$

$$= \sup_{\|u\|_p\leq 1}\left[\frac{1}{n}\left\langle -\sum_{i=1}^{n}\frac{1}{1+\exp(y_i\langle x_i,\theta\rangle)}y_i\,x_i, u\right\rangle\right]$$

$$\geq \left[\frac{1}{n}\left\langle -\sum_{i=1}^{n}\frac{1}{1+\exp(y_i\langle x_i,\theta\rangle)}\,y_i\,x_i, u\right\rangle\right] \qquad \text{(for all } u \text{ s.t. } \|u\|_p\leq 1)$$

$$\implies \|\nabla f(\theta)\|_q \geq \left[\frac{1}{n}\left\langle \sum_{i=1}^{n}\frac{1}{1+\exp(y_i\langle x_i,\theta\rangle)}y_i\,x_i, \frac{\theta^*}{\|\theta^*\|_p}\right\rangle\right] \qquad \text{(Setting } u = -\frac{\theta^*}{\|\theta^*\|_p})$$

$$\geq \min_j \frac{y_j\langle x_j,\theta^*\rangle}{\|\theta^*\|_p}\left[\frac{1}{n}\sum_{i=1}^{n}\frac{1}{1+\exp(y_i\langle x_i,\theta\rangle)}\right]$$

$$\implies \|\nabla f(\theta)\|_q \geq \gamma_p\left[\frac{1}{n}\sum_{i=1}^{n}\frac{1}{1+\exp(y_i\langle x_i,\theta\rangle)}\right] \qquad \text{(By definition of } \gamma_p)$$

**Case (1)**: If $f(\theta) \leq \frac{\ln(2)}{n}$, then,

$$\implies \sum_i \ln(1+\exp(-y_i\langle x_i,\theta\rangle)) \leq \ln(2) \implies \ln(1+\exp(-y_i\langle x_i,\theta\rangle)) \leq \ln(2) \qquad \text{(Since } f_i(\theta) > 0)$$

$$\implies y_i\langle x_i,\theta\rangle \geq 0$$

Hence, all examples are classified with a positive margin. In this case, we use the following inequality,

$$\frac{1}{1+\exp(z)} = \frac{\exp(-z)}{1+\exp(-z)} \geq \frac{\exp(-z)}{2} \qquad \text{(For } z \geq 0)$$

$$\geq \frac{\ln(1+\exp(-z))}{2} \qquad \text{(For all } z,\ \ln(1+z) \leq z)$$

Since $y_i\langle x_i,\theta\rangle > 0$ for all $i \leq [n]$,

$$\frac{1}{1+\exp(y_i\langle x_i,\theta\rangle)} \geq \frac{\ln(1+\exp(-y_i\langle x_i,\theta\rangle))}{2}$$

$$\implies \|\nabla f(\theta)\|_q \geq \frac{\gamma_p}{2}\left[\frac{1}{n}\sum_{i=1}^{n}\ln(1+\exp(-y_i\langle x_i,\theta\rangle))\right] = \frac{\gamma_p}{2}f(\theta)$$

**Case (2)**: On the other hand, if $f(\theta) > \frac{\ln(2)}{n}$ and $u_i := \exp(-y_i\langle x_i\theta\rangle) > 0$, then,

$$\sum_{i=1}^{n}\ln(1+\exp(-y_i\langle x_i,\theta\rangle)) > \ln(2) \implies \sum_{i=1}^{n}u_i > \ln(2) \qquad \text{(Since } \ln(1+u) < u \text{ for all } u > 0)$$

$$\implies \frac{1}{1+\exp(y_i\langle x_i,\theta\rangle)} = \frac{u_i}{1+u_i} \geq \frac{u_i}{1+\sum_{j=1}^{n}u_j} \qquad \text{(Since } u_j > 0)$$

$$\implies \sum_{i=1}^{n}\frac{1}{1+\exp(y_i\langle x_i,\theta\rangle)} \geq \frac{\sum_{i=1}^{n}u_i}{1+\sum_{j=1}^{n}u_j} \geq \frac{\ln(2)}{1+\ln(2)} \qquad \text{(Since } \frac{u}{1+u} \text{ is increasing for } u \geq 0)$$

Using the above lower-bound on the gradient norm,

$$\|\nabla f(\theta)\|_q \geq \gamma_p\left[\frac{1}{n}\sum_{i=1}^{n}\frac{1}{1+\exp(y_i\langle x_i,\theta\rangle)}\right] \geq \frac{\gamma_p}{n}\frac{\ln(2)}{1+\ln(2)} \geq \frac{\gamma_p}{3n}$$

$\square$

**Proposition 3.** *Given a multi-armed bandit problem with $K$ arms and known deterministic rewards $r \in [0,1]^K$, consider softmax policies $\pi_\theta \in \Delta_K$ parameterized by $\theta \in \mathbb{R}^K$ s.t. $\pi_\theta(a) = \exp(\theta(a))/\sum_{a'} \exp(\theta(a'))$. The softmax policy gradient objective is given by*

$$f(\theta) := r(a^*) - \langle \pi_\theta, r \rangle, \tag{6}$$

*where $a^* := \arg\max_{a \in [K]} r(a)$ is the optimal arm. $f(\theta)$ satisfies Assn. 1 and Assn. 2 with (i) $H_0 = 0$, $H_1 = 24$ for $p = q = 2$ and (ii) $H_0 = 0$, $H_1 = 6$ for $p = \infty, q = 1$ and $p = 1, q = \infty$. Furthermore, $f(\theta)$ satisfies Assn. 3 for all $q \geq 1$ with $\tau = 1$, $\mu(\theta) = \pi_\theta(a^*)$ and $f^* = 0$.*

*Proof.* Calculating the gradient,

$$\nabla f(\theta) = -\nabla[\langle \pi_\theta, r \rangle] = -G_\theta\, r \quad \text{where} \quad G_\theta \in \mathbb{R}^{K \times K} = \mathrm{diag}(\pi_\theta) - \pi_\theta\,[\pi_\theta]^T \quad \text{and} \quad G_\theta[i,j] = \frac{\partial \pi_\theta(i)}{\partial \theta_j}$$

$$\implies \nabla f(\theta) = -\mathrm{diag}(\pi_\theta)\, r + \langle \pi_\theta, r \rangle\, \pi_\theta \implies [\nabla f(\theta)]_j = -\pi_\theta(j)\,[r_j - \langle \pi_\theta, r \rangle] \tag{21}$$

Calculating the Hessian,

$$\frac{\partial^2 f}{\partial \theta_k\, \partial \theta_j} = \frac{\partial [\nabla f(\theta)]_j}{\partial \theta_k} = -\frac{\partial}{\partial \theta_k}[\pi_\theta(j)\,(r_j - \langle \pi_\theta, r \rangle)]$$

$$= -[G_{j,k}\,[r_j - \langle \pi_\theta, r \rangle] - \pi_\theta(j)\,(\pi_\theta(k)\,[r_k - \langle \pi_\theta, r \rangle])]$$

$$\implies \frac{\partial^2 f}{\partial \theta_k\, \partial \theta_j} = -G_{j,k}\,[r_j - \langle \pi_\theta, r \rangle] + \pi_\theta(j)\,\pi_\theta(k)\,[r_k - \langle \pi_\theta, r \rangle]$$

$$\implies \nabla^2 f(\theta) = -\underbrace{\mathrm{diag}(r - \langle \pi_\theta, r \rangle\, \mathbf{1})}_{K \times K}\, \underbrace{G_\theta}_{K \times K} + \underbrace{\pi_\theta}_{K \times 1}\, \underbrace{[G_\theta\, r]^T}_{1 \times K}$$

Define $w := G_\theta r = \mathrm{diag}(\pi_\theta)\,(r - \langle \pi_\theta, r \rangle \mathbf{1})$, and simplifying the first term,

$$\mathrm{diag}(r - \langle \pi_\theta, r \rangle\, \mathbf{1})\, G_\theta = \mathrm{diag}(r - \langle \pi_\theta, r \rangle\, \mathbf{1})\,\big(\mathrm{diag}(\pi_\theta) - \pi_\theta\,[\pi_\theta]^T\big)$$

$$= \mathrm{diag}(\pi_\theta \circ (r - \langle \pi_\theta, r \rangle)) - [\pi_\theta \circ (r - \langle \pi_\theta, r \rangle)][\pi_\theta]^T$$

$$= \mathrm{diag}(w) - w[\pi_\theta]^T$$

Combining the above relations,

$$\nabla^2 f(\theta) = -\mathrm{diag}(w) + w[\pi_\theta]^T + \pi_\theta[w]^T \tag{22}$$

**Case 1**: Consider $p = q = 2$. In this case,

$$\left\|\nabla^2 f(\theta)\right\|_{2\to2} \leq \|\mathrm{diag}(w)\|_{2\to2} + \left\|w[\pi_\theta]^T\right\|_{2\to2} + \left\|\pi_\theta[w]^T\right\|_{2\to2} \qquad \text{(Using triangle inequality)}$$

$$\leq \|w\|_2 + 2\,\|w\|_2\,\|\pi_\theta\|_2 \leq 3\,\|w\|_2 \qquad \text{(Since } \|\pi_\theta\|_2 \leq \|\pi_\theta\|_1 = 1)$$

$$\implies \left\|\nabla^2 f(\theta)\right\|_{2\to2} \leq 3\,\|w\|_2 = 3\,\|G_\theta r\| = 3\,\|\nabla f(\theta)\|_2 \qquad \text{(Using Eq. (21))}$$

Using (Vaswani & Harikandeh, 2025, Proposition 3), if $\left\|\nabla^2 f(\theta)\right\|_2 \leq 3\,\|\nabla f(\theta)\|_2$ and $f^* = 0$,

$$\|\nabla f(\theta)\|_2 \leq 8\,f(\theta)$$

Combining the above equations,

$$\left\|\nabla^2 f(\theta)\right\|_{2\to2} \leq 24\,f(\theta)$$

**Case 2**: Consider $p = \infty, q = 1$. In this case,

$$\left\|\nabla^2 f(\theta)\right\|_{\infty\to1} \leq \|\mathrm{diag}(w)\|_{\infty\to1} + \left\|w[\pi_\theta]^T\right\|_{\infty\to1} + \left\|\pi_\theta[w]^T\right\|_{\infty\to1} \qquad \text{(Using triangle inequality)}$$

$$\leq \max_{\|v\|_\infty \leq 1} \|\operatorname{diag}(w)\, v\| + \|w\|_1 \max_{\|v\|_\infty \leq 1} \langle \pi_\theta, v \rangle + \max_{\|v\|_\infty \leq 1} \langle w, v \rangle \sum_i \pi_\theta(i)$$

$$\leq 3\, \|w\|_1 = 3\, \|G_\theta\, r\|_1 \qquad\qquad \text{(Since } \|\pi_\theta\|_1 = \sum_i \pi_\theta(i) = 1\text{)}$$

$$\implies \left\|\nabla^2 f(\theta)\right\|_{\infty \to 1} \leq 3\, \|G_\theta\, r\|_1 \tag{23}$$

Calculating $\|G_\theta\, r\|_1$,

$$\|G_\theta\, r\|_1 = \sum_a \pi_\theta(a)\, |r(a) - \langle \pi_\theta, r \rangle| = \sum_a \pi_\theta(a)\, |r(a) - r(a^*) + r(a^*) - \langle \pi_\theta, r \rangle|$$

$$= \sum_a \pi_\theta(a)\, |r(a) - r(a^*) + f(\theta)| \leq f(\theta) + \sum_a \pi_\theta(a)|r(a) - r(a^*)|$$

$$\text{(Using triangle inequality and definition of } f(\theta)\text{)}$$

$$= f(\theta) + \sum_a \pi_\theta(a)[r(a^*) - r(a)] \qquad\qquad \text{(Since } r(a^*) > r(a) \text{ by definition of the optimal arm } a^*\text{)}$$

$$= f(\theta) + (r(a^*) - \langle \pi_\theta, r \rangle \implies \|G_\theta\, r\|_1 \qquad\qquad\qquad \leq 2\, f(\theta)$$

Combining the above inequality with Eq. (23),

$$\left\|\nabla^2 f(\theta)\right\|_{\infty \to 1} \leq 6\, f(\theta)$$

**Case 3:** Consider $p = 1, q = \infty$. Since $\|\cdot\|_{1,\infty} \leq \|\infty, 1\|$, using the result for case 2 immediately gives us:

$$\left\|\nabla^2 f(\theta)\right\|_{1 \to \infty} \leq 6\, f(\theta)$$

Verifying Assn. 3, for any $q \geq 1$,

$$\|\nabla f(\theta)\|_q = \|G_\theta\, r\|_q \geq |\pi_\theta(a^*)\, [r(a^*) - \langle \pi_\theta, r \rangle]| = \pi_\theta(a^*) f(\theta)$$

$\square$

**Proposition 4.** *Consider $n$ points where $x_i \in \mathbb{R}^D$ are the features and $y_i \in \{-1, 1\}$ are the corresponding labels, and a neural network,*

$$\Phi(\theta, x) := \sum_{j=1}^m a_j\, \sigma(\langle \theta_j, x \rangle), \tag{7}$$

*where $a_j$ are fixed, $m$ is the width of the layer and $\sigma$ is the activation function. Consider the case when $\sigma$ is smooth s.t. for all $t$, $|\sigma''(t)| \leq M$ and has bounded derivatives i.e. there exists positive constants $\alpha_1, \alpha_2$ such that $\alpha_1 \leq |\sigma'(t)| \leq \alpha_2$. Consider the loss*

$$f(\theta) := \frac{1}{n} \sum_{i=1}^n g\left(y_i\, \Phi(\theta, x_i)\right), \tag{8}$$

*where, $g : \mathbb{R} \to \mathbb{R}$ is differentiable everywhere and for all $s$, $g(s) \geq 0$, $g'(s) \leq 0$, $\frac{|g'(s)|}{g(s)} \in [c_1, c_2]$ and $|g''(s)| \leq c_2'\, g(s)$. $f$ satisfies Assn. 1 and Assn. 2 with $H_0 = 0$ and $H_1 = \left[c_2\, M\, \|a\|_1 + c_2'\, \alpha_2^2\, \|a\|_q^2\right] \max_i \|x_i\|_q^2$. Furthermore, if the data is linearly separable with a normalized margin $\gamma_p := \max_\theta \min_i \frac{y_i\, \langle x_i, \theta \rangle}{\|\theta\|_p} > 0$ then, $f(\theta)$ satisfies Assn. 3 with $\tau = 1$, $\mu = \frac{c_1\, \alpha_1\, \gamma\, \|a\|_2^2}{\|a\|_p}$ and $f^* = 0$.*

This proposition is a generalization of Lemmas 3 and 5 in Taheri & Thrampoulidis (2023), which is specific to the exponential loss, with $p = q = 2$.

*Proof.* Clearly $f$ satisfies Assn. 1 by construction. Moreover, for $\theta = [\theta_1, \theta_2, \ldots, \theta_m] \in \mathbb{R}^{mD}$ where for each $j \in [m]$, $\theta_j \in \mathbb{R}^D$,

$$f(\theta) = \frac{1}{n} \sum_{i=1}^{n} g(y_i \, \Phi(\theta, x_i)) \implies \nabla f(\theta) = \frac{1}{n} \sum_{i=1}^{n} g'(y_i \Phi(\theta, x_i)) \, y_i \nabla \Phi(\theta, x_i)$$

$$\Phi(\theta, x) = \sum_{j=1}^{m} a_j \sigma(\langle \theta_j, x \rangle) \implies \nabla \Phi(\theta, x) = [a_1 \sigma'(\langle \theta_1, x \rangle) x, a_2 \sigma'(\langle \theta_2, x \rangle) x, \ldots, a_m \sigma'(\langle \theta_m, x \rangle) x]$$

$$\implies \|\nabla \Phi(\theta, x)\|_q \leq \alpha_2 \|a\|_q \|x\|_q \tag{24}$$

$\nabla^2 \Phi(\theta, x)$ is a $\mathbb{R}^{mD \times mD}$ block-diagonal matrix where block $j$ is an $D \times D$ matrix equal to $a_j \, \sigma''(\langle \theta_j, x \rangle) x x^T$. Hence,

$$\left\| \nabla^2 \Phi(\theta, x) \right\|_{p \to q} \leq \sum_{j=1}^{m} \left\| a_j \, \sigma''(\langle \theta_j, x \rangle) x x^T \right\|_{p \to q} \leq \sum_{j=1}^{m} |a_j| \, M \, \|x\|_q^2 = M \, \|a\|_1 \, \|x\|_q^2 \tag{25}$$

Calculating the Hessian of $f$,

$$\nabla^2 f(\theta) = \frac{1}{n} \sum_{i=1}^{n} \left[ g'(y_i \, \Phi(\theta, x_i)) \, \nabla^2 \Phi(\theta, x_i) + g''(y_i \, \Phi(\theta, x_i)) \, [\nabla \Phi(\theta, x_i)] \, [\nabla \Phi(\theta, x_i)]^T \right]$$

$$\implies \left\| \nabla^2 f(\theta) \right\|_{p \to q} \leq \frac{1}{n} \sum_{i=1}^{n} \left[ |g'(y_i \, \Phi(\theta, x_i))| \, \left\| \nabla^2 \Phi(\theta, x_i) \right\|_{p \to q} + |g''(y_i \, \Phi(\theta, x_i))| \, \left\| [\nabla \Phi(\theta, x_i)] \, [\nabla \Phi(\theta, x_i)]^T \right\|_{p \to q} \right]$$

$$\text{(By triangle inequality)}$$

$$\leq \frac{1}{n} \sum_{i=1}^{n} g(y_i \, \Phi(\theta, x_i)) \left[ c_2 \left\| \nabla^2 \Phi(\theta, x_i) \right\|_{p \to q} + c_2' \left\| [\nabla \Phi(\theta, x_i)] \, [\nabla \Phi(\theta, x_i)]^T \right\|_{p \to q} \right]$$

$$\text{(Since } |g'(s)| \leq c_2 \, g(s) \text{ and } |g''(s)| \leq c_2' \, g(s) \text{ for all } s\text{)}$$

$$\leq \frac{1}{n} \sum_{i=1}^{n} g(y_i \, \Phi(\theta, x_i)) \left[ c_2 \, L \, \|a\|_1 \, \|x_i\|_q^2 + c_2' \, \alpha_2^2 \, \|a\|_q^2 \, \|x_i\|_q^2 \right]$$

$$= \left[ c_2 \, M \, \|a\|_1 + c_2' \, \alpha_2^2 \, \|a\|_q^2 \right] \, \max_i \|x_i\|_q^2 \, f(\theta)$$

Hence, $f$ satisfies Assn. 2 with $H_0 = 0$ and $H_1 = \left[ c_2 \, M \, \|a\|_1 + c_2' \, \alpha_2^2 \, \|a\|_q^2 \right] \, \max_i \|x_i\|_q^2$.

Next, using the construction $v_j = a_j \, w^*$ and $v = (v_1, v_2, \ldots) = [a_1 \, w^*, a_2 \, w^*, \ldots, a_m \, w^*]$, where $w^*$ is the max-margin separator that satisfies $\frac{y_i \langle x_i, w^* \rangle}{\|w^*\|_p} \geq \gamma$, we get that,

$$\|\nabla f(\theta)\|_q = \sup_u \frac{\langle -\nabla f(\theta), u \rangle}{\|u\|_p} \qquad \text{(By definition of dual norm)}$$

$$= \sup_u -\frac{1}{\|u\|_p} \frac{1}{n} \sum_{i=1}^{n} g'(y_i \Phi(\theta, x_i)) \, y_i \, \langle \nabla \Phi(\theta, x_i), u \rangle$$

$$\text{(Substituting } \nabla f(\theta) = \frac{1}{n} \sum_i g'(y_i \Phi(\theta, x_i)) \, y_i \nabla \Phi(\theta, x_i)\text{)}$$

$$\geq -\frac{1}{\|v\|_p} \frac{1}{n} \sum_{i=1}^{n} g'(y_i \Phi(\theta, x_i)) \, y_i \, \langle \nabla \Phi(\theta, x_i), v \rangle \qquad \text{(Substituting } u = v\text{)}$$

$$= -\frac{1}{n \|v\|_p} \sum_{i=1}^{n} g'(y_i \Phi(\theta, x_i)) \, y_i \sum_{j=1}^{m} a_j \, \sigma'(\langle \theta_j, x_i \rangle) \, \langle x_i, v_j \rangle \quad \text{(Substituting } \nabla \Phi(\theta, x) = \sum_j a_j \sigma'(\langle \theta_j, x \rangle) x\text{)}$$

$$\geq \frac{1}{n \, \|v\|_p} \sum_{i=1}^{n} |g'(y_i \Phi(\theta, x_i))| \sum_{j=1}^{m} a_j^2 \, \sigma'(\langle \theta_j, x_i \rangle) \, \langle y_i x_i, w^* \rangle \qquad \text{(Since } v_i = a_i \, w^* \text{ and } g'(s) \leq 0 \text{ for all } s\text{)}$$

$$= \frac{1}{n \|a\|_p} \sum_{i=1}^{n} |g'(y_i \Phi(\theta, x_i))| \sum_{j=1}^{m} a_j^2 \, \sigma'(\langle \theta_j, x_i \rangle) \frac{\langle y_i x_i, w^* \rangle}{\|w^*\|_p}$$

$$\geq \frac{\alpha_1}{n\|a\|_p} \sum_{i=1}^{n} |g'(y_i \Phi(\theta, x_i))| \underbrace{\sum_{j=1}^{m} a_j^2}_{\|a\|_2^2} \underbrace{\frac{\langle y_i x_i, w^* \rangle}{\|w^*\|_p}}_{\geq \gamma} \qquad \text{(Since } y_i \langle x_i, w^* \rangle > 0 \text{ and } \sigma'(t) \geq \alpha_1 \text{ for all } t\text{)}$$

$$\geq \frac{\alpha_1 \gamma \|a\|_2^2}{\|a\|_p} \frac{1}{n} \sum_{i=1}^{n} |g'(y_i \Phi(\theta, x_i))| = \frac{\alpha_1 \gamma \|a\|_2^2}{\|a\|_p} \frac{1}{n} \sum_{i=1}^{n} -g'(y_i \Phi(\theta, x_i)) \qquad \text{(Since } g'(s) \leq 0 \text{ for all } s\text{)}$$

$$\geq \frac{\alpha_1 \gamma \|a\|_2^2 c_1}{\|a\|_p} \underbrace{\frac{1}{n} \sum_{i=1}^{n} g(y_i \Phi(\theta, x_i))}_{f(\theta)} \qquad \text{(Since } -g'(s) \geq c_1\, g(s) \text{ for all } s\text{)}$$

Hence, $f$ satisfies Assn. 3 with $\mu = \frac{\alpha_1 \gamma \|a\|_2^2 c_1}{\|a\|_p}$ $\qquad\qquad\qquad\qquad\qquad\qquad\qquad\qquad\qquad\qquad\qquad\quad\square$

**Proposition 5.** *Consider $n$ points where $x^{(i)} \in \mathbb{R}^D$ are the features and $y^{(i)} \in \{0,1\}^K$ are the corresponding one-hot label vectors for $K$ classes. For each $i \in [n]$, $c_i \in [K]$ is the index of the true label such that $y^{(i)}(c_i) = 1$ and for all $j \neq c_i$, $y^{(i)}(j) = 0$. The loss for multi-class classification with the cross-entropy objective (multiclass logistic regression) is given as:*

$$f(\theta) = \frac{1}{n} \sum_{i=1}^{n} KL(y^{(i)} \| \pi_\theta^{(i)}), \text{ where } \forall i \in [n], \pi_\theta^{(i)} \in \Delta_K \text{ s.t. } \forall c \in [K], \pi_\theta^{(i)}(c) = \frac{\exp(\langle x^{(i)}, \theta_c \rangle)}{\sum_{k=1}^{K} \exp(\langle x^{(i)}, \theta_k \rangle)},$$

*where $\theta_c \in \mathbb{R}^D$ for $c \in [K]$ and $\theta = [\theta_1, \theta_2, \ldots, \theta_K]$. Multi-class logistic regression satisfies Assn. 1 and Assn. 2 with $H_0 = 0$ and $H_1 = 4 \max_i \|x_i\|_q^2$.*

*Furthermore, if the data is separable with a normalized margin $\gamma_p := \max_\theta \min_{i \in [n]} \langle x_i, \theta_{c_i} \rangle - \min_{\substack{k \in [K] \\ k \neq c_i}} \langle x_i, \theta_k \rangle > 0$, then,*

- *For $f(\theta) \leq \frac{\ln(2)}{n}$, $f$ satisfies Assn. 3 with $\tau = 1$, $\mu = \frac{\gamma_p}{K}$ and $f^* = 0$*

- *Else, if $f(\theta) > \frac{\ln(2)}{n}$ then $\|\nabla f(\theta)\|_q \geq \frac{\gamma_p}{2n}$.*

*Proof.* For the first part of the proof, note that,

$$f_i(\theta) = KL(y^{(i)} \| \pi_\theta^{(i)}) \text{ where } \pi_\theta^{(i)} \in \Delta_K \text{ s.t. } \pi_\theta^{(i)}(c) = \frac{\exp(\langle x^{(i)}, \theta_c \rangle)}{\sum_{k=1}^{K} \exp(\langle x^{(i)}, \theta_k \rangle)},$$

where $\theta_c \in \mathbb{R}^D$ for $c \in [K]$ and $\theta = [\theta_1, \theta_2, \ldots, \theta_K]$. Since $y^{(i)}$ is a one-hot vector, let $c_i$ be the index corresponding to the non-zero entry. Hence, $y_{c_i}^{(i)} = 1$ and for all $j \neq c_i$, $y_j^{(i)} = 0$, and hence,

$$f_i(\theta) = -\ln\left(\pi_\theta^{(i)}(c_i)\right)$$

The Hessian can be written as a Kronecker product of a $K \times K$ matrix which corresponds to the Jacobian of the softmax function, and a $d \times d$ rank-one matrix formed using the features. Specifically,

$$\nabla^2 f_i(\theta) = \underbrace{G^{(i)}(\theta)}_{K \times K} \otimes \underbrace{x^{(i)} [x^{(i)}]^T}_{d \times d} \text{ where, } G^{(i)}(\theta) := \text{diag}(\pi_\theta^i) - \pi_\theta^i [\pi_\theta^{(i)}]^T \qquad (26)$$

Using Lemma. 1 for the fixed $\theta$, and using $G^{(i)}$ as a shorthand for $G^{(i)}(\theta)$, for $q \in [1, \infty)$,

$$\left\|\nabla^2 f_i(\theta)\right\|_{p \to q} \leq \left(\sum_{k,l} |G_{k,l}^{(i)}|^q\right)^{1/q} \left(\sum_{k,l} \left(x_k^{(i)} x_l^{(i)}\right)^q\right)^{1/q}$$

$$= \left( \sum_{k,l} |G_{k,l}^{(i)}|^q \right)^{1/q} \left\| x^{(i)} \right\|_q^2$$

$$\leq \sum_{k,l} |G_{k,l}^{(i)}| \left\| x^{(i)} \right\|_q^2 \qquad \text{(Since } \|\cdot\|_q \leq \|\cdot\|_1 \text{ for all } q \geq 1\text{)}$$

On the other hand, using Lemma. 1 for $q = \infty$,

$$\left\| \nabla^2 f_i(\theta) \right\|_{1 \to \infty} \leq \max_{k,l} |G_{k,l}^{(i)}| \, (\max_k |x_k^{(i)}|)^2 \leq \sum_{k,l} |G_{k,l}^{(i)}| \left\| x^{(i)} \right\|_\infty^2 \qquad \text{(Since } \max_i v_i \leq \sum_i v_i\text{)}$$

Hence, in both cases,

$$\left\| \nabla^2 f_i(\theta) \right\|_{p \to q} \leq \sum_{k,l} |G_{k,l}^{(i)}| \left\| x^{(i)} \right\|_q^2$$

Bounding $\sum_{k,l} |G_{k,l}^{(i)}|$, if $z := \pi_\theta^{(i)}$, then,

$$\sum_{k,l} |G_{k,l}^{(i)}| = \sum_k z_k (1 - z_k) - \sum_{l \neq k} z_l z_k = 2 \sum_k z_k (1 - z_k) \qquad \text{(Since } \sum_k z_k = 1, \sum_{l \neq k} z_l = 1 - z_k\text{)}$$

$$= 2 \left( 1 - \|z\|_2^2 \right) \leq 2 \left( 1 - [z(c_i)]^2 \right) \qquad \text{(Since } \forall c \in [K], 1 - \|p\|^2 \leq 1 - [p(c)]^2\text{)}$$

$$\leq 4 \left( 1 - z(c_i) \right) \qquad \text{(Since for all } z, 1 - z^2 \leq 2(1 - z)\text{)}$$

$$\leq 4 \ln \left( \frac{1}{z(c_i)} \right) \qquad \text{(For all } z \in [0, 1], 1 - z \leq \ln(1/z)\text{)}$$

$$= 4 \, \text{KL}(y^{(i)} \| \pi_\theta^{(i)}) \qquad \text{(Using that } y^{(i)} \text{ is a one-hot vector and } z = \pi_\theta^{(i)}\text{)}$$

$$= 4 \, f_i(\theta)$$

Combining the above inequalities,

$$\left\| \nabla^2 f_i(\theta) \right\|_{p \to q} \leq 4 \left\| x^{(i)} \right\|_q^2 f_i(\theta)$$

$$\implies \left\| \nabla^2 f(\theta) \right\|_{p \to q} \leq \frac{1}{n} \sum_{i=1}^n \left\| \nabla^2 f_i(\theta) \right\|_{p \to q} \leq 4 \max_i \|x_i\|_q^2 f(\theta) \qquad \text{(Using triangle inequality and above relation)}$$

This proves Assn. 2. Note that, for all $i \in [n]$ and $j \in [K]$, if $\theta^* = [\theta_1^*, \theta_2^*, \ldots, \theta_K^*]$ with $\|\theta^*\|_p \leq 1$ is the max-margin solution, then,

$$\frac{\partial f_i(\theta)}{\partial \theta_j} = \left( \pi_\theta^{(i)}(j) - \mathcal{I}\{j = c_i\} \right) x^{(i)}$$

$$\implies -\langle \theta^*, \nabla f_i(\theta) \rangle = \sum_{j=1}^K \left( \mathcal{I}\{j = c_i\} - \pi_\theta^{(i)}(j) \right) \langle x^{(i)}, \theta_j^* \rangle = \langle x^{(i)}, \theta_{c_i}^* \rangle - \sum_{j=1}^K \pi_\theta^{(i)}(j) \langle x^{(i)}, \theta_j^* \rangle$$

$$= \sum_{j=1}^K \pi_\theta^{(i)}(j) \left[ \langle x^{(i)}, \theta_{c_i}^* - \theta_j^* \rangle \right] = \sum_{j=1, j \neq c_i}^K \pi_\theta^{(i)}(j) \left[ \langle x^{(i)}, \theta_{c_i}^* - \theta_j^* \rangle \right]$$

$$\geq \gamma_p \sum_{j=1, j \neq c_i}^K \pi_\theta^{(i)}(j) \qquad \text{(By definition of the margin } \gamma_p\text{)}$$

$$\implies -\langle \theta^*, \nabla f_i(\theta) \rangle \geq \gamma_p \left( 1 - \pi_\theta^{(i)}(c_i) \right)$$

$$\implies -\langle \theta^*, \nabla f(\theta) \rangle \geq \frac{\gamma_p}{n} \sum_{i=1}^n \left( 1 - \pi_\theta^{(i)}(c_i) \right)$$

By definition of the dual norm and using the above relation,

$$\|\nabla f(\theta)\|_q = \max_{\|v\|_p \leq 1} \langle v, -\nabla f(\theta) \rangle \geq \frac{\gamma_p}{n} \sum_{i=1}^{n} \left(1 - \pi_\theta^{(i)}(c_i)\right)$$

We will now relate $\pi_\theta^{(i)}(c_i)$ to the loss.

**Case (i)**: If $f(\theta) \leq \frac{\ln(2)}{n}$, then for all $i \in [n]$,

$$f_i(\theta) \leq \ln(2) \implies \pi_\theta^{(i)}(c_i) \geq \frac{1}{2} \implies 1 - \pi_\theta^{(i)}(c_i) \geq \frac{-\ln(\pi_\theta^{(i)}(c_i))}{2} = \frac{f_i(\theta)}{2}$$

$$\text{(Since for all } u \in \left[\tfrac{1}{2}, 1\right], 1 - u \geq -\tfrac{\ln(u)}{2})$$

$$\implies \|\nabla f(\theta)\|_q \geq \frac{\gamma_p}{n} \sum_{i=1}^{n} \left(1 - \pi_\theta^{(i)}(c_i)\right) \geq \frac{\gamma_p}{2} f(\theta)$$

**Case (ii)**: If $f(\theta) > \frac{\ln(2)}{n}$,

$$\sum_{i=1}^{n} f_i(\theta) \geq \ln(2) \implies -\sum_{i=1}^{n} \ln(\pi_\theta^{(i)}(c_i)) \geq \ln(2) \implies \pi_\theta^{(1)}(c_1) \times \pi_\theta^{(2)}(c_2) \times \ldots \times \pi_\theta^{(n)}(c_n) \leq \frac{1}{2}$$

$$\implies 1 - \sum_{i=1}^{n}[1 - \pi_\theta^{(i)}(c_i)] \leq \frac{1}{2} \implies \sum_{i=1}^{n}[1 - \pi_\theta^{(i)}(c_i)] \geq \frac{1}{2}$$

$$\text{(Since for probabilities } p_i, \prod_{i=1}^{n} p_i \geq 1 - \sum_{i=1}^{n}[1 - p_i])$$

$$\implies \|\nabla f(\theta)\|_q \geq \frac{\gamma_p}{n} \sum_{i=1}^{n} \left(1 - \pi_\theta^{(i)}(c_i)\right) \geq \frac{\gamma_p}{2n}$$

$\square$

**Proposition 6.** *Consider $n$ points where $x_i \in \mathbb{R}^D$ are the features and $y_i \in [0, 1]$ are the corresponding labels. If $\pi_i(\theta) = \sigma(\langle x_i, \theta \rangle) := \frac{1}{1 + \exp(-\langle x_i, \theta \rangle)}$, the GLM objective,*

$$f(\theta) = \frac{1}{2n} \sum_{i=1}^{n} \left(\pi_i(\theta) - y_i\right)^2 , \tag{27}$$

*satisfies Assn. 1 and Assn. 2 with $H_0 = \frac{3 \max_i\{\|x_i\|_q^2\}}{16}$ and $H_1 = \frac{\max_i\{\|x_i\|_q^2\}}{4}$. Furthermore, assuming that for all $i \in [n]$, $\|x_i\|_2 \leq 1$, $y_i = \pi_i(\theta^*)$ such that $\|\theta^*\| \leq D < \infty$ and $\upsilon(\theta) := \min_{i \in [n]} \{\pi_i(\theta) \cdot (1 - \pi_i(\theta))\}$, then the GLM objective in Eq. (27) satisfies Assn. 3 with $\tau = \frac{1}{2}$ and $\mu(\theta) = 64 \left[\upsilon(\theta)\right]^2 \left[\min\{\upsilon(\theta), \upsilon(\theta^*)\}\right]^2$.*

*Proof.* Clearly, $f_i(\theta) \geq 0$ and hence $f(\theta) \geq 0$ for all $\theta$. $f(\theta)$ is a finite-sum objective. Calculating the gradient and hessian for $f_i(\theta) = \frac{1}{2}\left(\pi_i(\theta) - y_i\right)^2$,

$$\nabla f_i(\theta) = (\pi_i(\theta) - y_i) \frac{1}{1 + \exp(-\langle x_i, \theta \rangle)} \frac{\exp(-\langle x_i, \theta \rangle)}{1 + \exp(-\langle x_i, \theta \rangle)} x_i = (\pi_i(\theta) - y_i)\, \pi_i(\theta)\,(1 - \pi_i(\theta))\, x_i$$

$$\nabla^2 f_i(\theta) = [1 - 2\,\pi_i(\theta)]\,\pi_i(\theta)\,[1 - \pi_i(\theta)]\,[\pi_i(\theta) - y_i]\, x_i x_i^T + [\pi_i(\theta)]^2\,[1 - \pi_i(\theta)]^2\, x_i x_i^T$$

$$\implies \left\|\nabla^2 f_i(\theta)\right\|_{p \to q} = \underbrace{\left|[1 - 2\,\pi_i(\theta)]\,\pi_i(\theta)\,[1 - \pi_i(\theta)]\,[\pi_i(\theta) - y_i] + [\pi_i(\theta)]^2\,[1 - \pi_i(\theta)]^2\right|}_{:=(*)} \|x_i\|_q^2$$

$$\text{(Since for a rank one matrix } \left\|uu^T\right\|_{p \to q} = \|u\|_q^2)$$

To bound (*), define $a_i := \pi_i(\theta) - y_i$ and $b_i := \pi_i(\theta)\,[1 - \pi_i(\theta)]$,

$$(*) = \left|[1 - 2\,\pi_i(\theta)]\, a_i b_i + b_i^2\right| \leq \underbrace{|1 - 2\pi_i(\theta)|}_{\leq 1}\, \underbrace{|a_i|}_{}\, \underbrace{|b_i|}_{\leq \frac{1}{4}} + \underbrace{|b_i|^2}_{\leq \frac{1}{16}}$$

$$\leq \frac{|a_i|}{4} + \frac{1}{16} \leq \frac{a_i^2}{8} + \frac{1}{8} + \frac{1}{16} \qquad \text{(Since for all } x, |x| \leq \frac{x^2+1}{2})$$

$$= \frac{a_i^2}{8} + \frac{3}{16}$$

$$(*) \leq \frac{1}{4}\frac{1}{2}\left(\pi_i(\theta) - y_i\right)^2 + \frac{3}{16} = \frac{1}{4}f_i(\theta) + \frac{3}{16}$$

$$\implies \left\|\nabla^2 f_i(\theta)\right\|_{p\to q} \leq \frac{1}{4}\left\|x_i\right\|_q^2 f_i(\theta) + \frac{3}{16}\left\|x_i\right\|_q^2$$

$$\implies \left\|\nabla^2 f(\theta)\right\|_{p\to q} \leq \frac{1}{n}\sum_{i=1}^{n}\left\|\nabla^2 f_i(\theta)\right\|_{p\to q} \leq \frac{1}{4}\frac{\max_i\left\{\|x_i\|_q^2\right\}}{n}\sum_{i=1}^{n}f_i(\theta) + \frac{3\max_i\left\{\|x_i\|_q^2\right\}}{16}$$

$$\text{(Using triangle inequality and above relation)}$$

$$= \frac{\max_i\left\{\|x_i\|_q^2\right\}}{4}f(\theta) + \frac{3\max_i\left\{\|x_i\|_q^2\right\}}{16}$$

Assn. 3 follows from (Mei et al., 2021, Lemma 9). $\qquad\square$

### B.1. Helper Lemmas

**Lemma 1.** *For $A \in \mathbb{R}^{m\times n}$ and $B \in \mathbb{R}^{r\times s}$ and $p, q$ such that $\frac{1}{p} + \frac{1}{q} = 1$,*

- *If $q \in [1, \infty)$, $\|A\otimes B\|_{p\to q} \leq \left(\sum_{i,j}|A_{i,j}|^q\right)^{1/q}\left(\sum_{k,l}|B_{k,l}|^q\right)^{1/q}$*

- *Else if $q = \infty$, $\|A\otimes B\|_{p\to q} \leq \max_{i,j}|A_{ij}|\max_{k,l}|B_{k,l}|$*

*Proof.* For proving the first part, note that, for any matrix $M$ and vector $x$, for $q \in [1, \infty)$,

$$\|Mx\|_q^q = \sum_i\left|\sum_j M_{ij}x_j\right|^q = \sum_i|\langle M_{i,:}, x\rangle|^q \qquad \text{(By definition of }\|\cdot\|_q)$$

$$\leq \sum_i\|M_{i,:}\|_q^q\|x\|_p^q \qquad \text{(Using Holder's inequality)}$$

$$\implies \|Mx\|_q \leq \left(\sum_i\|M_{i,:}\|_q^q\|x\|_p^q\right)^{1/q} = \|x\|_p\left(\sum_i\|M_{i,:}\|_q^q\right)^{1/q} \qquad (28)$$

$$\implies \|M\|_{p\to q} = \sup_{x\neq 0}\frac{\|Mx\|_q}{\|x\|_p} \leq \left(\sum_i\|M_{i,:}\|_q^q\right)^{1/q} = \left(\sum_{i,j}|M_{i,j}|^q\right)^{1/q} \qquad (29)$$

By definition of the Kronecker product and the above inequality,

$$\|A\otimes B\|_{p\to q} = \left(\sum_{i,j}\sum_{k,l}|A_{i,j}|^q|B_{k,l}|^q\right)^{1/q} \leq \left(\sum_{i,j}|A_{i,j}|^q\right)^{1/q}\left(\sum_{k,l}|B_{k,l}|^q\right)^{1/q}$$

For the second part, note that, for any matrix $M$, if $q = \infty$, $p = 1$ and hence,

$$\|M\|_{1\to\infty} = \sup_{\|x\|_1\leq 1}\|Mx\|_\infty = \sup_{\|x\|_1\leq 1}\max_i\left|\sum_j M_{ij}x_j\right| \qquad \text{(By definition of }\|\cdot\|_\infty)$$

$$= \max_i\sup_{\|x\|_1\leq 1}\left|\sum_j M_{ij}x_j\right| = \max_i\sup_{\|x\|_1\leq 1}|\langle M_{i,:}, x\rangle| = \max_i\|M_{i,:}\|_\infty \qquad \text{(By definition of dual norm)}$$

$$\implies \|M\|_{1\to\infty} = \max_i \max_j |M_{ij}| \tag{30}$$

By definition of the Kronecker product and the above inequality,

$$\|A \otimes B\|_{1\to\infty} = \max_{i,j} \max_{k,l} |A_{ij}B_{k,l}| \leq \max_{i,j}|A_{ij}| \max_{k,l}|B_{k,l}|$$

$\square$

## C. Properties of $(H_0, H_1)$-**NS Functions**

**Lemma 2.** *Consider a function $g : \mathbb{R} \to \mathbb{R}$. If $g(t) \geq 0$ for all t, and for constants $H_0 \geq 0, H_1 > 0$, $g''(t) \leq H_0 + H_1\, g(t)$, then, $g'(0) \leq \sqrt{2H_0\, g(0) + H_1\, g(0)^2}$.*

*Proof.* For convenience, we define $\lambda = \sqrt{H_1}$ and $c = \frac{H_0}{H_1}$. Define the function

$$\phi(t) := (g(0) + c)\, \cosh(\lambda t) + \frac{g'(0)}{\lambda}\, \sinh(\lambda t) - c,$$

where $\sinh(x) := \frac{\exp(x) - \exp(-x)}{2}$ and $\cosh(x) = \frac{\exp(x) + \exp(-x)}{2}$ are hyperbolic sine and cosine functions. We can verify the following relations: $\phi(0) = g(0)$, and

$$\phi'(t) = \frac{(g(0) + c)\,\lambda}{2}\,(\exp(\lambda t) - \exp(-\lambda t)) + \frac{g'(0)}{2}\,(\exp(\lambda t) + \exp(-\lambda t)) \implies \phi'(0) = g'(0)$$

$$\phi''(t) = \frac{(g(0) + c)\,\lambda^2}{2}\,(\exp(\lambda t) + \exp(-\lambda t)) + \frac{g'(0)\,\lambda}{2}\,(\exp(\lambda t) - \exp(-\lambda t)) \implies \phi''(t) = \lambda^2\,(\phi(t) + c) = H_0 + H_1\,\phi(t)$$

Hence, $\phi(t)$ satisfies a similar condition as $g$, but with an equality. We now show that $\phi(t)$ upper-bounds $g(t)$. For this define

$$\Delta(t) = \phi(t) - g(t)$$

and verify that (i) $\Delta''(t) \geq \lambda^2\,\Delta(t)$, (ii) $\Delta'(0) = 0$ and (iii) $\Delta(0) = 0$. We define $H(t)$ and calculate its derivative:

$$H(t) := \exp(-\lambda t)\,[\Delta'(t) + \lambda\,\Delta(t)] \quad ; \quad H'(t) = \exp(-\lambda t)\,[\Delta''(t) - \lambda^2\Delta(t)] \geq 0$$

Hence, $H(t)$ is a non-decreasing function, and hence for all $t \geq 0$, $H(t) \geq H(0) = 0$. Hence, for all $t \geq 0$,

$$H(t) = \exp(-\lambda t)\,[\Delta'(t) + \lambda\,\Delta(t)] \geq 0 \implies \exp(2\lambda(t))\,H(t) = \exp(\lambda t)\,[\Delta'(t) + \lambda\,\Delta(t)] \geq 0$$

$$\implies \frac{\partial \exp(\lambda t)\,\Delta(t)}{\partial t} \geq 0$$

Hence, $h(t) := \exp(\lambda t)\,\Delta(t)$ is a non-decreasing function, and hence, for all $t \geq 0$, $h(t) \geq h(0) = 0$. This implies that $\Delta(t) \geq 0$ for all $t \geq 0$.
Hence, for $t \geq 0$, $\phi(t) \geq g(t)$. By defining $\tilde{H}(t) := \exp(\lambda t)\,[\Delta'(t) - \lambda\,\Delta(t)]$ and following an analogous argument, we prove that for all $t \leq 0$, $\phi(t) \geq g(t)$.

$$\tilde{H}(t) := \exp(\lambda t)\,[\Delta'(t) - \lambda\,\Delta(t)] \quad ; \quad \tilde{H}'(t) = \exp(\lambda t)\,[\Delta''(t) - \lambda^2\Delta(t)] \geq 0$$

Hence, $\tilde{H}(t)$ is a non-decreasing function, and hence for all $t \leq 0$, $\tilde{H}(t) \leq \tilde{H}(0) = 0$. Hence, for all $t \leq 0$,

$$\tilde{H}(t) = \exp(\lambda t)\,[\Delta'(t) - \lambda\,\Delta(t)] \leq 0 \implies \exp(-2\lambda(t))\,\tilde{H}(t) = \exp(-\lambda t)\,[\Delta'(t) - \lambda\,\Delta(t)] \leq 0$$

$$\implies \frac{\partial \exp(-\lambda t)\,\Delta(t)}{\partial t} \leq 0$$

Hence, $\tilde{h}(t) := \exp(-\lambda t)\,\Delta(t)$ is a non-increasing function, and hence, for all $t \leq 0$, $\tilde{h}(t) \geq \tilde{h}(0) = 0$. This implies that $\Delta(t) \geq 0$ for all $t \leq 0$.

Hence, for all $t \in \mathbb{R}$, $\phi(t) \geq g(t) \geq 0 \implies \min_{t \in \mathbb{R}} \phi(t) \geq 0$. Therefore, $\phi''(t) \geq 0$ for all $t$ and $\phi(t)$ is convex. Therefore, we can minimize $\phi(t)$ by setting $\phi'(t) = 0$:

$$\phi'(t) = (g(0) + c)\lambda \sinh(\lambda t) + g'(0) \cosh(\lambda t) = 0 \iff \tanh(\lambda t^*) = -\frac{g'(0)}{(g(0) + c)\lambda}$$

so since $\cosh(\tanh^{-1}(x)) = \frac{1}{\sqrt{1-x^2}}$ then

$$\min_{t \in R} \phi(t) = \left( (g(0) + c) - \frac{g'(0)^2}{\lambda^2(g(0) + c)} \right) \cosh(\lambda t^*) - c = \frac{1}{\lambda}\sqrt{(g(0) + c)^2\lambda^2 - g'(0)^2} - c.$$

Therefore

$$(g(0) + c)^2 - \frac{g'(0)^2}{\lambda^2} \geq c^2 \qquad\qquad \text{(since } \phi(t) \geq 0 \text{ for all } t \in \mathbb{R}\text{)}$$

$$\implies g'(0) \leq \lambda \left[ \sqrt{2cg(0) + g(0)^2} \right] = \sqrt{2H_0\, g(0) + H_1\, g(0)^2}$$

$\square$

**Lemma 3.** *If Assn. 1 and 2 hold with $H_0 \geq 0, H_1 \geq 0$, then, for all $\theta$, $\|\nabla f(\theta)\|_q \leq \sqrt{2H_0\, f(\theta) + H_1\, [f(\theta)]^2}$.*

*Proof.* If $H_1 = 0$, then using the standard descent lemma and the non-negativity of $f$ gives us that, for all $\theta$,

$$\|\nabla f(\theta)\|_q^2 \leq 2H_0\, f(\theta)$$

For the case where $H_1 > 0$, we define $g(t) := f(\theta + tu)$ s.t. $\|u\|_p \leq 1$. From assumption 1, we know that $g(t) \geq 0$ for all $t$. Furthermore,

$$\begin{aligned}
g'(t) &= \langle \nabla f(\theta + t\,u), u \rangle \\
g''(t) &= u^T \nabla^2 f(\theta + t\,u)\, u \leq \|u\|_p \left\| \nabla^2 f(\theta + t\,u)\, u \right\|_q && \text{(Holder's inequality)} \\
&= \|u\|_p^2 \left\| \nabla^2 f(\theta + t\,u) \right\|_{p \to q} && \text{(Definition of matrix norm)} \\
&\leq \|u\|_p^2 \left( H_0 + H_1\, f(\theta + t\,u) \right) && \text{(Using assumption 2)} \\
\implies g''(t) &\leq H_0 + H_1\, g(t) && \text{(Since } \|u\|_p \leq 1 \text{ and using assumption 1)}
\end{aligned}$$

Using Lemma. 2 for $H_0 \geq 0$ and $H_1 > 0$, we get that, $g(t) \geq 0$ and $g''(t) \leq H_0 + H_1\, g(t)$, then, $g'(0) \leq \sqrt{2H_0\, g(0) + H_1\, g(0)^2}$. Hence,

$$\langle \nabla f(\theta), u \rangle \leq \sqrt{2H_0\, f(\theta) + H_1\, [f(\theta)]^2} \implies \max_{\|u\|_p \leq 1} \langle \nabla f(\theta), u \rangle \leq \sqrt{2H_0\, f(\theta) + H_1\, [f(\theta)]^2}$$

$$\implies \|\nabla f(\theta)\|_q \leq \sqrt{2H_0\, f(\theta) + H_1\, [f(\theta)]^2} \qquad\qquad \text{(By definition of the dual norm)}$$

$\square$

**Lemma 4.** *If Assn. 1 and 2 hold with $H_0 = 0$, then, $f(y) \leq f(x) \exp\left( \sqrt{H_1}\, \|y - x\|_p \right)$.*

*Proof.* Define $g(\theta) := \ln(f(\theta))$, and note that $\|\nabla g(\theta)\|_q = \frac{\|\nabla f(\theta)\|_q}{f(\theta)} \leq \sqrt{H_1}$, as a consequence of assumption 2 and Lemma 3. Hence, $g$ is $\sqrt{H_1}$-Lipschitz. Hence, for all $y, x$

$$g(y) - g(x) \leq \sqrt{H_1}\, \|y - x\|_p \implies \ln\left( \frac{f(y)}{f(x)} \right) \leq \sqrt{H_1}\, \|y - x\|_p \implies f(y) \leq f(x) \exp\left( \sqrt{H_1}\, \|y - x\|_p \right).$$

$\square$

**Lemma 5.** *If Assn. 1 and 2 hold with* $H_1 > 0$*, then,* $f(y) \leq \left( f(x) + \frac{H_0}{H_1} \right) \exp \left( \sqrt{H_1} \left\| y - x \right\|_p \right) - \frac{H_0}{H_1} \leq \left( f(x) + \frac{H_0}{H_1} \right) \exp \left( \sqrt{H_1} \left\| y - x \right\|_p \right).$

*Proof.* Define $\tilde{f}(\theta) := f(\theta) + \frac{H_0}{H_1}$. We will show that if $f$ satisfies assumption 2 with $H_0 > 0, H_1 > 0$, then $\tilde{f}$ satisfies assumption 2 with $H_0 = 0, H_1 > 0$. For all $\theta$,

$$\left\| \nabla^2 \tilde{f}(\theta) \right\|_{p \to q} = \left\| \nabla^2 f(\theta) \right\|_{p \to q} \leq H_0 + H_1 \, f(\theta) \qquad \text{(Using assumption 2 for } f\text{)}$$

$$= H_0 + H_1 \left( \tilde{f}(\theta) - \frac{H_0}{H_1} \right) = H_0 + H_1 \, \tilde{f}(\theta) - H_0 = H_1 \, \tilde{f}(\theta) \qquad \text{(Since } \tilde{f}(\theta) = f(\theta) + \frac{H_0}{H_1}\text{)}$$

$$\implies \left\| \nabla^2 \tilde{f}(\theta) \right\|_{p \to q} \leq H_1 \tilde{f}(\theta)$$

Hence, $\tilde{f}(\theta)$ satisfies assumption 2 with $H_0 = 0$. Using Lemma. 4 for $\tilde{f}$,

$$\tilde{f}(y) \leq \tilde{f}(x) \exp \left( \sqrt{H_1} \left\| y - x \right\|_p \right)$$

$$\implies f(y) + \frac{H_0}{H_1} \leq \left( f(x) + \frac{H_0}{H_1} \right) \exp \left( \sqrt{H_1} \left\| y - x \right\|_p \right)$$

$$\implies f(y) \leq \left( f(x) + \frac{H_0}{H_1} \right) \exp \left( \sqrt{H_1} \left\| y - x \right\|_p \right) - \frac{H_0}{H_1}$$

$\square$

**Lemma 6.** *If Assn. 1 and 2 hold with* $H_0 \geq 0, H_1 \geq 0$*, then, for all* $y, x$ *s.t.* $\left\| y - x \right\|_p \leq \frac{1}{\sqrt{H_1}}$*,*

$$\left\| \nabla f(y) - \nabla f(x) \right\|_q \leq [H_0 + H_1 \, f(x)] \, e \, \left\| y - x \right\|_p .$$

*Proof.* By the fundamental theorem of calculus,

$$\nabla f(y) - \nabla f(x) = \int_{t=0}^{1} \nabla^2 f((1 - t) \, x + t \, y) \, (y - x) \, dt$$

$$\implies \left\| \nabla f(y) - \nabla f(x) \right\|_q = \left\| \int_{t=0}^{1} \nabla^2 f((1 - t) \, x + t \, y) \, (y - x) \, dt \right\|_q$$

$$\leq \int_{t=0}^{1} \left\| \nabla^2 f((1 - t) \, x + t \, y) \, (y - x) \right\|_q dt \qquad \text{(Triangle inequality)}$$

$$\leq \int_{t=0}^{1} \left\| \nabla^2 f((1 - t) \, x + t \, y) \right\|_{p \to q} \left\| y - x \right\|_p dt \qquad \text{(By definition of matrix norm)}$$

$$\leq \left\| y - x \right\|_p \left[ \int_{t=0}^{1} H_0 + H_1 \, f((1 - t) \, x + t \, y) \, dt \right] \qquad \text{(Using assumption 2)}$$

$$= \left\| y - x \right\|_p \left[ H_0 + H_1 \int_{t=0}^{1} f((1 - t) \, x + t \, y) \, dt \right]$$

$$\leq \left\| y - x \right\|_p \left[ H_1 \int_{t=0}^{1} \left( f(x) + \frac{H_0}{H_1} \right) \exp(\sqrt{H_1} \, t \, \left\| y - x \right\|) \, dt \right]$$

$$\text{(Using Lemma. 5 with } \theta = ty + (1 - t)x \text{ and } \theta' = x\text{)}$$

$$= \left\| y - x \right\|_p \left[ H_1 \left( f(x) + \frac{H_0}{H_1} \right) \int_{t=0}^{1} \exp(\sqrt{H_1} \, t \, \left\| y - x \right\|) \, dt \right]$$

$$\leq \left\| y - x \right\|_p \left[ H_1 \left( f(x) + \frac{H_0}{H_1} \right) \int_{t=0}^{1} \exp(t) \, dt \right] \qquad \left( \text{Since } \left\| y - x \right\|_p \leq \frac{1}{\sqrt{H_1}} \right)$$

$$= \left[ H_1 \left( f(x) + \frac{H_0}{H_1} \right) (e - 1) \right] \|y - x\|_p$$

$$\leq [H_0 + H_1 f(x)] e \|y - x\|_p$$

$$\square$$

**Lemma 7.** *If Assn. 1 and 2 hold with $H_0 \geq 0, H_1 \geq 0$, for all $y, x$ s.t. $\|y - x\| \leq \frac{1}{\sqrt{H_1}}$,*

$$f(y) \leq f(x) + \langle \nabla f(x), y - x \rangle + (H_0 + H_1 f(x)) \|y - x\|_p^2$$

*Proof.* Define $u(t) = (1 - t)x + ty$ and $g(t) = f(u(t))$. Use Taylor's theorem for $g$,

$$g(b) = g(a) + (b - a) g'(a) + \int_{t=a}^{b} (b - t) g''(t) \, dt$$

$$\implies g(1) = g(0) + g'(0) + \int_{t=0}^{1} (1 - t) g''(t) \, dt \qquad \text{(Substituting } a = 0 \text{ and } b = 1\text{)}$$

We know that,

$$g'(t) = \frac{\partial f(u(t))}{\partial t} = \langle \nabla f(u(t)), y - x \rangle \quad ; \quad g''(t) = \frac{\partial^2 f(u(t))}{\partial t^2} = (y - x)^T \nabla^2 f(u(t))(y - x)$$

Combining the above relations, and using that $g(1) = f(y), g(0) = f(x), g'(0) = \langle \nabla f(x), y - x \rangle$.

$$f(y) = f(x) + \langle \nabla f(x), y - x \rangle + (y - x)^T \left[ \int_{t=0}^{1} (1 - t) \nabla^2 f(t y + (1 - t) x) \, dt \right] (y - x)$$

Simplifying the last term,

$$(y - x)^T \left[ \int_{t=0}^{1} (1 - t) \nabla^2 f(t y + (1 - t) x) \, dt \right] (y - x)$$

$$\leq \|y - x\|_p \left\| \left[ \int_{t=0}^{1} (1 - t) \nabla^2 f(t y + (1 - t) x) \, dt \right] (y - x) \right\|_q \qquad \text{(Holder's inequality)}$$

$$\leq \|y - x\|_p \left[ \int_{t=0}^{1} (1 - t) \left\| \nabla^2 f(t y + (1 - t) x) (y - x) \right\|_q dt \right] \qquad \text{(Triangle inequality)}$$

$$\leq \|y - x\|_p^2 \int_{t=0}^{1} (1 - t) \left\| \nabla^2 f(t y + (1 - t) x) \right\|_{p \to q} dt \qquad \text{(By definition of matrix norm)}$$

$$\leq \|y - x\|_p^2 \int_{t=0}^{1} (1 - t) [H_0 + H_1 f(t y + (1 - t) x)] \, dt \qquad \text{(Using assumption 2)}$$

$$\leq \|y - x\|_p^2 \int_{t=0}^{1} (1 - t) \left( H_0 + H_1 \left[ \left( f(x) + \frac{H_0}{H_1} \right) \exp(\sqrt{H_1} \, t \, \|y - x\|_p) - \frac{H_0}{H_1} \right] \right) dt$$

$$\text{(Using Lemma. 5 with } \theta = ty + (1 - t)x \text{ and } \theta' = x\text{)}$$

$$= \|y - x\|_p^2 \int_{t=0}^{1} (1 - t) \left( (H_0 + H_1 f(x)) \exp(\sqrt{H_1} \, t \, \|y - x\|_p) \right) dt$$

$$= \|y - x\|_p^2 (H_0 + H_1 f(x)) \int_{t=0}^{1} (1 - t) \left( \exp(\sqrt{H_1} \, t \, \|y - x\|_p) \right) dt$$

$$\leq \|y - x\|_p^2 (H_0 + H_1 f(x)) \int_{t=0}^{1} (1 - t) \left( \exp(t) \right) dt \qquad \text{(Since } \|y - x\|_p \leq \frac{1}{\sqrt{H_1}}\text{)}$$

$$\leq \|y - x\|_p^2 (H_0 + H_1 f(x)) \left[ \int_{t=0}^{1} \exp(t) \, dt - \int_{t=0}^{1} t \, \exp(t) \, dt \right]$$

$$\leq \|y - x\|_p^2 \, (H_0 + H_1 \, f(x)) \, (e - 2)$$
$$\leq \|y - x\|_p^2 \, (H_0 + H_1 \, f(x))$$

Putting everything together,

$$f(y) \leq f(x) + \langle \nabla f(x), y - x \rangle + (H_0 + H_1 \, f(x)) \, \|y - x\|_p^2$$

$\square$

**Lemma 8.** *If Assn. 1 and 2 hold with $H_0 = 0$, then, for all $x, y$, $\left\| \frac{\nabla f(y)}{f(y)} - \frac{\nabla f(x)}{f(x)} \right\|_q \leq 2\,H_1 \, \|y - x\|_p$.*

*Proof.*

$$\nabla^2 \ln(f(\theta)) = \frac{\nabla^2 f(\theta)}{f(\theta)} - \frac{[\nabla f(\theta)]\,[\nabla f(\theta)]^T}{[f(\theta)]^2}$$

$$\implies \left\| \nabla^2 \ln(f(\theta)) \right\|_{p \to q} \leq \left\| \frac{\nabla^2 f(\theta)}{f(\theta)} \right\|_{p \to q} + \left\| \frac{[\nabla f(\theta)]\,[\nabla f(\theta)]^T}{[f(\theta)]^2} \right\|_{p \to q} \qquad \text{(Triangle inequality)}$$

$$\leq H_1 + \frac{1}{[f(\theta)]^2} \left\| [\nabla f(\theta)]\,[\nabla f(\theta)]^T \right\|_{p \to q} \qquad \text{(Using assumption 2)}$$

$$\leq H_1 + \frac{\|\nabla f(\theta)\|_q^2}{[f(\theta)]^2} \qquad \text{(Since for any vector } x, \, \left\| x x^T \right\|_{p \to q} = \|x\|_q^2 \text{)}$$

$$\implies \left\| \nabla^2 \ln(f(\theta)) \right\|_{p \to q} \leq 2\,H_1 \qquad \text{(Using Lemma. 3 with } H_0 = 0 \text{)}$$

By the fundamental theorem of calculus,

$$\nabla \ln(f(y)) - \nabla \ln(f(x)) = \int_{t=0}^1 \nabla^2 \ln(f(t\,y + (1-t)x))(y - x)\,dt$$

$$\implies \|\nabla \ln(f(y)) - \nabla \ln(f(x))\|_q \leq \int_{t=0}^1 \left\| \nabla^2 \ln(f(t\,y + (1-t)x))(y-x) \right\|_q dt \qquad \text{(Triangle inequality)}$$

$$\leq \int_{t=0}^1 \left\| \nabla^2 \ln(f(t\,y + (1-t)x)) \right\|_{p \to q} \|y - x\|_p \, dt$$
$$\text{(By definition of matrix norm)}$$

$$\leq 2\,H_1 \, \|y - x\|_p \qquad \text{(Using the above relation)}$$

$$\implies \left\| \frac{\nabla f(y)}{f(y)} - \frac{\nabla f(x)}{f(x)} \right\|_q \leq 2\,H_1 \, \|y - x\|_p$$

$\square$

**Lemma 9.** *If Assn. 1 and 2 hold with $H_1 > 0$, then, for all $x, y$ s.t. $\|y - x\|_p \leq \frac{1}{\sqrt{H_1}}$, $\left\| \frac{\nabla f(y)}{f(y) + \frac{H_0}{H_1}} - \frac{\nabla f(x)}{f(x) + \frac{H_0}{H_1}} \right\|_q \leq 2\,H_1 \, \|y - x\|_p$.*

*Proof.* Define $\tilde{f}(\theta) := f(\theta) + \frac{H_0}{H_1}$. We will show that if $f$ satisfies assumption 2 with $H_0 > 0, H_1 > 0$, then $\tilde{f}$ satisfies assumption 2 with $H_0 = 0, H_1 > 0$. For all $\theta$,

$$\left\| \nabla^2 \tilde{f}(\theta) \right\|_{p \to q} = \left\| \nabla^2 f(\theta) \right\|_{p \to q} \leq H_0 + H_1 \, f(\theta) \qquad \text{(Using assumption 2 for f)}$$

$$= H_0 + H_1 \left( \tilde{f}(\theta) - \frac{H_0}{H_1} \right) = H_0 + H_1 \, \tilde{f}(\theta) - H_0 = H_1 \, \tilde{f}(\theta) \qquad \text{(Since } \tilde{f}(\theta) = f(\theta) + \frac{H_0}{H_1} \text{)}$$

$$\implies \left\| \nabla^2 \tilde{f}(\theta) \right\|_{p \to q} \leq H_1 \tilde{f}(\theta)$$

Hence, $\tilde{f}(\theta)$ satisfies assumption 2 with $H_0 = 0$. Using Lemma. 8 for $\tilde{f}$ and noting that $\nabla \tilde{f}(\theta) = f(\theta)$, then, for all $x, y$,

$$\left\| \frac{\nabla \tilde{f}(y)}{\tilde{f}(y)} - \frac{\nabla \tilde{f}(x)}{\tilde{f}(x)} \right\|_q \leq 2 H_1 \, \|y - x\|_p$$

$$\implies \left\| \frac{\nabla f(y)}{f(y) + \frac{H_0}{H_1}} - \frac{\nabla f(x)}{f(x) + \frac{H_0}{H_1}} \right\|_q \leq 2 H_1 \, \|y - x\|_p$$

$\square$

**Lemma 10.** *If Assn. 1 and 2 hold with $H_0 \geq 0$, $H_1 \geq 0$ and $\frac{1}{p} + \frac{1}{q} = 1$, then, for all $y, x$ s.t. $\|y - x\|_p \leq \frac{1}{\sqrt{H_1}}$ and $c > 0$,*

$$\|\nabla f(y)\|_q + c \leq \left( \|\nabla f(x)\|_q + c \right) \exp\left( (H_0 + H_1 \, f(x)) \, e \, \frac{\|y - x\|_p}{c} \right)$$

*Furthermore, if $H_1 > 0$ and assumption 3 holds with $\zeta = 1$ and $\mu$, then, for all $y, x$ and choosing $c = \frac{\mu \, (H_0 + H_1 \, f^*)}{H_1}$,*

$$\|\nabla f(y)\|_q + c \leq \left( \|\nabla f(x)\|_q + c \right) \exp\left( \frac{H_1}{\mu} \, \|y - x\|_p \right),$$

*Proof.* Define the function $h(\theta) := \ln(\|\nabla f(\theta)\|_q + c)$. We will first prove that $h(\theta)$ is $\frac{H_1}{\mu}$-Lipschitz w.r.t the $\ell_p$ norm. Since $\|\cdot\|_q$ can be non-smooth, consider a Clarke subgradient computed using Lemma 12

$$\partial h(\theta) = \frac{\nabla^2 f(\theta) \partial \|z\|_q}{\|z\|_q + c}, \qquad z = \nabla f(\theta).$$

Then, for $g = \nabla^2 f(\theta) s$ where $s \in \partial(\|\cdot\|_q)(\nabla f(\theta))$ is a subgradient of $\|\cdot\|_q$ evaluated at $\nabla f(\theta)$.

$$\|v\|_q = \frac{\left\| [\nabla^2 f(\theta)] \, s \right\|_q}{\|\nabla f(\theta)\|_q + c} \leq \frac{\left\| \nabla^2 f(\theta) \right\|_{p \to q} \|s\|_p}{\|\nabla f(\theta)\|_q + c} \qquad \text{(By definition of the matrix norm)}$$

$$\implies \|v\|_q \leq \frac{\left\| \nabla^2 f(\theta) \right\|_{p \to q}}{\|\nabla f(\theta)\|_q + c} \qquad \text{(If } s \in \partial(\|\cdot\|_q), \text{ then, } \|s\|_p \leq 1)$$

Using assumption 2 to simplify the numerator, and noting that lower-bounding the denominator,

$$\forall v \in \partial h(\theta), \quad \|v\|_q \leq \frac{H_0 + H_1 \, f(\theta)}{c}$$

By Lebourg's mean value theorem ((Clarke et al., 1998), Thm. 2.4), since $h$ is Lipschitz in an open set containing $y$ and $x$, then there exists a point $u = tx + (1 - t)y$, for some $t \in [0, 1]$ such that

$$h(y) - h(x) \in \langle g, y - x \rangle, \qquad g \in \partial h(u).$$

Therefore,

$$h(y) - h(x) \leq \max_{t \in [0,1]} \langle g, y - x \rangle \quad where \quad g \in \partial h(t \, x + (1 - t) \, y)$$

$$\leq \|y - x\|_p \max_{t \in [0,1]} \|g\|_q \qquad \text{(Using Holder's inequality)}$$

$$\leq \frac{\|y - x\|_p}{c} \max_{t \in [0,1]} (H_0 + H_1 \, f(t \, x + (1 - t) \, y)), \qquad \text{(Using the above bound on the subgradient)}$$

$$\leq \frac{\|y - x\|_p}{c} \max_{t \in [0,1]} \left[ H_0 + H_1 \left( \left( f(x) + \frac{H_0}{H_1} \right) \exp \left( \sqrt{H_1}\, t\, \|y - x\|_p \right) - \frac{H_0}{H_1} \right) \right] \quad \text{(Using Lemma. 5)}$$

$$= \frac{\|y - x\|_p}{c} \max_{t \in [0,1]} (H_0 + H_1\, f(x)) \exp \left( \sqrt{H_1}\, t\, \|y - x\|_p \right)$$

$$\leq \frac{\|y - x\|_p}{c} \max_{t \in [0,1]} (H_0 + H_1\, f(x)) \exp (t) \qquad \text{(Since } \|y - x\|_p \leq \tfrac{1}{\sqrt{H_1}} \text{)}$$

$$\leq \frac{\|y - x\|_p}{c} (H_0 + H_1\, f(x))\, e$$

$$\implies \ln(\|\nabla f(y)\|_q + c) - \ln(\|\nabla f(x)\|_q + c) \leq (H_0 + H_1\, f(x))\, e\, \frac{\|y - x\|_p}{c}$$

$$\implies \|\nabla f(y)\|_q + c \leq \left( \|\nabla f(x)\|_q + c \right) \exp \left( (H_0 + H_1\, f(x))\, e\, \frac{\|y - x\|_p}{c} \right)$$

This proves the first part of the lemma. For the second part, using assumption 2 to simplify the numerator of the above relation,

$$\left\| \nabla^2 f(\theta) \right\|_{p \to q} \leq H_0 + H_1\, f(\theta) = [H_0 + H_1\, f^*] + H_1\, [f(\theta) - f^*]$$

$$\leq [H_0 + H_1\, f^*] + \frac{H_1}{\mu} \|\nabla f(\theta)\|_q \qquad \text{(Using assumption 3 with } \zeta = 1 \text{ and } \mu \text{)}$$

$$= \frac{H_1}{\mu} \left[ \|\nabla f(\theta)\|_q + \frac{\mu\, [H_0 + H_1\, f^*]}{H_1} \right] = \frac{H_1}{\mu} \left[ \|\nabla f(\theta)\|_q + c \right] \qquad \text{(By definition of } c \text{)}$$

$$\implies \forall v \in \partial h(\theta), \quad \|v\|_q \leq \frac{H_1}{\mu}$$

Again, using the mean value theorem

$$h(y) - h(x) \leq \max_{t \in [0,1]} \langle g, y - x \rangle \quad where \quad g \in \partial h(t\, x + (1 - t)\, y)$$

$$\leq \|y - x\|_p \max_{t \in [0,1]} \|g\|_q \qquad \text{(Using Holder's inequality)}$$

$$\leq \|y - x\|_p \frac{H_1}{\mu} \qquad \text{(Using the above bound on the subgradient)}$$

$$\implies h(y) - h(x) \leq \frac{H_1}{\mu} \|y - x\|_p$$

$$\implies \|\nabla f(y)\|_q + c \leq \left( \|\nabla f(x)\|_q + c \right) \exp \left( \frac{H_1}{\mu} \|y - x\|_p \right),$$

$$\square$$

**Lemma 11.** *If $f$ satisfies Assn. 1 and Assn. 2 with constants $H_0 \geq 0, H_1 \geq 0$ such that $f^* = \min f(\theta) \neq 0$, then, $h(\theta) = f(\theta) - f^*$ satisfies Assn. 1 with $h^* = \min h(\theta) = 0$ and Assn. 2 with $L_0' := H_0 + H_1\, f^*$ and $L_1' = H_1$.*

*Proof.* By definition, since $f(\theta) \geq f^*$, $h$ satisfies Assn. 1 and $h^* = 0$. Since $f$ is $(H_0, H_1)$ NS,

$$\left\| \nabla^2 f(\theta) \right\| \leq H_0 + H_1\, f(\theta)$$

$$\implies \left\| \nabla^2 h(\theta) \right\| = \left\| \nabla^2 f(\theta) \right\| \leq H_0 + H_1\, f(\theta)$$

$$= \underbrace{H_0 + H_1\, f^*}_{:= L_0'} + \underbrace{H_1}_{:= L_1'} [f(\theta) - f^*]$$

$$\implies \left\| \nabla^2 h(\theta) \right\| \leq L_0' + L_1'\, h(\theta)$$

$$\square$$

### C.1. Definitions & Helper Lemmas

**Definition 1** (Clarke subdifferential (Clarke et al., 1998))**.** *For a nonconvex locally Lipschitz function $f : \mathbb{R}^n \to \mathbb{R}$, the Clarke subdifferential is defined as*

$$\partial f(x) := \mathrm{cl\,co} \left\{ \lim_{x_k \to x} \nabla f(x_k) \right\}$$

*where* $\mathrm{cl\,co}$ *is the closed convex hull of the set, and $x_k$ forms a trajectory of points for which $\nabla f(x_k)$ exists. If additionally $f$ is convex, then $\partial f(x)$ is the standard convex subdifferential.*

**Lemma 12** (Chain rule for Clarke subdifferentials)**.** *Suppose $h(x) = f(g(x))$, and both $f$ and $g$ are locally Lipschitz. Then,*

1. *if $f : \mathbb{R} \to \mathbb{R}$ smooth and $g : \mathbb{R}^n \to \mathbb{R}$ convex but not smooth, then*

$$\partial h(x) = f'(g(x)) \cdot \partial g(x)$$

2. *If $f : \mathbb{R}^n \to \mathbb{R}$ nonsmooth but continuous, and $g : \mathbb{R}^p \to \mathbb{R}^n$ smooth then*

$$\partial h(x) = J(x)^T y, \qquad u \in \partial f(u), \ u = g(x)$$

*where $J(x) = [\nabla g_1(x), ..., \nabla g_n(x)]^T$ the Jacobian of the mapping $g$.*

*Proof.* The first statement is given exactly in (Clarke et al., 1998). Both are the result of the following statement, which is follows from the standard definition of lim sup:

Consider $A(t) \to a$ and $\limsup_{t \to 0} B(t) = \mathcal{B}$ where $a$ is a point and $\mathcal{B}$ is a closed set. Then if $a \in \mathbb{R}$ or $\mathcal{B} \subset \mathbb{R}$, then

$$\limsup_{t \to 0} A(t) \cdot B(t) \subset a \cdot \mathcal{B} = \{ab : b \in \mathcal{B}\}.$$

If $h(x)$ is smooth at $x$, then

$$\partial h(x) = \{\nabla h(x)\} = \begin{cases} \{f'(g(x)) \cdot \nabla g(x)\}, & g : \mathbb{R}^n \to \mathbb{R} \\ \{J(x)^T \nabla f(g(x))\}, & g : \mathbb{R}^p \to \mathbb{R}^n \end{cases}$$

If $h(x)$ is not smooth at $x$, then in the first case,

$$\limsup_{z \to x} \nabla h(z) = \limsup_{z \to x} f'(g(z)) \cdot \nabla g(z) \subseteq f'(g(x)) \cdot \partial g(x).$$

Since subdifferentials are closed and convex, then

$$\partial h(x) = \mathrm{co\,cl} \limsup_{z \to x} \nabla h(z) = f'(g(x)) \cdot \partial g(x).$$

By similar logic, in the second case, for $w = g(z)$, $u = g(x)$

$$\limsup_{z \to x} \nabla h(z) = \limsup_{z \to x} J(z)^T \nabla f(w) \subset J(x)^T \limsup_{w \to u} \nabla f(w) = J(x)^T \partial f(u)$$

which are also closed and convex sets. So,

$$\partial h(x) = \{J(x)^T y : y \in \partial f(u), \ u = g(x)\}.$$

$\square$

## D. Normalized Steepest Descent

**Proposition 7.** *If $\nabla_t := \nabla f(\theta_t)$ with $\nabla_{t,i}$ denoting coordinate $i$ of this vector, and we use $\text{sign}(\nabla_t) \in \{-1, 0, 1\}^D$ to denote the element-wise sign operation with $\text{sign}(0) := 0$ and $e_i$ denotes the $i$-th standard basis vector, then the* `NSD` *update in Eq.* (14) *recovers the following special cases:*

- $(p, q) = (\infty, 1)$, $\theta_{t+1} = \theta_t - \eta_t \, \text{sign}(\nabla_t)$ *(*`Sign GD`*)*

- $(p, q) = (2, 2)$, $\theta_{t+1} = \theta_t - \eta_t \frac{\nabla_t}{\|\nabla_t\|_2}$ *(*`Norm.GD`*)*

- $(p, q) = (1, \infty)$, $\theta_{t+1} = \theta_t - \eta_t \, \text{sign}(\nabla_{t,i_t}) \, e_{i_t}$, *where,* $i_t \in \arg\max_{i \in [D]} |\nabla_{t,i}|$ *(*`Sign CD-GS`*)*

*Proof.* • For $(p, q) = (\infty, 1)$, for $d$ s.t. $\|d\|_\infty \leq 1$,

$$\langle \nabla_t, d \rangle \leq \|\nabla_t\|_1 \, \|d\|_\infty \leq \|\nabla_t\|_1 \qquad \text{(By Holder's inequality, and since } \|d\|_\infty \leq 1)$$

For $d^* = \text{sign}(\nabla_t)$ where $\text{sign}(0) = 0$, $\|d^*\|_\infty = 1$ and

$$\langle \nabla_t, d^* \rangle = \sum_{i \in [D], \nabla_{t,i} \neq 0} \nabla_{t,i} \, d_i^* + \sum_{i \in [D], \nabla_{t,i} = 0} \nabla_{t,i} \, d_i^* = \sum_{i \in [D], \nabla_{t,i} \neq 0} \nabla_{t,i} \, \text{sign}(\nabla_{t,i}) = \sum_{i \in [D], \nabla_{t,i} \neq 0} |\nabla_{t,i}| = \|\nabla_t\|_1$$

Hence, $d^*$ attains the upper-bound, and is therefore optimal. Hence, the `NSD` update simplifies to $\theta_{t+1} = \theta_t - \eta_t \, \text{sign}(\nabla_t)$.

• For $(p, q) = (2, 2)$, for $d$ s.t. $\|d\|_2 \leq 1$,

$$\langle \nabla_t, d \rangle \leq \|\nabla_t\|_2 \, \|d\|_2 \leq \|\nabla_t\|_2 \qquad \text{(By Cauchy Schwarz and since } \|d\|_2 \leq 1)$$

For $d^* := \frac{\nabla_t}{\|\nabla_t\|_2}$, $\|d^*\|_2 = 1$, and $\langle \nabla_t, d^* \rangle = \|\nabla_t\|_2$. Hence, $d^*$ attains the upper-bound, and is therefore optimal. Hence, the `NSD` update simplifies to $\theta_{t+1} = \theta_t - \eta_t \frac{\nabla_t}{\|\nabla_t\|_2}$

• For $(p, q) = (1, \infty)$, for $d$ s.t. $\|d\|_1 \leq 1$,

$$\langle \nabla_t, d \rangle \leq \|\nabla_t\|_\infty \, \|d\|_1 \leq \|\nabla_t\|_\infty \qquad \text{(By Holder's inequality, and since } \|d\|_1 \leq 1)$$

Pick a coordinate $i_t \in \arg\max_{i \in [D]} |\nabla_{t,i}|$ and define $d^* = \text{sign}(\nabla_{t,i_t}) \, e_{i_t}$. For $d^*$, $\|d^*\|_1 = 1$ and

$$\langle \nabla_t, d^* \rangle = \sum_i \nabla_{t,i} \, d_i^* = \nabla_{t,i_t} \, d_{i_t}^* = \nabla_{t,i_t} \, \text{sign}(\nabla_{t,i_t}) = |\nabla_{t,i_t}| = \max_{i \in [D]} |\nabla_{t,i}| = \|\nabla_t\|_\infty$$

Hence, $d^*$ attains the upper-bound, and is therefore optimal. Hence, the `NSD` update simplifies to $\theta_{t+1} = \theta_t - \eta_t \, \text{sign}(\nabla_{t,i_t}) \, e_{i_t}$, where, $i_t \in \arg\max_{i \in [D]} |\nabla_{t,i}|$. $\qquad\square$

**Theorem 1.** *Under Assn.* 1, *Assn.* 2 *and Assn.* 3 *with $\mu(\theta) = \mu$ for all $\theta$, $f^* = 0$,* `NSD` *converges as:*

- *if $\epsilon > \frac{H_0}{H_1}$, using $\eta_t = \eta = O(1)$ guarantees that after $T = O\left(\ln\left(\frac{1}{\epsilon}\right)\right)$ iterations, $f(\theta_{T+1}) \leq \epsilon$;*

- *else, using $\eta_t = \eta = O(\epsilon^\tau)$ guarantees that after $T = O\left(\frac{1}{\epsilon^\tau}\right)$ iterations, $f(\theta_{T+1}) \leq \epsilon$.*

*Proof.* Using Lemma. 15 for $\eta \leq \frac{1}{\sqrt{H_1}}$,

$$f(\theta_{t+1}) \leq f(\theta_t) - \eta \, \|\nabla f(\theta_t)\|_q + (H_0 + H_1 \, f(\theta_t)) \, \eta^2$$
$$\leq f(\theta_t) - \eta \, \mu \, [f(\theta_t)]^\tau + (H_0 + H_1 \, f(\theta_t)) \, \eta^2 \qquad \text{(Using Assn. 3 with } f^* = 0 \text{ and } \mu(\theta) = \mu)$$
$$\implies f(\theta_{t+1}) \leq f(\theta_t) - \eta \, \mu \, [f(\theta_t)]^\tau + (H_0 + H_1 \, f(\theta_t)) \, \eta^2 \qquad (31)$$

Define $T_0$ to be the first iteration s.t. $f(\theta_{T_0}) < \max\left\{\epsilon, \frac{H_0}{H_1}\right\}$.

**Phase 1**: We first analyze Eq. (31) for $t < T_0$ where $f(\theta_t) \geq \max\left\{\epsilon, \frac{H_0}{H_1}\right\}$. Simplifying Eq. (31) in this case,

$$f(\theta_{t+1}) \leq f(\theta_t) - \eta \, \mu \, [f(\theta_t)]^\tau + 2 \, H_1 \, f(\theta_t) \, \eta^2$$

We will now use an inductive proof and prove that for all $t \leq T_0$, $f(\theta_t) \leq f(\theta_1)$.

**Base Case**: For $t = 1$, this is true by definition.

**Inductive Hypothesis**: Assume for iteration $t < T_0$, $f(\theta_t) \leq f(\theta_1)$.

**Induction**: We will prove that $f(\theta_{t+1}) \leq f(\theta_1)$. For this, note that $f(\theta_t) \leq f(\theta_1)$ by the inductive hypothesis and $1 - \tau \geq 0$. Hence, the above inequality can simplified as:

$$f(\theta_{t+1}) \leq f(\theta_t) - \eta \, \mu \, \frac{f(\theta_t)}{[f(\theta_1)]^{1-\tau}} + 2H_1 \, f(\theta_t) \, \eta^2$$

Using Lemma. 13 with $A = \frac{\mu}{[f(\theta_1)]^{1-\tau}}$, $B = 2H_1$, $\bar{\eta}_1 = \frac{1}{\sqrt{H_1}}$ and $\delta = \max\left\{\epsilon, \frac{H_0}{H_1}\right\}$, we get that with $\eta = \min\left\{\bar{\eta}_1, \frac{\mu}{4\,H_1\,[f(\theta_1)]^{1-\tau}}\right\}$, $f(\theta_{t+1}) \leq f(\theta_t)$. This completes the induction. Moreover, $f(\theta_{T_0}) \leq \max\left\{\epsilon, \frac{H_0}{H_1}\right\}$ after

$$T_0 \geq O\left(\frac{[f(\theta_1)]^{1-\tau}}{\mu\,\eta}\ln\left(\frac{f(\theta_1)}{\delta}\right)\right) = O\left(\ln\left(\min\left\{\frac{1}{\epsilon}, \frac{H_1}{H_0}\right\}\right)\right) \text{ iterations.}$$

Hence, if $\epsilon \geq \frac{H_0}{H_1}$, NSD with a constant step-size requires $O(\ln(1/\epsilon))$ iterations to guarantee convergence to an $\epsilon$ sub-optimality.

**Phase 2**: Consider iteration $t > T_0$ such that $f(\theta_t) < \frac{H_0}{H_1}$. Simplifying Eq. (31) in this case,

$$f(\theta_{t+1}) \leq f(\theta_t) - \eta \, \mu \, [f(\theta_t)]^\tau + 2H_0 \, \eta^2$$

Using Lemma. 14 with $A = \mu$, $B = 2H_0$, $r = \tau$, $\bar{\eta}_1 = \frac{1}{\sqrt{H_1}}$ and $\delta = \epsilon$, we get that with $\eta = \min\left\{\frac{1}{\sqrt{H_1}}, \frac{\mu\,\epsilon^\tau}{4H_0}\right\}$, $f(\theta_{t+1}) \leq f(\theta_t) \leq f(\theta_{T_0}) \leq \frac{H_0}{H_1}$ for all $t \geq T_0$.

Furthermore, if $\tau = 1$, $f(\theta_{T_\epsilon + T_0}) \leq \epsilon$ after

$$T_\epsilon \geq \frac{2}{\mu\,\eta}\ln\left(\frac{f(\theta_1)}{\delta}\right) = O\left(\frac{1}{\epsilon}\right) \text{ iterations.}$$

Else, if $\tau < 1$, $f(\theta_{T_\epsilon + T_0}) \leq \epsilon$ after

$$T_\epsilon \geq \frac{2\,[f(\theta_{T_0})]^{1-\tau}}{\mu\,\eta\,(1-\tau)} = O\left(\frac{1}{\epsilon^\tau}\right) \text{ iterations.}$$

Hence, for $T := T + T_\epsilon$, $f(\theta_{T+1}) \leq \epsilon$ after,

$$O\left(\frac{1}{\epsilon^\tau} + \ln\left(\frac{H_1}{H_0}\right)\right) \text{ iterations.}$$

$\square$

**Corollary 1.** *For the multi-armed bandit problem in Prop. 3, NSD with a uniform initialization i.e. $\forall a$, $\pi_{\theta_1}(a) = 1/K$ and $\eta = O(1)$ requires $T = O\left(\ln\left(1/\epsilon\right)\right)$ iterations to guarantee $\langle \pi_{\theta_{T+1}}, r \rangle \geq r(a^*) - \epsilon$.*

*Proof.* From Prop. 3, we know that the multi-armed bandit problem satisfies Assn. 2 with $H_0 = 0$ and Assn. 3 with $\tau = 1$ and $f^* = 0$, though with a non-uniform $\mu(\theta)$. In order to use Thm. 1, we need to lower-bound $\mu(\theta) = \pi_\theta(a^*)$ by a constant $\mu$. For this, we use a uniform initialization which guarantees that $\pi_{\theta_1}(a^*) = \frac{1}{K}$.

We then use Mei et al. (2020, Proposition 2) which implies that if $f(\theta_{t+1}) \leq f(\theta_t)$, $\pi_{\theta_{t+1}}(a^*) \geq \pi_{\theta_t}(a^*)$. For NSD on a function with a fixed $\mu$, Thm. 1 guarantees descent for all $t$ in both phases. Consequently, we can inductively conclude that for all $t$, $\pi_{\theta_{t+1}}(a^*) \geq \pi_{\theta_t}(a^*) \geq \pi_{\theta_1}(a^*) \geq \mu := \frac{1}{K}$.

Hence, Thm. 1 immediately recovers a linear convergence rate with $\mu = \frac{1}{K}$. $\square$

**Theorem 2.** *For logistic regression on separable data with the margin $\gamma_p$ (see Prop. 2), NSD with $\theta_1 = 0$, $\eta_t = \eta = O(1)$ and $T = O\left(\frac{H_1}{\gamma_p^2}\left[n^2 + \ln\left(\frac{1}{n\epsilon}\right)\right]\right)$ iterations guarantees that $f(\theta_{T+1}) \leq \epsilon$.*

*Proof.* Since we are considering logistic regression on linearly separable data, $f^* = 0$ and $H_0 = 0$. We will use a similar proof as that for Thm. 1. In particular, using Lemma. 15 for $\eta \leq \bar{\eta}_0 := \frac{1}{\sqrt{H_1}}$,

$$f(\theta_{t+1}) \leq f(\theta_t) - \eta\, \|\nabla f(\theta_t)\|_q + H_1\, f(\theta_t)\, \eta^2 \tag{32}$$

We will use an inductive argument and prove that for all $t \geq 0$, $f(\theta_t) \leq f(\theta_1)$.

**Base Case**: For $t = 1$, this is true by definition.

**Inductive Hypothesis**: Assume for iteration $t > 0$, $f(\theta_t) \leq f(\theta_1)$.

**Induction**: We will prove that $f(\theta_{t+1}) \leq f(\theta_1)$. To complete the induction, we will consider two phases. For this, define $T_0$ to be the first iteration s.t. $f(\theta_{T_0}) < \max\left\{\epsilon, \frac{\ln(2)}{n}\right\}$.

**Phase 1:** For all $t < T_0$, $f(\theta_t) > \frac{\ln(2)}{n}$ and we use part (2) of Prop. 2 to conclude that $\|\nabla_t\|_q \geq \frac{\gamma_p}{3n}$. Simplifying Eq. (32),

$$
\begin{aligned}
f(\theta_{t+1}) &\leq f(\theta_t) - \eta\,\|\nabla f(\theta_t)\|_q + H_1\, f(\theta_1)\, \eta^2 && \text{(Using the inductive hypothesis)} \\
&\leq f(\theta_t) - \frac{\eta\,\gamma_p}{4n} + H_1\, f(\theta_1)\, \eta^2 && \text{(Using that for } t < T_0, \|\nabla_t\|_q \geq \tfrac{\gamma_p}{3n} \geq \tfrac{\gamma_p}{4n}) \\
&\leq f(\theta_t) - \frac{\eta\,\gamma_p}{8n} && \text{(Setting } \eta \leq \bar{\eta}_1 := \tfrac{\gamma_p}{8\,H_1\,f(\theta_1)\,n})
\end{aligned}
$$

Since $f(\theta_{t+1}) \leq f(\theta_t) \leq f(\theta_1)$, this completes the induction for Phase 1. By recursing for $T_0 - 1$ iterations,

$$\implies f(\theta_{T_0}) \leq f(\theta_1) - \frac{\eta\,\gamma_p}{8n}(T_0 - 1) = \ln(2) - \frac{\eta\,\gamma_p}{8n}(T_0 - 1) \qquad \text{(Since } \theta_1 = 0)$$

Hence, by using that $\eta \leq \min\{\bar{\eta}_0, \bar{\eta}_1\}$,

$$T_0 \geq O\left(\frac{n\,\ln(2)}{\eta\,\gamma_p}\right) = O\left(\left(\frac{n}{\gamma_p}\right)^2 H_1\right) \text{ iterations}$$

guarantee that $f(\theta_{T_0}) \leq \frac{\ln(2)}{n}$.

**Phase 2:** After $T_0$ iterations, $f(\theta_{T_0}) \leq \frac{\ln(2)}{n} < f(\theta_1)$. We will now prove that for $t > T_0$, $f(\theta_t) \leq f(\theta_{T_0}) < f(\theta_1)$, thus completing the induction in Phase 2. We will again do this proof via induction.

**Base Case**: For $t = T_0$, this is true by definition.

**Inductive Hypothesis**: Assume for iteration $t > T_0$, $f(\theta_t) \leq f(\theta_{T_0})$.

**Induction**: We will prove that $f(\theta_{t+1}) \leq f(\theta_{T_0})$. Since $f(\theta_t) \leq f(\theta_{T_0}) < f(\theta_1)$ by the inductive hypothesis, combining the above relation with Eq. (32),

$$
\begin{aligned}
f(\theta_{t+1}) &\leq f(\theta_t) - \eta\,\|\nabla_t\|_q + H_1\, f(\theta_t)\, \eta^2 \\
&\leq f(\theta_t) - \frac{\eta\,\gamma_p}{2}f(\theta_t) + H_1\, f(\theta_t)\, \eta^2 && \text{(Using Part (1) of Prop. 2)} \\
&\leq f(\theta_t) - \frac{\eta\,\gamma_p}{4}f(\theta_t) && \text{(Setting } \eta \leq \bar{\eta}_2 := \tfrac{\gamma_p}{4\,H_1})
\end{aligned}
$$

Since $f(\theta_{t+1}) < f(\theta_t) < f(\theta_{T_0})$, this completes the induction. Furthermore, by recursing for $T$ iterations

$$\implies f(\theta_{T_0+T}) \leq f(\theta_{T_0})\exp\left(-\frac{\eta\,\gamma_p}{4}T\right) \leq \frac{\ln(2)}{n}\exp\left(-\frac{\eta\,\gamma_p}{4}T\right)$$

Hence, by using a step-size $\eta \leq \min\{\bar{\eta}_0, \bar{\eta}_2\}$,

$$T \geq O\left(\frac{1}{\eta\,\gamma_p}\ln\left(\frac{\ln(2)}{n\epsilon}\right)\right) = O\left(\frac{H_1}{\gamma_p^2}\ln\left(\frac{1}{n\epsilon}\right)\right)$$

iterations suffice to guarantee that $f(\theta_{T+T_0}) \leq \epsilon$. Hence, attaining an $\epsilon$ sub-optimality for logistic regression requires,

$$T_{\text{final}} = O\left(\frac{H_1}{\gamma_p^2} \ln\left(\frac{1}{n\epsilon}\right)\right) + O\left(\left(\frac{n}{\gamma_p}\right)^2 H_1\right) = O\left(\frac{H_1}{\gamma_p^2}\left[n^2 + \ln\left(\frac{1}{n\epsilon}\right)\right]\right) \text{ iterations.}$$

$\square$

**Proposition 8.** *Consider minimizing a one-dimensional quadratic $f(\theta) = \frac{\theta^2}{2}$ using* NSD *with a constant step-size $\eta$. For a fixed $\eta$ and $\epsilon \in \left(0, \frac{1}{8}\right)$, there exists an initialization $\theta_1$ such that the hitting time $T_\epsilon := \inf\{t \geq 1 : f(x_t) \leq \epsilon\}$ is bounded as $T_\epsilon = \Omega\left(\frac{1}{\sqrt{\epsilon}}\right)$.*

*Proof.* We first prove for a fixed $\eta > \sqrt{8\epsilon}$, we can find an initialization $\theta_1$ such that the NSD iterates oscillate around $\theta^* = 0$ and that $f(\theta_t) > \epsilon$ for all $t \geq 1$.

For this, consider the NSD update at $t = 1$ with $\theta_1 = \frac{\eta}{2}$. In this case,

$$d_1 = \arg\max_{|d| \leq 1} d\, f'(\theta_1) = \arg\max_{|d| \leq 1} d\,\theta_1 = 1 \implies \theta_2 = \theta_1 - \eta\, d_1 = \frac{\eta}{2} - \eta = -\frac{\eta}{2}$$

Similarly, for $t = 2$, $d_2 = -1 \implies \theta_3 = \frac{\eta}{2}$. Hence, the iterates oscillate between $\eta/2$ and $-\eta/2$. For all $t \geq 1$, $f(\theta_t) = \frac{\theta_t^2}{2} = \frac{\eta^2}{8} > \epsilon$.

Hence, ensuring $\epsilon$ convergence from any initialization requires $\eta \leq \sqrt{8\epsilon}$. In this case, choose the initialization $\theta_1 = 1$ and define $T_{\text{sign}} := \inf_{t \geq 1} \theta_t < 0$. Using the NSD update, for all $t < T_{\text{sign}}$, $\theta_t > 0$ and,

$$\theta_{t+1} = \theta_t - \eta \implies \theta_{T_{\text{sign}}} = 1 - \eta\,(T_{\text{sign}} - 1) \implies T_{\text{sign}} > 1 + \frac{1}{\eta} \geq 1 + \frac{1}{\sqrt{8\epsilon}}$$

**Case 1**: If $T_\epsilon \geq T_{\text{sign}}$, then $T_\epsilon \geq T_{\text{sign}} > 1 + \frac{1}{\eta} \geq 1 + \frac{1}{\sqrt{8\epsilon}}$, which gives the desired lower bound.

**Case 2**: If $T_\epsilon < T_{\text{sign}}$, then for all $t < T_\epsilon < T_{\text{sign}}$, $f(\theta_t) = \frac{(1 - \eta\,(t-1))^2}{2}$. By definition of $T_\epsilon$,

$$\frac{(1 - \eta\,(T_\epsilon - 1))^2}{2} \leq \epsilon \implies T_\epsilon \geq 1 + \frac{1 - \sqrt{2\epsilon}}{\eta} \geq 1 + \frac{1 - \sqrt{2\epsilon}}{\sqrt{8\epsilon}}$$

Hence, the hitting time $T_\epsilon = \Omega\left(\frac{1}{\sqrt{\epsilon}}\right)$. $\square$

### D.1. Helper Lemmas

**Lemma 13.** *For constants $A, B > 0$, if for all iterations $t \geq 1$,*

$$f(\theta_{t+1}) \leq f(\theta_t) - \eta\, A\, f(\theta_t) + \eta^2\, B\, f(\theta_t),$$

*and requires $\eta \leq \bar\eta_1$, then, with $\eta = \min\{\bar\eta_1, \frac{A}{2B}\}$, after $T = \frac{2\,\max\left\{\frac{1}{\bar\eta_1}, \frac{2B}{A}\right\}}{A} \ln\left(\frac{f(\theta_1)}{\delta}\right)$ iterations,*

$$f(\theta_{T+1}) \leq \delta \text{ and } \forall t \in [T],\, f(\theta_{t+1}) \leq f(\theta_t) \tag{33}$$

*Proof.* Setting $\eta \leq \frac{A}{2B}$,

$$f(\theta_{t+1}) \leq f(\theta_t) - \frac{A\eta}{2}\, f(\theta_t) \implies f(\theta_{t+1}) \leq f(\theta_t)\left(1 - \frac{A\eta}{2}\right)$$

$$\implies f(\theta_{T+1}) \leq f(\theta_1) \exp\left(-T\,\frac{A\eta}{2}\right) \qquad \text{(By recursing from } t = 1 \text{ to } T\text{)}$$

Hence, in order to guarantee that $f(\theta_{T+1}) \leq \delta$, it is sufficient to set,

$$T \geq \frac{2}{A\eta} \ln\left(\frac{f(\theta_1)}{\delta}\right) = \frac{2 \max\left\{\frac{1}{\bar{\eta}_1}, \frac{2B}{A}\right\}}{A} \ln\left(\frac{f(\theta_1)}{\delta}\right) = O\left(\ln\left(\frac{1}{\delta}\right)\right)$$

$\square$

**Lemma 14.** *For constants $A, B > 0$, and $r > 0$, if for all iterations $t \geq 1$,*

$$f(\theta_{t+1}) \leq f(\theta_t) - \eta A [f(\theta_t)]^r + \eta^2 B$$

*and requires $\eta \leq \bar{\eta}_1$, then, with $\eta = \min\{\bar{\eta}_1, \frac{A\delta^r}{2B}\}$*

***Case 1:*** *If $r = 1$, after $T \geq \frac{2 \max\left\{\frac{1}{\bar{\eta}_1}, \frac{2B}{A\delta}\right\}}{A} \ln\left(\frac{f(\theta_1)}{\delta}\right) + 1$ iterations,*

***Case 2:*** *If $r < 1$, after $T \geq \frac{2 [f(\theta_1)]^{1-r} \max\left\{\frac{1}{\bar{\eta}_1}, \frac{2B}{A\delta^r}\right\}}{A(1-r)} + 1$ iterations,*

***Case 3:*** *If $r > 1$, after $T \geq \frac{2 \max\left\{\frac{1}{\bar{\eta}_1}, \frac{2B}{A\delta^r}\right\}}{A(r-1)\delta^{r-1}} + 1$ iterations,*

$$f(\theta_{T+1}) \leq \delta \text{ and } \forall t \in [T], f(\theta_{t+1}) \leq f(\theta_t) \tag{34}$$

*Proof.* For the desired sub-optimality $\delta$, setting $\eta = \min\left\{\bar{\eta}_1, \frac{A\delta^r}{2B}\right\}$,

$$f(\theta_{t+1}) \leq f(\theta_t) - \eta A [f(\theta_t)]^r + \frac{A\eta}{2}\delta^r$$

Let $T_\delta$ be the first iteration s.t. $f(\theta_t) \leq \delta$. Hence, for all $t < T_\delta$, $f(\theta_t) \geq \delta \implies [f(\theta_t)]^r \geq \delta^r$. Hence, for $t < T_\delta$,

$$f(\theta_{t+1}) \leq f(\theta_t) - \frac{A\eta}{2}[f(\theta_t)]^r \implies f(\theta_{t+1}) \leq f(\theta_t)\left(1 - \frac{A\eta}{2}\right) \tag{35}$$

**Case 1**: If $r = 1$,

$$f(\theta_{t+1}) \leq f(\theta_t)\left[1 - \frac{A\eta}{2}\right]$$

$$\implies f(\theta_{T_\delta}) \leq f(\theta_1) \exp\left(-(T_\delta - 1)\frac{A\eta}{2}\right)$$

For $f(\theta_{T_\delta}) \leq \delta$, it is sufficient to set $T_\delta$ s.t.,

$$T_\delta \geq \frac{2}{A\eta} \ln\left(\frac{f(\theta_1)}{\delta}\right) + 1 = \frac{2 \max\left\{\frac{1}{\bar{\eta}_1}, \frac{2B}{A\delta}\right\}}{A} \ln\left(\frac{f(\theta_1)}{\delta}\right) + 1 = O\left(\frac{1}{\delta} \ln\left(\frac{1}{\delta}\right)\right)$$

**Case 2:** If $r < 1$, continuing from Eq. (35), note that $g(x) = x^{1-r}$ is concave in $x$ if $r < 1$. Furthermore, $g'(x) = (1-r)x^{-r}$. Using concavity, for any $y, x$,

$$g(y) \leq g(x) + g'(x)(y - x) \implies y^{1-r} \leq x^{1-r} + (1-r)x^{-r}(y - x)$$

$$\implies [f(\theta_{t+1})]^{1-r} \leq [f(\theta_t)]^{1-r} + (1-r)[f(\theta_t)]^{-r}(f(\theta_{t+1}) - f(\theta_t))$$

$$\leq [f(\theta_t)]^{1-r} - (1-r)[f(\theta_t)]^{-r}\frac{A\eta}{2}[f(\theta_t)]^r \qquad \text{(Using Eq. (35))}$$

$$= [f(\theta_t)]^{1-r} - (1-r)\frac{A\eta}{2}$$

$$\implies [f(\theta_{T_\delta})]^{1-r} \leq [f(\theta_1)]^{1-r} - (1-r)\frac{A\eta}{2}(T_\delta - 1)$$

For $f(\theta_{T_\delta}) \leq \delta$, it is sufficient to set $T_\delta$ s.t.,

$$[f(\theta_1)]^{1-r} - (1-r)\frac{A\eta}{2}(T_\delta - 1) \leq \delta^{1-r}$$

Hence, it is sufficient to set $T_\delta$ s.t.

$$T_\delta \geq \frac{2\,[f(\theta_1)]^{1-r}}{A\,\eta\,(1-r)} + 1 = \frac{2\,[f(\theta_1)]^{1-r}\,\max\left\{\frac{1}{\eta_1}, \frac{2\,B}{A\delta^r}\right\}}{A(1-r)} + 1 = O\left(\frac{1}{\delta^r}\right)$$

**Case 3:** If $r > 1$, continuing from Eq. (35), note that $g(x) = x^{1-r}$ is convex in $x$ if $r > 1$. Furthermore, $g'(x) = (1-r)\,x^{-r}$. Using convexity, for any $y, x$,

$$g(y) \geq g(x) + g'(x)(y-x) \implies y^{1-r} \geq x^{1-r} + (1-r)\,x^{-r}\,(y-x)$$

$$\implies [f(\theta_{t+1})]^{1-r} \geq [f(\theta_t)]^{1-r} + (1-r)\,[f(\theta_t)]^{-r}\,(f(\theta_{t+1}) - f(\theta_t))$$

$$\geq [f(\theta_t)]^{1-r} - (1-r)\,[f(\theta_t)]^{-r}\,\frac{A\eta}{2}\,[f(\theta_t)]^r \qquad \text{(Using Eq. (35))}$$

$$= [f(\theta_t)]^{1-r} - (1-r)\,\frac{A\eta}{2}$$

$$\implies [f(\theta_{T_\delta})]^{1-r} \geq [f(\theta_1)]^{1-r} - (1-r)\,\frac{A\eta}{2}\,(T_\delta - 1) = [f(\theta_1)]^{1-r} + (r-1)\,\frac{A\eta}{2}\,(T_\delta - 1)$$

$$\implies [f(\theta_{T_\delta})]^{r-1} \leq \frac{1}{[f(\theta_1)]^{1-r} + (r-1)\,\frac{A\eta}{2}\,(T_\delta - 1)}$$

For $f(\theta_{T_\delta}) \leq \delta$, it is sufficient to set $T_\delta$ s.t.,

$$\frac{1}{[f(\theta_1)]^{1-r} + (r-1)\,\frac{A\eta}{2}\,(T_\delta - 1)} \leq \delta^{r-1} \implies [f(\theta_1)]^{1-r} + (r-1)\,\frac{A\eta}{2}\,(T_\delta - 1) \geq \delta^{1-r}$$

Hence, it is sufficient to set $T_\delta$ s.t.

$$T_\delta \geq \frac{2\,\delta^{1-r}}{A\,\eta\,(r-1)} + 1 = \frac{2\,\max\left\{\frac{1}{\eta_1}, \frac{2\,B}{A\delta^r}\right\}\,\delta^{1-r}}{A(r-1)} + 1 = O\left(\frac{1}{\delta^{2\,r-1}}\right)$$

$\square$

**Lemma 15.** *Under Assn. 1, Assn. 2 with $H_0 \geq 0$ and $H_1 \geq 0$, for the normalized steepest descent update in Eq. (14), with $\eta \leq \frac{1}{\sqrt{H_1}}$,*

$$f(\theta_{t+1}) \leq f(\theta_t) - \eta\,\|\nabla f(\theta_t)\|_q + (H_0 + H_1\,f(\theta_t))\,\eta^2$$

*Proof.* If $\|\theta_{t+1} - \theta_t\|_p \leq \frac{1}{\sqrt{H_1}}$, using Lemma. 7 and denoting $\nabla_t := \nabla f(\theta_t)$ for convenience,

$$f(\theta_{t+1}) \leq f(\theta_t) - \eta\,\langle \nabla f(\theta_t), d_t \rangle + (H_0 + H_1\,f(\theta_t))\,\eta^2\,\|d_t\|_p^2$$

$$\leq f(\theta_t) - \eta\,\langle \nabla f(\theta_t), d_t \rangle + (H_0 + H_1\,f(\theta_t))\,\eta^2 \qquad \text{(By definition } \|d_t\|_p \leq 1)$$

$$\implies f(\theta_{t+1}) \leq f(\theta_t) - \eta\,\|\nabla f(\theta_t)\|_q + (H_0 + H_1\,f(\theta_t))\,\eta^2$$

$$\text{(By definition of } d_t \text{ and since } \|z\|_q := \max_{\|u\|_p \leq 1}\langle z, u\rangle)$$

$\square$

# E. Convergence of RMSProp

**Theorem 3.** *Under Assn. 1, Assn. 2 and Assn. 3 with $f^* = 0$, $\mu(\theta) = \mu$ for all $\theta$, `RMSProp` with $\delta = 0$ converges as:*

• *If $\epsilon > \frac{H_0}{H_1}$, using $\eta_t = \eta = O(1)$ guarantees that after $T = O\left(\ln\left(\frac{1}{\epsilon}\right)\right)$ iterations, $f(\theta_{T+1}) \leq \epsilon$.*

• *Else, using $\eta_t = \eta = O(\epsilon^{2\tau})$ guarantees that $f(\theta_{T+1}) \leq \epsilon$ after $T$ iterations, where,*

    • *If $\tau \leq \frac{1}{2}$, $T = O\left(\frac{1}{\epsilon^{2\tau}}\right)$.*

    •*Else, if $\tau > \frac{1}{2}$, $T = O\left(\frac{1}{\epsilon^{4\tau-1}}\right)$.*

*Proof.* Using Lemma. 16 for $\eta \leq \frac{\sqrt{1-\beta}}{\sqrt{H_1}}$, $\|d_t\|_\infty \leq \frac{1}{\sqrt{1-\beta}}$ and,

$$f(\theta_{t+1}) \leq f(\theta_t) - \eta \langle \nabla_t, d_t \rangle + (H_0 + H_1 f(\theta_t)) \frac{\eta^2}{1-\beta} \tag{36}$$

Using Lemma. 17 to simplify the second term on the RHS of Eq. (36),

$$\langle \nabla_t, d_t \rangle = \sum_i [\nabla_t]_i [d_t]_i = \sum_i \frac{g_{t,i}^2}{\sqrt{v_{t,i}}} \geq \|\nabla_t\|_1 \frac{\|\nabla_t\|_1}{\sqrt{1-\beta} \sum_{j=0}^{t-1} (\sqrt{\beta})^j \|\nabla_{t-j}\|_1} \tag{37}$$

Combining the above inequalities,

$$f(\theta_{t+1}) \leq f(\theta_t) - \eta \|\nabla_t\|_1 \frac{\|\nabla_t\|_1}{\sqrt{1-\beta} \sum_{j=0}^{t-1} (\sqrt{\beta})^j \|\nabla_{t-j}\|_1} + (H_0 + H_1 f(\theta_t)) \frac{\eta^2}{1-\beta} \tag{38}$$

**Phase 1:** Define $T_0$ to be the first iteration s.t. $f(\theta_{T_0}) < \max\left\{\epsilon, \frac{H_0}{H_1}\right\}$, and analyze the above inequality for $t < T_0$ where $f(\theta_t) \geq \max\left\{\epsilon, \frac{H_0}{H_1}\right\}$. Using Eq. (38) in this case,

$$f(\theta_{t+1}) \leq f(\theta_t) - \eta \|\nabla_t\|_1 \underbrace{\frac{\|\nabla_t\|_1}{\sqrt{1-\beta} \sum_{j=0}^{t-1} (\sqrt{\beta})^j \|\nabla_{t-j}\|_1}}_{(*)} + 2 H_1 f(\theta_t) \frac{\eta^2}{1-\beta} \tag{39}$$

We will now use an inductive proof and prove that for all $t \leq T_0$, $f(\theta_t) \leq f(\theta_1)$.

**Base Case**: For $t = 1$, this is true by definition.

**Inductive Hypothesis**: Assume for iteration $t < T_0$, $f(\theta_t) \leq f(\theta_1)$.

**Induction**: We will prove that $f(\theta_{t+1}) \leq f(\theta_1)$. First, for $\eta \leq \frac{\mu\sqrt{1-\beta}}{e} \left(\frac{H_0}{H_1}\right)^\tau \frac{1}{H_0 + H_1 f(\theta_1)} \ln(1/\beta^{1/4})$ and all $t$,

$$\|\theta_{t+1} - \theta_t\|_\infty = |\eta_t| \underbrace{\max_i |\frac{g_{t,i}}{\sqrt{v_{t,i}}}|}_{\leq \frac{1}{\sqrt{1-\beta}}} \leq \mu \underbrace{\left(\frac{H_0}{H_1}\right)^\tau}_{c} \frac{1}{e(H_0 + H_1 f(\theta_1))} \underbrace{\ln(1/\beta^{1/4})}_{a}.$$

Since $f(\theta_t) \leq f(\theta_1)$ by the inductive hypothesis, we can use Lemma. 18 and obtain that,

$$(*) = \frac{\|\nabla_t\|_1}{\sqrt{1-\beta} \sum_{j=0}^{t-1} (\sqrt{\beta})^j \|\nabla_{t-j}\|_1} \geq \underbrace{\frac{1 - \beta^{1/4}}{\sqrt{1-\beta}}}_{:=C} \frac{1}{1 + \left(\frac{H_0}{H_1 f(\theta_t)}\right)^\tau} = \frac{C}{1 + \left(\frac{H_0}{H_1 f(\theta_t)}\right)^\tau} \tag{40}$$

Combining Eq. (39) and Eq. (40) and noting that in Phase (1), $\frac{H_0}{H_1 f(\theta_t)} \leq 1 \implies (*) \geq \frac{C}{2}$,

$$f(\theta_{t+1}) \leq f(\theta_t) - \eta \|\nabla_t\|_1 \frac{C}{2} + 2 H_1 f(\theta_t) \frac{\eta^2}{1-\beta}$$

$$\leq f(\theta_t) - \eta \, [f(\theta_t)]^\tau \, \frac{C \, \mu}{2} + 2 \, H_1 \, f(\theta_t) \, \frac{\eta^2}{1 - \beta} \qquad \text{(Using Assn. 3 with } f^* = 0 \text{ and } \mu(\theta) = \mu)$$

$$\leq f(\theta_t) - \eta \, \frac{f(\theta_t)}{[f(\theta_1)]^{1-\tau}} \, \frac{C \, \mu}{2} + 2 \, H_1 \, f(\theta_t) \, \frac{\eta^2}{1 - \beta} \qquad \text{(Since } f(\theta_t) \leq f(\theta_1) \text{ by the induction hypothesis)}$$

Using Lemma. 13 with $A = \frac{C}{2} \, \frac{\mu}{[f(\theta_1)]^{1-\tau}}$, $B = \frac{2H_1}{1-\beta}$, $\bar{\eta}_1 = \min \left\{ \frac{\sqrt{1-\beta}}{\sqrt{H_1}}, \frac{\mu \sqrt{1-\beta}}{2e} \left( \frac{H_0}{H_1} \right)^\tau \frac{1}{H_0 + H_1 \, f(\theta_1)} \ln(1/\beta^{1/4}) \right\}$ and $\delta = \max \left\{ \epsilon, \frac{H_0}{H_1} \right\}$, we get that with $\eta = \min \left\{ \bar{\eta}_1, \frac{C \, \mu \, (1-\beta)}{8 \, H_1 \, [f(\theta_1)]^{1-\tau}} \right\}$, $f(\theta_{t+1}) \leq f(\theta_t)$. This completes the induction. Moreover, $f(\theta_{T_0}) \leq \max \left\{ \epsilon, \frac{H_0}{H_1} \right\}$ after

$$T_0 \geq O \left( \frac{[f(\theta_1)]^{1-\tau}}{C \, \mu \, \eta} \ln \left( \frac{f(\theta_1)}{\delta} \right) \right) = O \left( \ln \left( \min \left\{ \frac{1}{\epsilon}, \frac{H_1}{H_0} \right\} \right) \right) \quad \text{iterations.}$$

Hence, if $\epsilon \geq \frac{H_0}{H_1}$, RMSProp with a constant step-size requires $O(\ln(1/\epsilon))$ iterations to guarantee convergence to an $\epsilon$ sub-optimality.

**Phase 2**: Consider iteration $t > T_0$ such that $f(\theta_t) < \frac{H_0}{H_1}$. Simplifying Eq. (38) in this case,

$$f(\theta_{t+1}) \leq f(\theta_t) - \eta \, \|\nabla_t\|_1 \underbrace{\frac{\|\nabla_t\|_1}{\sqrt{1 - \beta} \sum_{j=0}^{t-1} (\sqrt{\beta})^j \, \|\nabla_{t-j}\|_1}}_{(*)} + 2 \, H_0 \, \frac{\eta^2}{1 - \beta} \tag{41}$$

We will now use an inductive proof and prove that for all $t \geq T_0$, $f(\theta_t) \leq f(\theta_{T_0}) \leq \frac{H_0}{H_1}$.

**Base Case**: For $t = T_0$, this is true by definition.

**Induction**: We will prove that $f(\theta_{t+1}) \leq f(\theta_{T_0})$. Since $f(\theta_t) \leq f(\theta_{T_0}) \leq \frac{H_0}{H_1} \leq f(\theta_1)$ by the inductive hypothesis, and ensuring that $\eta \leq \frac{\mu \sqrt{1-\beta}}{2e} \left( \frac{H_0}{H_1} \right)^\tau \frac{1}{H_0 + H_1 \, f(\theta_1)} \ln(1/\beta^{1/4})$, we can use Lemma. 18 and obtain that,

$$(*) = \frac{\|\nabla_t\|_1}{\sqrt{1 - \beta} \sum_{j=0}^{t-1} (\sqrt{\beta})^j \, \|\nabla_{t-j}\|_1} \geq \underbrace{\frac{1 - \beta^{1/4}}{\sqrt{1 - \beta}}}_{:=C} \frac{1}{1 + \left( \frac{H_0}{H_1 \, f(\theta_t)} \right)^\tau} = \frac{C}{1 + \left( \frac{H_0}{H_1 \, f(\theta_t)} \right)^\tau} \tag{42}$$

Combining Eq. (41) and Eq. (42) and noting that, $\frac{H_0}{H_1 \, f(\theta_t)} \geq 1 \implies (*) \geq \frac{C}{2 \left( \frac{H_0}{H_1 \, f(\theta_t)} \right)^\tau}$,

$$f(\theta_{t+1}) \leq f(\theta_t) - \eta \, \|\nabla_t\|_1 \, \frac{C}{2} \left( \frac{H_1}{H_0} \right)^\tau [f(\theta)]^\tau + 2 \, H_0 \, \frac{\eta^2}{1 - \beta}$$

$$\leq f(\theta_t) - \eta \, \frac{C \, \mu}{2} \left( \frac{H_1}{H_0} \right)^\tau [f(\theta_t)]^{2\tau} + 2 \, H_0 \, \frac{\eta^2}{1 - \beta} \qquad \text{(Using Assn. 3 with } f^* = 0 \text{ and } \mu(\theta) = \mu)$$

Using Lemma. 14 with $A = \frac{C \, \mu}{2} \left( \frac{H_1}{H_0} \right)^\tau$, $B = \frac{2 \, H_0}{1 - \beta}$, $\bar{\eta}_1 = \min \left\{ \frac{\sqrt{1-\beta}}{\sqrt{H_1}}, \frac{\mu \sqrt{1-\beta}}{2e} \left( \frac{H_1}{H_0} \right)^\tau \frac{1}{H_0 + H_1 \, f(\theta_1)} \ln(1/\beta^{1/4}) \right\}$, $r = 2\tau$, $\delta = \epsilon$, we get that with $\eta = \min \left\{ \bar{\eta}_1, \frac{C \, \mu \, (1-\beta) \epsilon^{2\tau}}{8 \, H_0} \left( \frac{H_1}{H_0} \right)^\tau \right\}$, $f(\theta_{t+1}) \leq f(\theta_t) \leq f(\theta_{T_0}) \leq \frac{H_0}{H_1}$ for all $t \geq T_0$. This completes the induction.

Furthermore, for $\tau = \frac{1}{2}$, $f(\theta_{T_\epsilon + T_0}) \leq \epsilon$ after

$$T_\epsilon \geq O \left( \frac{1}{C \, \mu \, \eta} \sqrt{\frac{H_0}{H_1}} \ln \left( \frac{f(\theta_1)}{\delta} \right) \right) = \tilde{O} \left( \frac{1}{\epsilon} \right) \quad \text{iterations.}$$

Else, for $\tau < \frac{1}{2}$, $f(\theta_{T_\epsilon + T_0}) \leq \epsilon$ after

$$T_\epsilon \geq O \left( \frac{[f(\theta_{T_0})]^{1-2\tau}}{C \, \mu \, \eta \, (1 - 2\tau)} \left( \frac{H_0}{H_1} \right)^\tau \right) = O \left( \frac{1}{\epsilon^{2\tau}} \right) \quad \text{iterations.}$$

Else, for $\tau > \frac{1}{2}$, $f(\theta_{T_\epsilon + T_0}) \leq \epsilon$ after

$$T_\epsilon \geq O\left(\frac{1}{C\,\mu\,\eta\,(2\,\tau-1)\,\epsilon^{2\tau-1}}\left(\frac{H_0}{H_1}\right)^\tau\right) = O\left(\frac{1}{\epsilon^{4\tau-1}}\right) \text{ iterations.}$$

Hence, for $T := T + T_\epsilon$, $f(\theta_{T+1}) \leq \epsilon$ after,

$$O\left(\frac{1}{\epsilon^{2\tau}} + \ln\left(\frac{H_1}{H_0}\right)\right) \text{ iterations, if } \tau \leq \frac{1}{2}\ ;$$

$$O\left(\frac{1}{\epsilon^{4\tau-1}} + \ln\left(\frac{H_1}{H_0}\right)\right) \text{ iterations, if } \tau > \frac{1}{2}$$

$\square$

### E.1. Helper Lemmas

**Lemma 16.** *Under Assn. 1, Assn. 2 with $H_0 \geq 0$ and $H_1 \geq 0$, for the coordinate-wise RMSProp update in Eq. (16), with $\eta \leq \frac{\sqrt{1-\beta}}{\sqrt{H_1}}$,*

$$f(\theta_{t+1}) \leq f(\theta_t) - \eta\,\langle\nabla_t, d_t\rangle + (H_0 + H_1\,f(\theta_t))\,\frac{\eta^2}{1-\beta}\quad ;\quad \|d_t\|_\infty \leq \frac{1}{\sqrt{1-\beta}}$$

*Proof.* If $\|\theta_{t+1} - \theta_t\|_\infty \leq \frac{1}{\sqrt{H_1}}$, using Lemma. 7 with $p = \infty$ and $q = 1$, and denoting $\nabla_t := \nabla f(\theta_t)$ for convenience,

$$f(\theta_{t+1}) \leq f(\theta_t) - \eta\,\langle\nabla_t, d_t\rangle + (H_0 + H_1\,f(\theta_t))\,\eta^2\,\|d_t\|_\infty^2$$

Simplifying the third term on the RHS,

$$\|d_t\|_\infty = \max_i \frac{g_{t,i}}{\sqrt{v_{t,i}}} \leq \frac{1}{\sqrt{1-\beta}} \qquad \text{(Since } v_{t,i} \geq (1-\beta)\,g_{t,i}^2\text{)}$$

Ensuring that $\eta \leq \frac{\sqrt{1-\beta}}{\sqrt{H_1}}$ guarantees that $\|\theta_{t+1} - \theta_t\|_\infty \leq \frac{1}{\sqrt{H_1}}$. Combining the above inequalities,

$$f(\theta_{t+1}) \leq f(\theta_t) - \eta\,\langle\nabla_t, d_t\rangle + (H_0 + H_1\,f(\theta_t))\,\frac{\eta^2}{1-\beta} \tag{43}$$

$\square$

**Lemma 17.** *For $\beta \in (0,1)$, if $g_{t,i} = [\nabla f(\theta_t)]_i$ and $v_{t,i} = (1-\beta)\sum_{s=1}^t \beta^{t-s}\,g_{s,i}^2 = \beta\,v_{t-1,i} + (1-\beta)\,g_{t,i}^2$ with $[v_0]_i = 0$, then,*

$$\sum_i \frac{g_{t,i}^2}{\sqrt{v_{t,i}}} \geq \|\nabla_t\|_1\,\frac{\|\nabla_t\|_1}{\sqrt{1-\beta}\,\sum_{j=0}^{t-1}(\sqrt{\beta})^j\,\|\nabla_{t-j}\|_1}$$

*Proof.*

$$\|\nabla_t\|_1 = \sum_i |g_{t,i}| = \sum_i \frac{|g_{t,i}|}{v_{t,i}^{1/4}}\,v_{t,i}^{1/4} \leq \sqrt{\sum_i \frac{g_{t,i}^2}{\sqrt{v_{t,i}}}}\sqrt{\sum_i \sqrt{v_{t,i}}} \qquad \text{(By Cauchy Schwarz)}$$

$$\implies \|\nabla_t\|_1^2 \leq \left(\sum_i \frac{g_{t,i}^2}{\sqrt{v_{t,i}}}\right)\left(\sum_i \sqrt{v_{t,i}}\right) \implies \left(\sum_i \frac{g_{t,i}^2}{\sqrt{v_{t,i}}}\right) \geq \frac{\|\nabla_t\|_1^2}{\underbrace{\sum_i \sqrt{v_{t,i}}}_{:=u_t}} \tag{44}$$

Simplifying $u_t$,

$$u_t = \sum_i \sqrt{v_{t,i}} = \sum_i \sqrt{\beta\, v_{t-1,i} + (1-\beta)\, g_{t,i}^2} \leq \sum_i \sqrt{\beta}\, \sqrt{v_{t-1,i}} + \sqrt{1-\beta}\, |g_{t,i}| \quad \text{(Since } \sqrt{a+b} \leq \sqrt{a} + \sqrt{b}\text{)}$$

$$= \sqrt{\beta} \sum_i \sqrt{v_{t-1,i}} + \sqrt{1-\beta}\, \|\nabla_t\|_1 = \sqrt{\beta}\, u_{t-1} + \sqrt{1-\beta}\, \|\nabla_t\|_1 \tag{45}$$

Recursing from $j = 0$ to $t-1$, and using that $v_{0,i} = 0$,

$$\implies u_t \leq \sqrt{1-\beta} \sum_{j=0}^{t-1} (\sqrt{\beta})^j\, \|\nabla_{t-j}\|_1 \tag{46}$$

$$\implies \left( \sum_i \frac{g_{t,i}^2}{\sqrt{v_{t,i}}} \right) \geq \|\nabla_t\|_1\, \frac{\|\nabla_t\|_1}{\sqrt{1-\beta} \sum_{j=0}^{t-1} (\sqrt{\beta})^j\, \|\nabla_{t-j}\|_1} \qquad \text{(Combining with Eq. (44))}$$

$\square$

**Lemma 18.** *Under Assn. 1, Assn. 2 with $H_0 \geq 0, H_1 > 0$, fix constants $c > 0$ and $a > 0$ such that $\sqrt{\beta} \exp(a) < 1$. If for all iterations $t \geq 1$, (i) $\|\theta_t - \theta_{t+1}\|_p \leq \frac{c}{e} \frac{a}{H_0 + H_1 f(\theta_1)}$ such that and (ii) $f(\theta_t) \leq f(\theta_1)$, then,*

$$\frac{\|\nabla_t\|_1}{\sqrt{1-\beta} \sum_{j=0}^{t-1} (\sqrt{\beta})^j\, \|\nabla_{t-j}\|_1} \geq \frac{\|\nabla_t\|_1}{\|\nabla_t\|_1 + c}\, \frac{1 - \sqrt{\beta}\, \exp(a)}{\sqrt{1-\beta}}$$

*Furthermore, if $f$ satisfies Assn. 3 with $f^* = 0$ and $\mu(\theta) = \mu$ for all $\theta$, then, by choosing $c = \mu \left( \frac{H_0}{H_1} \right)^\tau > 0$,*

$$\frac{\|\nabla_t\|_1}{\sqrt{1-\beta} \sum_{j=0}^{t-1} (\sqrt{\beta})^j\, \|\nabla_{t-j}\|_1} \geq \frac{1 - \sqrt{\beta}\, \exp(a)}{\sqrt{1-\beta}}\, \frac{1}{1 + \left( \frac{H_0}{H_1} \frac{1}{f(\theta_t)} \right)^\tau}$$

*Proof.* Using part (1) of Lemma. 10 with $p = \infty$, $q = 1$ and $y = \theta_{t-1}$, $x = \theta_t$, $c > 0$,

$$\|\nabla_{t-1}\|_1 + c \leq (\|\nabla_t\|_1 + c) \exp\left( (H_0 + H_1 f(\theta_t))\, e\, \frac{\|\theta_{t-1} - \theta_t\|_\infty}{c} \right)$$

$$\leq (\|\nabla_t\|_1 + c) \exp\left( (H_0 + H_1 f(\theta_1))\, e\, \frac{\|\theta_{t-1} - \theta_t\|_\infty}{c} \right) \qquad \text{(Using assumption (ii))}$$

$$\leq (\|\nabla_t\|_1 + c) \exp(a) \qquad \text{(Using assumption (i))}$$

$$\implies \|\nabla_{t-j}\|_1 \leq (\|\nabla_t\|_1 + c) \exp(a\, j) \qquad \text{(Recursing and since } c > 0\text{)}$$

Using the above relation to lower-bound $\sqrt{1-\beta} \sum_{j=0}^{t-1} (\sqrt{\beta})^j\, \|\nabla_{t-j}\|_1$, first note that,

$$\sqrt{1-\beta} \sum_{j=0}^{t-1} (\sqrt{\beta})^j\, \|\nabla_{t-j}\|_1 \leq \sqrt{1-\beta} \sum_{j=0}^{t-1} (\sqrt{\beta})^j (\|\nabla_t\|_1 + c) \exp(a\, j)$$

$$\leq \frac{\sqrt{1-\beta}\, (\|\nabla_t\|_1 + c)}{1 - \sqrt{\beta}\, \exp(a)} \qquad \text{(Since } \sqrt{\beta}\, \exp(a) < 1\text{)}$$

$$\implies \frac{\|\nabla_t\|_1}{\sqrt{1-\beta} \sum_{j=0}^{t-1} (\sqrt{\beta})^j\, \|\nabla_{t-j}\|_1} \geq \frac{\|\nabla_t\|_1}{\|\nabla_t\|_1 + c}\, \frac{1 - \sqrt{\beta}\, \exp(a)}{\sqrt{1-\beta}}$$

Furthermore, if $f$ satisfies Assn. 3 with $f^* = 0$ and $\mu(\theta) = \mu$, then,

$$\frac{\|\nabla_t\|_1}{\sqrt{1-\beta} \sum_{j=0}^{t-1} (\sqrt{\beta})^j\, \|\nabla_{t-j}\|_1} \geq \frac{1 - \sqrt{\beta}\, \exp(a)}{\sqrt{1-\beta}}\, \frac{1}{1 + \frac{c}{\mu\, [f(\theta_t)]^\tau}} \qquad \text{(Using Assn. 3 with } f^* = 0 \text{ and } \mu(\theta) = \mu\text{)}$$

$$\implies \frac{\|\nabla_t\|_1}{\sqrt{1-\beta}\sum_{j=0}^{t-1}(\sqrt{\beta})^j\|\nabla_{t-j}\|_1} \geq \frac{1-\sqrt{\beta}\exp(a)}{\sqrt{1-\beta}}\frac{1}{1+\left(\frac{H_0}{H_1}\frac{1}{f(\theta_t)}\right)^\tau} \qquad \text{(Setting } c = \mu\left(\frac{H_0}{H_1}\right)^\tau\text{)}$$

$\square$

## F. Convergence of Adam

**Theorem 4.** *Under Assn. 1, Assn. 2 with $H_0 \geq 0$ and $H_1 > 0$, and Assn. 3 with $f^* = 0$ and $\mu(\theta) = \mu$ for all $\theta$,* Adam *with the update in Eq. (18) with $\beta_1 \leq \beta_2$ has the following convergence rate:*

• *If $\epsilon > \frac{H_0}{H_1}$, using $\eta_t = \eta = O(1)$ guarantees that after $T = O\left(\ln\left(\frac{1}{\epsilon}\right)\right)$ iterations, $f(\theta_{T+1}) \leq \epsilon$.*

• *Else, using $\eta_t = \eta = O(\epsilon^{2\tau})$ guarantees that $f(\theta_{T+1}) \leq \epsilon$ after $T$ iterations, where,*

• *If $\tau \leq \frac{1}{2}$, $T = O\left(\frac{1}{\epsilon^{2\tau}}\right)$.*

•*Else, if $\tau > \frac{1}{2}$, $T = O\left(\frac{1}{\epsilon^{4\tau-1}}\right)$.*

*Proof.* Using Lemma. 19 for $\eta \leq \frac{1}{C_3\sqrt{H_1}}$ and $\beta_1 \leq \beta_2$ where $C_3 := \sqrt{\frac{1-\beta_1}{1-\beta_2}}$, and,

$$f(\theta_{t+1}) \leq f(\theta_t) - \eta\langle\nabla_t, d_t\rangle + (H_0 + H_1 f(\theta_t))\eta^2 C_3^2 \qquad (47)$$

Simplifying the second term on the RHS of Eq. (47), first note that,

$$\langle\nabla_t, d_t\rangle = \sum_i [\nabla_t]_i [d_t]_i = \sum_i \frac{g_{t,i} m_{t,i}}{\sqrt{v_{t,i}}} = \sum_i \frac{g_{t,i}(m_{t,i} - g_{t,i})}{\sqrt{v_{t,i}}} + \frac{g_{t,i}^2}{\sqrt{v_{t,i}}}$$

$$\geq -\sum_i \frac{|g_{t,i}|\,|m_{t,i} - g_{t,i}|}{\sqrt{v_{t,i}}} + \sum_i \frac{g_{t,i}^2}{\sqrt{v_{t,i}}}$$

$$\implies -\langle\nabla_t, d_t\rangle \leq \underbrace{\sum_i \frac{|g_{t,i}|\,|m_{t,i} - g_{t,i}|}{\sqrt{v_{t,i}}}}_{:=\text{Term (i)}} - \underbrace{\sum_i \frac{g_{t,i}^2}{\sqrt{v_{t,i}}}}_{:=\text{Term (ii)}} \qquad (48)$$

Bounding Term (i) using Lemma. 20 and using a constant step-size $\eta \leq \frac{1}{C_3\sqrt{H_1}}\ln(1/\sqrt{\beta_1})$, we get that,

$$\text{Term (i)} \leq \frac{\beta_1\,e\,C_3\sum_{s=1}^{t-1}(\sqrt{\beta_1})^{t-1-s}\eta_s}{\sqrt{1-\beta_2}}(H_0 + H_1 f(\theta_t)) \qquad (49)$$

$$\leq \eta\underbrace{\frac{\beta_1\,e\,C_3}{\sqrt{1-\beta_2}(1-\sqrt{\beta_1})}}_{:=C_1}(H_0 + H_1 f(\theta_t)) \qquad \text{(Since } \eta_s = \eta \text{ for all } s\text{)}$$

$$\implies \text{Term (i)} \leq \eta\,C_1(H_0 + H_1 f(\theta_t)) \qquad (50)$$

Similar to the proof of Thm. 3, we use Lemma. 17 to bound Term (ii),

$$\text{Term (ii)} \geq \|\nabla_t\|_1 \underbrace{\frac{\|\nabla_t\|_1}{\sqrt{1-\beta_2}\sum_{j=0}^{t-1}(\sqrt{\beta_2})^j\|\nabla_{t-j}\|_1}}_{(*)} \qquad (51)$$

**Phase 1:** Define $T_0$ to be the first iteration s.t. $f(\theta_{T_0}) < \max\left\{\epsilon, \frac{H_0}{H_1}\right\}$, and analyze Eq. (47) for $t < T_0$ where $f(\theta_t) \geq \max\left\{\epsilon, \frac{H_0}{H_1}\right\}$. Using Eqs. (50) and (51) and simplifying Eq. (47) in this case,

$$f(\theta_{t+1}) \leq f(\theta_t) - \eta\|\nabla_t\|_1 \underbrace{\frac{\|\nabla_t\|_1}{\sqrt{1-\beta_2}\sum_{j=0}^{t-1}(\sqrt{\beta_2})^j\|\nabla_{t-j}\|_1}}_{(*)} + 2H_1 f(\theta_t)\eta^2(C_1 + C_3^2) \qquad (52)$$

We will now use an inductive proof and prove that for all $t \leq T_0$, $f(\theta_t) \leq f(\theta_1)$.

**Base Case**: For $t = 1$, this is true by definition.

**Inductive Hypothesis**: Assume for iteration $t < T_0$, $f(\theta_t) \leq f(\theta_1)$.

**Induction**: We will prove that $f(\theta_{t+1}) \leq f(\theta_1)$. Since $f(\theta_t) \leq f(\theta_1)$ by the inductive hypothesis, and $\eta \leq \frac{\mu}{2eC_3} \left(\frac{H_0}{H_1}\right)^\tau \frac{1}{H_0 + H_1 f(\theta_1)} \ln(1/\beta_2^{1/4})$ so

$$\|\theta_{t+1} - \theta_t\|_\infty \leq \eta \, C_3 \leq \underbrace{\mu \left(\frac{H_0}{H_1}\right)^\tau}_{c} \frac{1}{e} \frac{1}{H_0 + H_1 \, f(\theta_1)} \underbrace{\ln(1/\beta_2^{1/4})}_{a},$$

we can use Lemma. 18 and obtain that,

$$(*) = \frac{\|\nabla_t\|_1}{\sqrt{1 - \beta_2} \sum_{j=0}^{t-1} (\sqrt{\beta_2})^j \, \|\nabla_{t-j}\|_1} \geq \underbrace{\frac{1 - \beta_2^{1/4}}{\sqrt{1 - \beta_2}}}_{:=C_2} \frac{1}{1 + \left(\frac{H_0}{H_1 \, f(\theta_t)}\right)^\tau} = \frac{C_2}{1 + \left(\frac{H_0}{H_1 \, f(\theta_t)}\right)^\tau} \tag{53}$$

Combining Eq. (52) and Eq. (53) and noting that in Phase (1), $\frac{H_0}{H_1 \, f(\theta_t)} \leq 1 \implies (*) \geq \frac{C_2}{2}$,

$$
\begin{aligned}
f(\theta_{t+1}) &\leq f(\theta_t) - \eta \, \|\nabla_t\|_1 \, \frac{C_2}{2} + 2 \, H_1 \, f(\theta_t) \, \eta^2 \, (C_1 + C_3^2) \\
&\leq f(\theta_t) - \eta \, [f(\theta_t)]^\tau \, \frac{C_2 \, \mu}{2} + 2 \, H_1 \, f(\theta_t) \, \eta^2 \, (C_1 + C_3^2) && \text{(Using Assn. 3 with } f^* = 0 \text{ and } \mu(\theta) = \mu) \\
&\leq f(\theta_t) - \eta \, \frac{f(\theta_t)}{[f(\theta_1)]^{1-\tau}} \, \frac{C_2 \, \mu}{2} + 2 \, H_1 \, f(\theta_t) \, \eta^2 \, (C_1 + C_3^2) && \text{(Since } f(\theta_t) \leq f(\theta_1) \text{ by the induction hypothesis)}
\end{aligned}
$$

Using Lemma. 13 with $A = \frac{C_2}{2} \frac{\mu}{[f(\theta_1)]^{1-\tau}}$, $B = 2 \, H_1 (C_1 + C_3^2)$, $\bar{\eta}_1 = \min \left\{ \frac{1}{C_3 \sqrt{H_1}}, \frac{1}{C_3 \sqrt{H_1}} \ln(1/\sqrt{\beta_1}), \frac{\mu}{2eC_3} \left(\frac{H_0}{H_1}\right)^\tau \frac{1}{H_0 + H_1 \, f(\theta_1)} \ln(1/\beta_2^{1/4}) \right\}$, $\delta = \max \left\{ \epsilon, \frac{H_0}{H_1} \right\}$, we get that with $\eta = \min \left\{ \bar{\eta}_1, \frac{C_2 \, \mu}{8 \, (C_1 + C_3^2) \, H_1 \, [f(\theta_1)]^{1-\tau}} \right\}$, $f(\theta_{t+1}) \leq f(\theta_t)$. This completes the induction. Moreover, $f(\theta_{T_0}) \leq \max \left\{ \epsilon, \frac{H_0}{H_1} \right\}$ after

$$T_0 \geq O \left( \frac{[f(\theta_1)]^{1-\tau}}{C_2 \, \mu \, \eta} \right) = O \left( \ln \left( \min \left\{ \frac{1}{\epsilon}, \frac{H_1}{H_0} \right\} \right) \right) \text{ iterations.}$$

Hence, if $\epsilon \geq \frac{H_0}{H_1}$, Adam with a constant step-size requires $O(\ln(1/\epsilon))$ iterations to guarantee convergence to an $\epsilon$ sub-optimality.

**Phase 2**: Consider iteration $t > T_0$ such that $f(\theta_t) < \frac{H_0}{H_1}$. Simplifying Eq. (47) in this case,

$$f(\theta_{t+1}) \leq f(\theta_t) - \eta \, \|\nabla_t\|_1 \, \underbrace{\frac{\|\nabla_t\|_1}{\sqrt{1 - \beta_2} \sum_{j=0}^{t-1} (\sqrt{\beta_2})^j \, \|\nabla_{t-j}\|_1}}_{(*)} + 2 \, H_0 \, \eta^2 (C_1 + C_3^2) \tag{54}$$

We will now use an inductive proof and prove that for all $t \geq T_0$, $f(\theta_t) \leq f(\theta_{T_0}) \leq \frac{H_0}{H_1} \leq f(\theta_1)$.

**Base Case**: For $t = T_0$, this is true by definition.

**Induction**: We will prove that $f(\theta_{t+1}) \leq f(\theta_{T_0})$. Since $f(\theta_t) \leq f(\theta_{T_0}) \leq \frac{H_0}{H_1} \leq f(\theta_1)$ by the inductive hypothesis, and ensuring that $\eta \leq \frac{\mu}{2C_3 e} \left(\frac{H_0}{H_1}\right)^\tau \frac{1}{H_0 + H_1 \, f(\theta_1)} \ln(1/\beta^{1/4})$, we can use Lemma. 18 and obtain that,

$$(*) = \frac{\|\nabla_t\|_1}{\sqrt{1 - \beta} \sum_{j=0}^{t-1} (\sqrt{\beta})^j \, \|\nabla_{t-j}\|_1} \geq \underbrace{\frac{1 - \beta^{1/4}}{\sqrt{1 - \beta}}}_{:=C_2} \frac{1}{1 + \left(\frac{H_0}{H_1 \, f(\theta_t)}\right)^\tau} = \frac{C_2}{1 + \left(\frac{H_0}{H_1 \, f(\theta_t)}\right)^\tau} \tag{55}$$

Combining Eq. (54) and Eq. (55) and noting that, $\frac{H_0}{H_1 f(\theta_t)} \geq 1 \implies (*) \geq \frac{C_2}{2\left(\frac{H_0}{H_1 f(\theta_t)}\right)^\tau}$,

$$f(\theta_{t+1}) \leq f(\theta_t) - \eta \, \|\nabla_t\|_1 \, \frac{C_2}{2} \left(\frac{H_1}{H_0}\right)^\tau [f(\theta)]^\tau + 2\, H_0\, \eta^2 (C_1 + C_3^2)$$

$$\leq f(\theta_t) - \eta \, \frac{C_2\, \mu}{2} \left(\frac{H_1}{H_0}\right)^\tau [f(\theta_t)]^{2\,\tau} + 2\, H_0\, \eta^2\, (C_1 + C_3^2) \qquad \text{(Using Assn. 3 with } f^* = 0 \text{ and } \mu(\theta) = \mu)$$

Using Lemma. 14 with $A = \frac{C_2\, \mu}{2}\left(\frac{H_1}{H_0}\right)^\tau$, $B = 2\, H_0\, (C_1 + C_3^2)$,

$\bar\eta_1 = \min\left\{\frac{1}{C_3\, \sqrt{H_1}}, \frac{1}{C_3\, \sqrt{H_1}} \ln(1/\sqrt{\beta_1}), \frac{\mu}{2eC_3}\left(\frac{H_0}{H_1}\right)^\tau \frac{1}{H_0 + H_1 f(\theta_1)} \ln(1/\beta_2^{1/4})\right\}$, $r = 2\tau$, $\delta = \epsilon$, we get that with

$\eta = \min\left\{\bar\eta_1, \frac{C_2\, \mu\, \epsilon^{2\tau}}{8\, H_0\, (C_1 + C_3^2)}\left(\frac{H_1}{H_0}\right)^\tau\right\}$, $f(\theta_{t+1}) \leq f(\theta_t) \leq f(\theta_{T_0}) \leq \frac{H_0}{H_1}$ for all $t \geq T_0$. This completes the induction.

Furthermore, for $\tau = \frac{1}{2}$, $f(\theta_{T_\epsilon + T_0}) \leq \epsilon$ after

$$T_\epsilon \geq O\left(\frac{1}{C_2\, \mu\, \eta} \sqrt{\frac{H_0}{H_1}} \ln\left(\frac{f_{T_0}}{2}\right)\right) = \tilde{O}\left(\frac{1}{\epsilon}\right) \text{ iterations.}$$

Else, for $\tau < \frac{1}{2}$, $f(\theta_{T_\epsilon + T_0}) \leq \epsilon$ after

$$T_\epsilon \geq O\left(\frac{[f(\theta_{T_0})]^{1-2\,\tau}}{C_2\, \mu\, \eta\, (1 - 2\,\tau)} \left(\frac{H_0}{H_1}\right)^\tau\right) = O\left(\frac{1}{\epsilon^{2\tau}}\right) \text{ iterations.}$$

Else, for $\tau > \frac{1}{2}$, $f(\theta_{T_\epsilon + T_0}) \leq \epsilon$ after

$$T_\epsilon \geq O\left(\frac{1}{C_2\, \mu\, \eta\, (2\,\tau - 1)\, \epsilon^{2\tau - 1}} \left(\frac{H_0}{H_1}\right)^\tau\right) = O\left(\frac{1}{\epsilon^{4\tau - 1}}\right) \text{ iterations.}$$

Hence, for $T := T + T_\epsilon$, $f(\theta_{T+1}) \leq \epsilon$ after,

$$O\left(\frac{1}{\epsilon^{2\tau}} + \ln\left(\frac{H_1}{H_0}\right)\right) \text{ iterations, if } \tau \leq \frac{1}{2} \; ;$$

$$O\left(\frac{1}{\epsilon^{4\tau - 1}} + \ln\left(\frac{H_1}{H_0}\right)\right) \text{ iterations, if } \tau > \frac{1}{2}$$

$\square$

**Theorem 7.** *For the logistic regression defined in Prop. 2 on linearly separable data with the normalized margin equal to* $\gamma_p := \min_{i \in [n]} \frac{y_i \, \langle x_i, \theta^* \rangle}{\|\theta^*\|_p} > 0$, *Adam with the update in Eq. (18) with* $\theta_1 = 0$ *has the following convergence rate:*

- *Using constant step size* $\eta = O(1)$ *guarantees that after*

$$T = O\left(\frac{1}{\gamma_1^2}\left[n^2 + \ln\left(\frac{1}{n\epsilon}\right)\right]\right)$$

*iterations,* $f(\theta_{T+1}) \leq \epsilon$.

*Proof.* Since we are considering logistic regression on linearly separable data, $f^* = 0$ and $H_0 = 0$. We will use a similar proof as that for Thm. 4. In particular, using Lemma. 19 for $\eta \leq \bar\eta_0 := \frac{1}{C_3\, \sqrt{H_1}}$ and $\beta_1 \leq \beta_2$ where $C_3 := \sqrt{\frac{1-\beta_1}{1-\beta_2}}$, and,

$$f(\theta_{t+1}) \leq f(\theta_t) - \eta \, \langle \nabla_t, d_t \rangle + (H_1\, f(\theta_t))\, \eta^2\, C_3 \tag{56}$$

Simplifying the second term on the RHS of Eq. (56), first note that,

$$\langle \nabla_t, d_t \rangle = \sum_i [\nabla_t]_i [d_t]_i = \sum_i \frac{g_{t,i} \, m_{t,i}}{\sqrt{v_{t,i}}} = \sum_i \frac{g_{t,i} \, (m_{t,i} - g_{t,i})}{\sqrt{v_{t,i}}} + \frac{g_{t,i}^2}{\sqrt{v_{t,i}}}$$

$$\geq - \sum_i \frac{|g_{t,i}| \, |m_{t,i} - g_{t,i}|}{\sqrt{v_{t,i}}} + \sum_i \frac{g_{t,i}^2}{\sqrt{v_{t,i}}}$$

$$\implies -\langle \nabla_t, d_t \rangle \leq \underbrace{\sum_i \frac{|g_{t,i}| \, |m_{t,i} - g_{t,i}|}{\sqrt{v_{t,i}}}}_{:=\text{Term (i)}} - \underbrace{\sum_i \frac{g_{t,i}^2}{\sqrt{v_{t,i}}}}_{:=\text{Term (ii)}} \tag{57}$$

Bounding Term (i) using Lemma. 20 and using a constant step-size equal to $\eta$, we get that,

$$\text{Term (i)} \leq \frac{\beta_1 \, e \, C_3 \, \sum_{s=1}^{t-1} (\sqrt{\beta_1})^{t-1-s} \, \eta_s}{\sqrt{1 - \beta_2}} \, (H_1 \, f(\theta_t)) \tag{58}$$

$$\leq \eta \, \underbrace{\frac{\beta_1 \, e \, C_3}{\sqrt{1 - \beta_2} \, (1 - \sqrt{\beta_1})}}_{:=C_1} (H_1 \, f(\theta_t)) \qquad \text{(Since } \eta_s = \eta\text{)}$$

$$\implies \text{Term (i)} \leq \eta C_1 \, (H_1 \, f(\theta_t)) \tag{59}$$

Similar to the proof of Thm. 3, we use Lemma. 17 to bound Term (ii),

$$\text{Term (ii)} \geq \|\nabla_t\|_1 \underbrace{\frac{\|\nabla_t\|_1}{\sqrt{1 - \beta_2} \, \sum_{j=0}^{t-1} (\sqrt{\beta_2})^j \, \|\nabla_{t-j}\|_1}}_{(*)} \tag{60}$$

We will use Lemma. 18 to bound (*). For this, we need to ensure that (i) the value of $\eta$ ensures that $\|\theta_{t+1} - \theta_t\|_\infty = \eta \, \|d_t\|_\infty = \eta \, C_3 < \underbrace{\frac{\gamma_1}{3n \, e}}_{c/e} \, \frac{1}{H_1 \, f(\theta_1)} \, \underbrace{\ln(1/\beta_2^{1/4})}_{a}$. Setting

$$\eta \leq \bar{\eta}_1 := \frac{\gamma_1}{4 \, n \, e \, C_3} \, \frac{1}{H_1 \, f(\theta_1)} \, \ln(1/\beta_2^{1/4})$$

ensures that this condition is satisfied. Furthermore, we need to ensure that (ii) $f(\theta_t) \leq f(\theta_1)$. For this, we will use an inductive argument and prove that for all $t \geq 1$, $f(\theta_t) \leq f(\theta_1)$.

**Base Case**: For $t = 1$, this is true by definition.

**Inductive Hypothesis**: Assume for iteration $t > 1$, $f(\theta_t) \leq f(\theta_1)$.

**Induction**: We will prove that $f(\theta_{t+1}) \leq f(\theta_1)$. Since $f(\theta_t) \leq f(\theta_1)$ by the inductive hypothesis, and $\eta \leq \bar{\eta}_1$, we can use Lemma. 18 and obtain that,

$$(*) = \frac{\|\nabla_t\|_1}{\sqrt{1 - \beta} \, \sum_{j=0}^{t-1} (\sqrt{\beta_2})^j \, \|\nabla_{t-j}\|_1} \geq \frac{\|\nabla_t\|_1}{\|\nabla_t\|_1 + c} \, \frac{1 - \sqrt{\beta_2} \, \exp(a)}{\sqrt{1 - \beta_2}} \tag{61}$$

To complete the induction, we will consider two phases. For this, define $T_0$ to be the first iteration s.t. $f(\theta_{T_0}) < \max \left\{ \epsilon, \frac{\ln(2)}{n} \right\}$.

**Phase 1:** For all $t < T_0$, $f(\theta_t) > \frac{\ln(2)}{n}$ and we use part (2) of Prop. 2 to conclude that $\|\nabla_t\|_1 \geq \frac{\gamma_1}{3n}$. Using this relation with Eq. (61) and using that $c = \frac{\gamma_1}{3n}, \alpha = \ln(\frac{1}{\beta_2^{1/4}})$

$$(*) \geq \frac{1}{2} \underbrace{\frac{1 - \beta_2^{1/4}}{\sqrt{1 - \beta_2}}}_{C_4} \qquad \text{(Using that } \eta \leq \bar{\eta}_1 \text{ and since } \beta_2 < 1\text{)}$$

Combining the above relation with Eqs. (56) and (59),

$$f(\theta_{t+1}) \leq f(\theta_t) - \frac{\eta\, C_4\, \|\nabla_t\|_1}{2} + H_1\, f(\theta_t)\, \eta^2 (C_1 + C_3^2)$$

$$\leq f(\theta_t) - \frac{\eta\, C_4\, \|\nabla_t\|_1}{2} + H_1\, f(\theta_1)\, \eta^2 (C_1 + C_3^2) \qquad \text{(Using the inductive hypothesis)}$$

$$\leq f(\theta_t) - \frac{\eta\, C_4\, \gamma_1}{6n} + H_1\, f(\theta_1)\, \eta^2 (C_1 + C_3^2) \qquad \text{(Using that for } t < T_0,\, \|\nabla_t\|_1 \geq \tfrac{\gamma_1}{3n})$$

$$\leq f(\theta_t) - \frac{\eta\, C_4\, \gamma_1}{12n} \qquad \text{(Setting } \eta \leq \bar{\eta}_2 := \tfrac{\gamma_1\, C_4}{12\, H_1\, f(\theta_1)\, n\, (C_1 + C_3^2)})$$

Since $f(\theta_{t+1}) \leq f(\theta_t) \leq f(\theta_1)$, this completes the induction for Phase 1. By recursing for $T_0 - 1$ iterations,

$$\implies f(\theta_{T_0}) \leq f(\theta_1) - \frac{\eta\, C_4\, \gamma_1}{12n}\,(T_0 - 1) = \ln(2) - \frac{\eta\, C_4\, \gamma_1}{12n}\,(T_0 - 1)$$

Hence, by using that $\eta \leq \min\{\bar{\eta}_0, \bar{\eta}_1, \bar{\eta}_2\}$,

$$T_0 \geq O\left(\frac{n}{\eta\, C_4\, \gamma_1}\right) = O\left(\left(\frac{n}{\gamma_1}\right)^2\right)$$

iterations guarantee that $f(\theta_{T_0}) \leq \frac{\ln(2)}{n}$.

**Phase 2:** After $T_0$ iterations, $f(\theta_{T_0}) \leq \frac{\ln(2)}{n} < f(\theta_1)$. We will now prove that for $t > T_0$, $f(\theta_t) \leq f(\theta_{T_0}) < f(\theta_1)$, thus completing the induction in Phase 2. We will again do this proof via induction.

**Base Case**: For $t = T_0$, this is true by definition.

**Inductive Hypothesis**: Assume for iteration $t > T_0$, $f(\theta_t) \leq f(\theta_{T_0})$.

**Induction**: We will prove that $f(\theta_{t+1}) \leq f(\theta_{T_0})$. Since $f(\theta_t) \leq f(\theta_{T_0}) < f(\theta_1)$ by the inductive hypothesis, we can again use an appropriate step-size

$$\eta \leq \bar{\eta}_3 := \frac{\gamma_1}{2e\, C_3\, H_1} \ln\left(\frac{1}{\beta_2^{1/4}}\right),$$

which then guarantees

$$\|\theta_{t+1} - \theta_t\|_p \leq \eta\, C_3 \leq \frac{\gamma_1}{2eH_1} \ln\left(\frac{1}{\beta_2^{1/4}}\right) = \frac{ca}{eH_1 f(\theta)}, \qquad c = \frac{\gamma_1\, f(\theta_1)}{2},\; a = \ln\left(\frac{1}{\beta_2^{1/4}}\right)$$

and Eq. (61) to bound (*) as follows:

$$(*) = \frac{\|\nabla_t\|_1}{\sqrt{1-\beta_2} \sum_{j=0}^{t-1}(\sqrt{\beta_2})^j\, \|\nabla_{t-j}\|_1} \geq \frac{\|\nabla_t\|_1}{\|\nabla_t\|_1 + c} \frac{1 - \sqrt{\beta_2}\, \exp(a)}{\sqrt{1-\beta_2}}$$

Using part (1) of Prop. 2 to conclude that $\|\nabla_t\|_1 \geq \frac{\gamma_p}{2} f(\theta_t)$, using the induction hypothesis to further bound $f(\theta_t)$ by $f(\theta_1)$ and using that $c = \frac{\gamma_1\, f(\theta_1)}{2}$,

$$(*) \geq \frac{1}{2}\, \underbrace{\frac{1 - \beta_2^{1/4}}{\sqrt{1-\beta_2}}}_{=:C_4} \qquad \text{(Using that } \eta \leq \bar{\eta}_3)$$

Combining the above relation with Eqs. (56) and (59),

$$f(\theta_{t+1}) \leq f(\theta_t) - \frac{\eta\, C_4\, \|\nabla_t\|_1}{2} + H_1\, f(\theta_t)\, \eta^2\, (C_1 + C_3^2)$$

$$\leq f(\theta_t) - \frac{\eta\, C_4\, \gamma_1}{4} f(\theta_t) + H_1\, f(\theta_t)\, \eta^2\, (C_1 + C_3^2) \qquad \text{(Using Part (1) of Prop. 2)}$$

$$\leq f(\theta_t) - \frac{\eta\, C_4\, \gamma_1}{8} f(\theta_t) \qquad\qquad \text{(Setting } \eta \leq \bar{\eta}_4 := \frac{\gamma_1}{8\, H_1\, (C_1 + C_3^2)})$$

Since $f(\theta_{t+1}) < f(\theta_t) < f(\theta_{T_0})$, this completes the induction. Furthermore, by recursing for $T$ iterations

$$\implies f(\theta_{T_0+T}) \leq f(\theta_{T_0}) \exp\left(-\frac{\eta\, C_4\, \gamma_1}{8} T\right) \leq \frac{\ln(2)}{n} \exp\left(-\frac{\eta\, C_4\, \gamma_1}{8} T\right)$$

Hence, by using a step-size $\eta \leq \min\{\bar{\eta}_0, \bar{\eta}_3, \bar{\eta}_4\}$,

$$T \geq O\left(\frac{1}{\eta\, C_4\, \gamma_1} \ln\left(\frac{\ln(2)}{n\, \epsilon}\right)\right) = O\left(\frac{1}{\gamma_1^2} \ln\left(\frac{1}{n\epsilon}\right)\right)$$

iterations suffice to guarantee that $f(\theta_{T+T_0}) \leq \epsilon$. Hence, attaining an $\epsilon$ sub-optimality for logistic regression requires,

$$T_{\text{final}} = O\left(\frac{1}{\gamma_1^2} \ln\left(\frac{1}{n\epsilon}\right)\right) + O\left(\left(\frac{n}{\gamma_1}\right)^2\right)$$

$$= O\left(\frac{1}{\gamma_1^2} \left[n^2 + \ln\left(\frac{1}{n\epsilon}\right)\right]\right) \text{ iterations.}$$

$\square$

### F.1. Helper Lemmas

**Lemma 19.** *Under Assn. 1, Assn. 2 with $H_0 \geq 0$ and $H_1 \geq 0$, for the coordinate-wise Adam update in Eq. (18), with $\eta \leq \frac{1}{B\sqrt{H_1}}$ and $\beta_1 \leq \beta_2$, where $B := \sqrt{\frac{1-\beta_1}{1-\beta_2}}$,*

$$f(\theta_{t+1}) \leq f(\theta_t) - \eta\, \langle \nabla_t, d_t \rangle + (H_0 + H_1\, f(\theta_t))\, \eta^2\, B^2 \quad ; \quad \|d_t\|_\infty \leq B$$

*Proof.* If $\|\theta_{t+1} - \theta_t\|_\infty \leq \frac{1}{\sqrt{H_1}}$, using Lemma. 7 with $p = \infty$ and $q = 1$, and denoting $\nabla_t := \nabla f(\theta_t)$ for convenience,

$$f(\theta_{t+1}) \leq f(\theta_t) - \eta\, \langle \nabla_t, d_t \rangle + (H_0 + H_1\, f(\theta_t))\, \eta^2\, \|d_t\|_\infty^2$$

Simplifying the third term on the RHS, $\|d_t\|_\infty = \max_i \frac{m_{t,i}}{\sqrt{v_{t,i}}}$, and

$$m_{t,i} = (1 - \beta_1) \sum_{s=1}^{t} \beta_1^{t-s} g_{s,i}$$

$$\implies m_{t,i}^2 \leq (1 - \beta_1)^2 \left(\sum_{s=1}^{t} \beta_1^{t-s}\right) \left(\sum_{s=1}^{t} \beta_1^{t-s} g_{s,i}^2\right) \qquad\qquad \text{(By Cauchy–Schwarz)}$$

$$\leq (1 - \beta_1) \sum_{s=1}^{t} \beta_1^{t-s} g_{s,i}^2 \qquad\qquad \text{(By geometric series)}$$

$$\leq \frac{1 - \beta_1}{1 - \beta_2} \left[(1 - \beta_2) \sum_{s=1}^{t} \beta_2^{t-s} g_{s,i}^2\right] \qquad\qquad \text{(Since } \beta_1 \leq \beta_2)$$

$$= \frac{1 - \beta_1}{1 - \beta_2} v_{t,i} \qquad\qquad \text{(By definition of } v_{t,i})$$

$$\implies \frac{m_{t,i}^2}{v_{t,i}} \leq \frac{1 - \beta_1}{1 - \beta_2} \implies \frac{m_{t,i}}{\sqrt{v_{t,i}}} \leq B := \sqrt{\frac{1 - \beta_1}{1 - \beta_2}}$$

$$\implies \|\theta_{t+1} - \theta_t\|_\infty \leq \eta\, \|d_t\|_\infty \leq \eta\, B$$

Ensuring that $\eta \leq \frac{1}{B\sqrt{H_1}}$ guarantees that $\|\theta_{t+1} - \theta_t\|_\infty \leq \frac{1}{\sqrt{H_1}}$. Combining the above inequalities,

$$f(\theta_{t+1}) \leq f(\theta_t) - \eta \langle \nabla_t, d_t \rangle + (H_0 + H_1 f(\theta_t)) \eta^2 B^2 \tag{62}$$

$\square$

**Lemma 20.** *Under Assn. 1, Assn. 2, if $B = \sqrt{\frac{1-\beta_1}{1-\beta_2}}$, then, for the Adam update in Eq. (18) with $\eta \leq \frac{1}{B\sqrt{H_1}} \min\{1, \ln(1/\sqrt{\beta_1})\}$ for all $t$,*

$$\sum_i \frac{|g_{t,i}| \, |m_{t,i} - g_{t,i}|}{\sqrt{v_{t,i}}} \leq C_1 \left(H_0 + H_1 f(\theta_t)\right) \quad \text{where,} \quad C_1 = \frac{\beta_1 \, e \, B \sum_{s=1}^{t-1} (\sqrt{\beta_1})^{t-1-s} \eta_s}{\sqrt{1-\beta_2}}.$$

*Proof.*

$$\sum_i \frac{|g_{t,i}| \, |m_{t,i} - g_{t,i}|}{\sqrt{v_{t,i}}} \leq \frac{1}{\sqrt{1-\beta_2}} \sum_i |m_{t,i} - g_{t,i}| \qquad \text{(Since } v_{t,i} \geq (1-\beta_2) g_{t,i}^2)$$

$$\leq \frac{\|m_t - g_t\|_1}{\sqrt{1-\beta_2}} \tag{63}$$

In order to bound $\|m_t - g_t\|_1$, note that for all $i \in [D]$,

$$|m_{t,i} - g_{t,i}| = |\beta_1 m_{t-1,i} + (1-\beta_1) g_{t,i} - g_{t,i}| = \beta_1 |m_{t-1,i} - g_{t,i}| \tag{64}$$

$$\leq \beta_1 |m_{t-1,i} - g_{t-1,i}| + \beta_1 |g_{t-1,i} - g_{t,i}| \qquad \text{(By triangle inequality)}$$

$$\implies \sum_i |m_{t,i} - g_{t,i}| \leq \beta_1 \sum_i |m_{t-1,i} - g_{t-1,i}| + \beta_1 \sum_i |g_{t-1,i} - g_{t,i}|$$

$$\implies \|m_t - g_t\|_1 \leq \beta_1 \|m_{t-1} - \nabla_{t-1}\|_1 + \beta_1 \|\nabla_t - \nabla_{t-1}\|_1$$

Simplifying the second term on the RHS using Lemma. 6 with $p = \infty$, $q = 1$, $y = \theta_{t-1}$ and $x = \theta_t$. Ensuring that $\eta_{t-1} \leq \frac{1}{B\sqrt{H_1}}$ guarantees that $\|\theta_t - \theta_{t+1}\|_\infty \leq \frac{1}{\sqrt{H_1}}$. Hence,

$$\implies \|m_t - g_t\|_1 \leq \beta_1 \|m_{t-1} - \nabla_{t-1}\|_1 + \beta_1 (H_0 + H_1 f(\theta_t)) e \|\theta_t - \theta_{t-1}\|_\infty$$

$$\leq \beta_1 \|m_{t-1} - \nabla_{t-1}\|_1 + \beta_1 \underbrace{(H_0 + H_1 f(\theta_t))}_{:= \bar{L}_t} e \eta_{t-1} B \qquad \text{(Using Lemma. 19)}$$

$$\implies \frac{\|m_t - g_t\|_1}{\bar{L}_t} \leq \beta_1 \frac{\|m_{t-1} - \nabla_{t-1}\|_1}{\bar{L}_t} + \beta_1 e \eta_{t-1} B \tag{65}$$

We now write $\bar{L}_t$ in terms of $\bar{L}_{t-1}$. In particular, using Lemma. 5 with $p = \infty$, $q = 1$ and $y = \theta_{t-1}$, $x = \theta_t$,

$$\bar{L}_t = H_0 + H_1 f(\theta_t) \geq H_0 + H_1 \left[\left(f(\theta_{t-1}) + \frac{H_0}{H_1}\right) \exp\left(-\sqrt{H_1} \|\theta_t - \theta_{t-1}\|_\infty\right) - \frac{H_0}{H_1}\right]$$

$$= (H_0 + H_1 f(\theta_{t-1})) \exp\left(-\sqrt{H_1} \|\theta_t - \theta_{t-1}\|_\infty\right)$$

$$= \bar{L}_{t-1} \exp\left(-\sqrt{H_1} \|\theta_t - \theta_{t-1}\|_\infty\right)$$

$$\implies \frac{1}{\bar{L}_t} \leq \frac{1}{\bar{L}_{t-1}} \exp\left(\sqrt{H_1} \|\theta_t - \theta_{t-1}\|_\infty\right) \leq \frac{1}{\bar{L}_{t-1}} \frac{1}{\sqrt{\beta_1}}$$

$$\text{(Using Eq. (62) and since } \eta_{t-1} \leq \frac{1}{B\sqrt{H_1}} \ln(1/\sqrt{\beta_1}))$$

Combining the above inequality with Eq. (65),

$$\frac{\|m_t - g_t\|_1}{\bar{L}_t} \leq \sqrt{\beta_1} \frac{\|m_{t-1} - \nabla_{t-1}\|_1}{\bar{L}_{t-1}} + \beta_1 e \eta_{t-1} B \tag{66}$$

$$\le \beta_1 \, e \, B \sum_{s=1}^{t-1} (\sqrt{\beta_1})^{t-1-s} \eta_s \qquad \text{(Recursing and using that } m_{0,i} = g_{1,i})$$

$$\implies \|m_t - g_t\|_1 \le (H_0 + H_1 \, f(\theta_t)) \, \beta_1 \, e \, B \sum_{s=1}^{t-1} (\sqrt{\beta_1})^{t-1-s} \eta_s \tag{67}$$

Combining the above inequality with Eq. (63),

$$\sum_i \frac{|g_{t,i}| \, |m_{t,i} - g_{t,i}|}{\sqrt{v_{t,i}}} \le (H_0 + H_1 \, f(\theta_t)) \underbrace{\frac{\beta_1 \, e \, B \sum_{s=1}^{t-1} (\sqrt{\beta_1})^{t-1-s} \eta_s}{\sqrt{1 - \beta_2}}}_{:=C_1} = C_1 \, (H_0 + H_1 \, f(\theta_t))$$

□

# G. Lower-Bound

**Theorem 6.** *Starting from $\theta_1 = 0$, consider $T$ iterations of the form $\theta_{t+1} = \theta_t - \eta_t \, m_t$ s.t. (i) the effective step-size $\eta_t$ is bounded, non-increasing and independent of $T$, (ii) for $\beta \in [0,1)$, $m_t = (1 - \beta) \sum_{s=1}^t \beta^{t-s} \nabla f(\theta_s)$ is the momentum vector and (iii) $\eta_1 \le \ln(1/\beta)$. When minimizing the logistic loss with these update restrictions, the convergence rate is lower-bounded by $\Omega(1/T)$.*

*Proof.* Note that $\nabla f(\theta) = \frac{-1}{1 + \exp(\theta)} < 0$ for all finite $\theta$. We define $\nabla_t := \nabla f(\theta_t)$. Since $\beta \in [0,1)$, for all $t$,

$$m_t = (1 - \beta) \sum_{s=1}^t \beta^{t-s} \nabla_s \implies m_t < 0 \tag{68}$$

$$\implies \theta_{t+1} = \theta_t - \eta_t \, m_t = \theta_t + \eta_t |m_t| \le \theta_t + \eta_1 |m_t| \qquad \text{(Since } \eta_t \text{ is non-increasing)}$$

$$\implies \theta_{t+1} = \theta_t + \eta_1 |m_t| \tag{69}$$

Hence, $\theta_{t+1} > \theta_t$ for all $t$. Since $\theta_1 = 0$, $\theta_t \ge 0$ for all $t \ge 1$ and consequently, $|\nabla_t| \le 1$. Using this relation to uniformly bound the magnitude of $m_t$,

$$|m_t| = |(1 - \beta) \sum_{s=1}^t \beta^{t-s} \nabla f(\theta_s)| \le (1 - \beta) \sum_{s=1}^t \beta^{t-s} |\nabla_s| \qquad \text{(By triangle inequality)}$$

$$\implies |m_t| \le 1 \tag{70}$$

We will now relate $|m_t|$ to $|\nabla_t|$. For this note that the logistic loss satisfies Assn. 1, Assn. 2 with $H_0 = 0$ and $H_1 = 1$, and Assn. 3 with $\zeta = 1$, $f^* = 0$ and $\mu = 1$. Using Part 2 of Lemma. 10 and noting that $c := \frac{\mu(H_0 + H_1 f^*)}{H_1} = 0$, we get that,

$$|\nabla_{t-1}| \le |\nabla_t| \exp(|\theta_t - \theta_{t-1}|) = |\nabla_t| \exp(\eta_{t-1} |m_{t-1}|) \le |\nabla_t| \exp(\eta_1 |m_{t-1}|) \qquad \text{(Since } \eta_t \text{ is non-increasing)}$$

$$\implies |\nabla_{t-1}| \le |\nabla_t| \exp(\eta_1) \implies |\nabla_{t-j}| \le |\nabla_t| \exp(\eta_1 j) \qquad \text{(Using Eq. (70))}$$

Using the above inequality to bound $|m_t|$ in terms of $|\nabla_t|$,

$$|m_t| \le (1 - \beta) \sum_{j=0}^t \beta^j |\nabla_{t-j}| \qquad \text{(Using the definition of } m_t \text{ and triangle inequality)}$$

$$\le (1 - \beta) |\nabla_t| \sum_{j=0}^t \beta^j \exp(\eta_1 j)$$

$$\le \frac{1 - \beta}{1 - \beta \exp(\eta_1)} |\nabla_t| \qquad \text{(Using the assumption that } \eta_1 < \ln(1/\beta))$$

$$\implies |m_t| \le \frac{1 - \beta}{1 - \beta \exp(\eta_1)} |\nabla_t| \tag{71}$$

Using Eq. (71) with Eq. (69),

$$\theta_{t+1} = \theta_t + \eta_1 \, |m_t| \leq \theta_t + \underbrace{\frac{\eta_1 \, (1 - \beta)}{1 - \beta \, \exp(\eta_1)}}_{:=C} \, |\nabla_t| = \theta_t + \frac{C}{1 + \exp(\theta_t)} \qquad \text{(Using the gradient expression)}$$

In order to complete the proof we will use the following inequalities to bound $\frac{1}{1 + \exp(\theta)}$.

$$\forall \theta, \quad \frac{1}{1 + \exp(\theta)} < \frac{1}{\exp(\theta)} = \exp(-\theta) \quad ; \quad \forall \theta \geq 0, \; \frac{1}{1 + \exp(\theta)} = \frac{\exp(-\theta)}{1 + \exp(-\theta)} > \frac{\exp(-\theta)}{2} \qquad (72)$$

Using the above bounds, since $C > 0$,

$$\theta_{t+1} - \theta_t \leq C \, \exp(-\theta_t)$$
$$\implies \exp(\theta_{t+1}) - \exp(\theta_t) = \exp(\theta_t)[\exp(\theta_{t+1} - \theta_t) - 1] \leq \exp(\theta_t) \, [\exp(C \, \exp(-\theta_t)) - 1]$$

For $x \in (0, 1]$ and $c > 0$, $\exp(cx) - 1 \leq (\exp(c)) \, c \, x$. Since $\theta_t \geq 0$, using this inequality with $x = \exp(-\theta_t) \leq 1$ and $c = C > 0$,

$$\exp(\theta_{t+1}) - \exp(\theta_t) \leq \exp(\theta_t) \, [C \, (\exp(C)) \, \exp(-\theta_t)] = C \, \exp(C)$$

Summing up from $t = 1$ to $T$ and telescoping,

$$\exp(\theta_{T+1}) - \exp(\theta_1) \leq C \, \exp(C) \, T \implies \exp(\theta_{T+1}) \leq 1 + C \, \exp(C) \, T \qquad \text{(Since } \theta_1 = 0\text{)}$$
$$\implies \exp(-\theta_{T+1}) \geq \frac{1}{1 + C \, \exp(C) \, T} \qquad (73)$$

For all $x \in [0, 1]$, $\ln(1 + x) > \frac{x}{2}$. Since $\theta_{T+1} \geq 0$, $\exp(-\theta_{T+1}) \leq 1$, using this inequality with $x = \exp(-\theta_{T+1})$,

$$f(\theta_{T+1}) = \ln(1 + \exp(-\theta_{T+1})) \geq \frac{\exp(-\theta_{T+1})}{2} \geq \frac{1}{2 \, (1 + C \, \exp(C) \, T)} \qquad (74)$$

Since $C$ is a constant independent of $T$, the convergence rate is lower-bounded by $\Omega(1/T)$. $\qquad \square$

### G.1. Methods satisfying the update restrictions

**Gradient Descent:** For a bounded, constant step-size $\eta > 0$ independent of $T$,

$$\theta_{t+1} = \theta_t - \eta \, \nabla f(\theta_t)$$

Hence, (i) $\eta_t = \eta$ is non-increasing, (ii) $\beta = 0 \in [0, 1)$ and (iii) since $\beta = 0$, there is no restriction on $\eta$.

**Heavy-Ball Momentum:** For a constant momentum parameter $\beta_{\text{HB}} \in (0, 1)$ and a bounded, constant step-size $\eta > 0$ independent of $T$, such that $\eta \leq \ln(1/\beta_{\text{HB}})$,

$$\theta_{t+1} = \theta_t - \eta \, m_t \quad ; \quad m_t = (1 - \beta_{\text{HB}}) \sum_{s=1}^{t} \beta_{\text{HB}}^{t-s} \, \nabla f(\theta_s)$$

Hence, (i) $\eta_t = \eta$ is non-increasing, (ii) $\beta = \beta_{\text{HB}} \in [0, 1)$ and (iii) for $\eta \leq \ln(1/\beta_{\text{HB}})$.

**AdaGrad:** For a bounded, constant step-size $\eta > 0$ independent of $T$,

$$\theta_{t+1} = \theta_t - \frac{\eta}{\sqrt{v_t}} \, \nabla f(\theta_t) \quad ; \quad v_t = \sum_{s=1}^{t} \|\nabla f(\theta_s)\|^2$$

Hence, (i) $\eta_t = \frac{\eta}{\sqrt{v_t}}$ is non-increasing, (ii) $\beta = 0 \in [0, 1)$ and (iii) since $\beta = 0$, there is no restriction on $\eta$.

**AMSGrad:** For a constant $\beta_1 \in (0,1)$, $\beta_2 \in (0,1)$ and a bounded, constant step-size $\eta > 0$ independent of $T$, such that $\eta \le \frac{1}{2} \ln{(1/\beta_1)}$, $m_0 = g_1$, $v_0 = 0$, and $\hat{v}_0 = 0$,

$$\theta_{t+1} = \theta_t - \frac{\eta\, m_t}{\sqrt{\hat{v}_t}} \quad ; \quad \hat{v}_t = \max\{\hat{v}_{t-1}, v_t\}$$

$$v_t = (1 - \beta_2) \sum_{s=1}^{t} \beta_2^{t-s} \|\nabla f(\theta_s)\|^2 \quad ; \quad m_t = (1 - \beta_1) \sum_{s=1}^{t} \beta_1^{t-s} \nabla f(\theta_s)$$

Hence, (i) $\eta_t = \frac{\eta}{\sqrt{\max\{v_t, \hat{v}_{t-1}\}}}$ is non-increasing, (ii) $\beta_1 \in (0,1)$ and (iii) the effective step-size

$$\eta_1 = \frac{\eta}{\sqrt{\max\{v_1, \hat{v}_0\}}} = \frac{\eta}{|\nabla_1|} = 2\,\eta \le \ln{(1/\beta_1)} \qquad\qquad \text{(Since } \hat{v}_0 = 0 \text{ and } \theta_1 = 0\text{)}$$

## H. Adam with $(L_0, L_1)$ Non-Uniform Smoothness

**Assumption 4.** $f$ is $(L_0, L_1)$ *non-uniform smooth i.e. for constants* $L_0 \ge 0$, $L_1 > 0$, *and* $p, q \ge 1$ *s.t.* $\frac{1}{p} + \frac{1}{q} = 1$, *for all* $\theta$, $\left\|\nabla^2 f(\theta)\right\|_{p \to q} \le L_0 + L_1 \left\|\nabla f(\theta)\right\|_q$.

In the following proofs, for simplicity, we use $\nabla_t := \nabla f(\theta_t)$, and $g_{t,i}$ is the $i$-th component of this vector. The following theorem establishes an $O\left(\frac{1}{\epsilon}\right)$ rate for `Adam` when $\epsilon = O\left(\frac{L_0}{L_1}\right)$.

### H.1. Adam convergence under General NUS

**Theorem 5.** *Under Assn. 1 and Assn. 4 with $L_0 \ge 0$ and $L_1 > 0$, and $f^* = 0$, Adam with the update in Eq. (18) has the following convergence rate:*

- *If $\epsilon \ge \frac{L_0}{L_1}$, using $\eta_t = \eta = O(1)$ guarantees that after $T = O\left(\frac{1}{\epsilon}\right)$ iterations, $\|\nabla f(\theta_T)\|_1 \le \epsilon$.*

- *Else, using $\eta_t = \eta = O(\epsilon^2)$ guarantees that after $T = O\left(\frac{1}{\epsilon^2}\right)$ iterations, $\min_{t \le T} \|\nabla f(\theta_t)\|_1 \le \epsilon$.*

*Proof.* Using Lemma. 25 for $\eta \le \bar{\eta}_0 := \frac{1}{C_3 L_1}$ and $\beta_1 \le \beta_2$ where $C_3 := \sqrt{\frac{1-\beta_1}{1-\beta_2}}$, and,

$$f(\theta_{t+1}) \le f(\theta_t) - \eta \langle \nabla_t, d_t \rangle + (L_0 + L_1 \|\nabla f(\theta_t)\|_1)\, \eta^2\, C_3^2 \tag{75}$$

Simplifying the second term on the RHS of Eq. (75), first note that,

$$\langle \nabla_t, d_t \rangle = \sum_i [\nabla_t]_i\, [d_t]_i = \sum_i \frac{g_{t,i}\, m_{t,i}}{\sqrt{v_{t,i}}} = \sum_i \frac{g_{t,i}\,(m_{t,i} - g_{t,i})}{\sqrt{v_{t,i}}} + \frac{g_{t,i}^2}{\sqrt{v_{t,i}}}$$

$$\ge -\sum_i \frac{|g_{t,i}|\,|m_{t,i} - g_{t,i}|}{\sqrt{v_{t,i}}} + \sum_i \frac{g_{t,i}^2}{\sqrt{v_{t,i}}}$$

$$\implies -\langle \nabla_t, d_t \rangle \le \underbrace{\sum_i \frac{|g_{t,i}|\,|m_{t,i} - g_{t,i}|}{\sqrt{v_{t,i}}}}_{:=\text{Term (i)}} - \underbrace{\sum_i \frac{g_{t,i}^2}{\sqrt{v_{t,i}}}}_{:=\text{Term (ii)}} \tag{76}$$

Bounding Term (i) using Lemma. 26 and using a constant step-size equal to $\eta$, we get that,

$$\text{Term (i)} \le \frac{\beta_1\, e\, C_3 \sum_{s=1}^{t-1}(\sqrt{\beta_1})^{t-1-s} \eta_s}{\sqrt{1 - \beta_2}}\,(L_0 + L_1 \|\nabla f(\theta_t)\|_1) \tag{77}$$

$$\le \eta \underbrace{\frac{\beta_1\, e\, C_3}{\sqrt{1 - \beta_2}\,(1 - \sqrt{\beta_1})}}_{:=C_1}(L_0 + L_1 \|\nabla f(\theta_t)\|_1) \qquad\qquad \text{(Since } \eta_s = \eta\text{)}$$

$$\implies \text{Term (i)} \le \eta C_1 \left( L_0 + L_1 \left\| \nabla f(\theta_t) \right\|_1 \right) \tag{78}$$

We use Lemma. 17 to bound Term (ii) $= \sum_i \frac{g_{t,i}^2}{\sqrt{v_{t,i}}}$, with $u_t := \sqrt{1 - \beta_2} \sum_{j=0}^{t-1} (\sqrt{\beta_2})^j \left\| \nabla_{t-j} \right\|_1$,

$$\sum_i \frac{g_{t,i}^2}{\sqrt{v_{t,i}}} \ge \left\| \nabla_t \right\|_1 \frac{\left\| \nabla_t \right\|_1}{\sqrt{1 - \beta_2} \sum_{j=0}^{t-1} (\sqrt{\beta_2})^j \left\| \nabla_{t-j} \right\|_1} = \frac{\left\| \nabla_t \right\|_1^2}{u_t} \tag{79}$$

Combining Eq. (75) with the bounds on Term (i) and Term (ii),

$$f(\theta_{t+1}) \le f(\theta_t) - \eta \frac{\left\| \nabla_t \right\|_1^2}{u_t} + \left( L_0 + L_1 \left\| \nabla f(\theta_t) \right\|_1 \right) \eta^2 \left( C_3^2 + C_1 \right) \tag{80}$$

**Case (1): For $\epsilon \ge \frac{L_0}{L_1}$:** To bound $u_t$, we use Lemma. 22, with $a = \ln(1/\beta_2^{1/4})$, while $\sqrt{\beta_2} \exp(\ln(1/\beta_2^{1/4})) = \beta_2^{1/4} \le 1$ and setting $\eta \le \bar{\eta}_1 := \frac{1}{2 C_3 L_1} \ln(1/\beta_2^{1/4})$

$$u_t \le \frac{\sqrt{1 - \beta_2} \left( \left\| \nabla_t \right\|_1 + \frac{L_0}{L_1} \right)}{1 - \sqrt{\beta_2} \exp(a)} \tag{81}$$

Combining the above inequality with Eq. (80) and $C_2 := \frac{1 - \beta_2^{1/4}}{\sqrt{1 - \beta_2}}$, we have

$$f(\theta_{t+1}) \le f(\theta_t) - \eta C_2 \frac{\left\| \nabla_t \right\|_1^2}{\left\| \nabla_t \right\|_1 + \frac{L_0}{L_1}} + \left( L_0 + L_1 \left\| \nabla_t \right\|_1 \right) \eta^2 \left( C_1 + C_3^2 \right) \tag{82}$$

Define $T$ to be the first iteration s.t. $\left\| \nabla f(\theta_T) \right\|_1 < \epsilon, \frac{L_0}{L_1}$, and analyze the above inequality for $t < T$ where $\left\| \nabla f(\theta_t) \right\|_1 \ge \epsilon \ge \frac{L_0}{L_1}$. Setting $\eta \le \bar{\eta}_2 := \frac{C_2}{8 L_1 (C_1 + C_3^2)}$ and simplifying Eq. (80) in this case,

$$f(\theta_{t+1}) \le f(\theta_t) - \frac{\eta C_2}{2} \left\| \nabla_t \right\|_1 + \left( 2 L_1 \left\| \nabla_t \right\|_1 \right) \eta^2 \left( C_1 + C_3^2 \right)$$

$$\le f(\theta_t) - \frac{\eta C_2}{4} \left\| \nabla_t \right\|_1 \qquad \text{(Since } \eta \le \bar{\eta}_2\text{)}$$

$$\implies \frac{\eta C_2}{4} \left\| \nabla_t \right\|_1 \le f(\theta_t) - f(\theta_{t+1}) \tag{83}$$

Note that from the above we have $f(\theta_{t+1}) \le f(\theta_t)$ for all $t \le T$ which indicates $f(\theta_T) \le f(\theta_1)$. Recursing for $T$ iterations, we have

$$\frac{\eta C_2}{4} T \left\| \nabla_T \right\| \le \frac{\eta C_2}{4} \sum_{t=1}^{T} \left\| \nabla_t \right\|_1 \qquad \text{(Since } \left\| \nabla_T \right\| \le \frac{L_0}{L_1} \text{ and } \left\| \nabla_t \right\| \ge \frac{L_0}{L_1} \ \forall t < T\text{)}$$

$$\le f(\theta_1) - f(\theta_{T+1})$$

$$\le f(\theta_1) \qquad \text{(Since } f(\theta) \ge 0\text{)}$$

$$\implies \left\| \nabla_T \right\| \le \frac{4 f(\theta_1)}{\eta C_2 T}$$

Hence when $T \ge \frac{4 f(\theta_1)}{\eta C_2 \max\{\epsilon, \frac{L_0}{L_1}\}} = O(\frac{1}{\epsilon})$, then $\left\| \nabla_T \right\|_1 \le \epsilon$, where $\eta = \min \{ \bar{\eta}_0, \bar{\eta}_1, \bar{\eta}_2 \}$.

**Case (2): For $\epsilon < \frac{L_0}{L_1}$:** Starting from Eq. (80) and summing from $t = 1$ to $T$,

$$f(\theta_{T+1}) \le f(\theta_1) - \eta \underbrace{\sum_{t=1}^{T} \frac{\left\| \nabla_t \right\|_1^2}{u_t}}_{:= (*)} + L_0 \eta^2 \left( C_3^2 + C_1 \right) T + L_1 \eta^2 \left( C_3^2 + C_1 \right) \sum_{t=1}^{T} \left\| \nabla_t \right\|_1$$

In order to simplify (*), note that by Cauchy Schwarz,

$$\sum_{t=1}^{T} \|\nabla_t\| = \sum_{t=1}^{T} \frac{\|\nabla_t\|}{\sqrt{u_t}} \sqrt{u_t} \leq \sqrt{\sum_{t=1}^{T} \frac{\|\nabla_t\|^2}{u_t}} \sqrt{\sum_{t=1}^{T} u_t} \implies (*) \geq \frac{\left(\sum_{t=1}^{T} \|\nabla_t\|\right)^2}{\sum_{t=1}^{T} u_t}$$

Simplifying $\sum_{t=1}^{T} u_t$,

$$\sum_{t=1}^{T} u_t = \sqrt{1-\beta_2} \sum_{t=1}^{T} \left[\sum_{j=0}^{t-1} (\sqrt{\beta_2})^j \|\nabla_{t-j}\|_1\right] = \sqrt{1-\beta_2} \sum_{t=1}^{T} \left[\sum_{s=1}^{t} (\sqrt{\beta_2})^{t-s} \|\nabla_s\|_1\right]$$

$$= \sqrt{1-\beta_2} \sum_{s=1}^{T} \|\nabla_s\|_1 \sum_{t=s}^{T} (\sqrt{\beta_2})^{t-s} = \sqrt{1-\beta_2} \sum_{s=1}^{T} \|\nabla_s\|_1 \sum_{j=0}^{T-s} (\sqrt{\beta_2})^j$$

$$\left(\text{Since } \sum_{t=1}^{T} \sum_{s=1}^{t} = \sum_{s=1}^{T} \sum_{t=s}^{T}\right)$$

$$\implies \sum_{t=1}^{T} u_t \leq \frac{\sqrt{1-\beta_2}}{1-\sqrt{\beta_2}} \sum_{t=1}^{T} \|\nabla_t\|_1 \qquad\qquad \text{(Geometric series)}$$

Combining the above inequalities, we can conclude that $(*) \geq \frac{1-\sqrt{\beta_2}}{\sqrt{1-\beta_2}} \sum_{t=1}^{T} \|\nabla_t\|_1$, and therefore, if $C_4 := \frac{1-\sqrt{\beta_2}}{\sqrt{1-\beta_2}}$, then,

$$f(\theta_{T+1}) \leq f(\theta_1) - \eta \frac{C_4}{2} \sum_{t=1}^{T} \|\nabla_t\|_1 + L_0 \eta^2 (C_3^2 + C_1) T \qquad \left(\text{Setting } \eta < \bar{\eta}_1 := \frac{C_4}{2L_1(C_3^2 + C_1)}\right)$$

$$\implies \sum_{t=1}^{T} \|\nabla_t\|_1 \leq \frac{2 f(\theta_1)}{C_4 \eta} + \frac{2 L_0 \eta (C_3^2 + C_1) T}{C_4}$$

$$\implies \min_{t \leq T} \|\nabla_t\|_1 \leq \frac{\sum_{t=1}^{T} \|\nabla_t\|_1}{T} \leq \frac{2 f(\theta_1)}{C_4 \eta T} + \frac{2 L_0 \eta (C_3^2 + C_1)}{C_4}$$

Setting $\eta = \min\{\bar{\eta}_0, \bar{\eta}_1, \frac{C_4}{4 L_0 (C_3^2 + C_1)} \epsilon\}$,

$$\min_{t \leq T} \|\nabla_t\|_1 \leq \frac{2 f(\theta_1)}{C_4 \eta T} + \frac{\epsilon}{2}$$

Setting $T \geq \frac{4 f(\theta_1)}{C_4 \eta \epsilon} = O\left(\frac{1}{\epsilon^2}\right)$) is sufficient to ensure that,

$$\min_{t \leq T} \|\nabla_t\|_1 \leq \epsilon$$

$\square$

### H.2. Helper Lemmas

**Lemma 21.** *If Assn. 1 and 4 hold with $L_0 \geq 0$, $L_1 \geq 0$ and $\frac{1}{p} + \frac{1}{q} = 1$, then, for all $y, x$ s.t. $\|y - x\|_p \leq \frac{1}{L_1}$,*

$$\|\nabla f(y)\|_q + \frac{L_0}{L_1} \leq \left(\|\nabla f(x)\|_q + \frac{L_0}{L_1}\right) \exp\left(L_1 \|y - x\|_p\right)$$

*Proof.* Define the function $h(\theta) := \ln(\|\nabla f(\theta)\|_q + \frac{L_0}{L_1})$. We will first prove that $h(\theta)$ is $L_1$-Lipschitz w.r.t the $\ell_p$ norm. Since $\|\cdot\|_q$ can be non-smooth, consider a Clarke subgradient computed using Lemma 12

$$\partial h(\theta) = \frac{\nabla^2 f(\theta) \partial \|z\|_q}{\|z\|_q + \frac{L_0}{L_1}}, \qquad z = \nabla f(\theta).$$

Then, for $g = \nabla^2 f(\theta) s$ where $s \in \partial(\|\cdot\|_q)(\nabla f(\theta))$ is a subgradient of $\|\cdot\|_q$ evaluated at $\nabla f(\theta)$, we write the norm of $v \in \partial h(\theta)$ as

$$\|v\|_q = \frac{\left\|[\nabla^2 f(\theta)] s\right\|_q}{\|\nabla f(\theta)\|_q + \frac{L_0}{L_1}} \leq \frac{\left\|\nabla^2 f(\theta)\right\|_{p \to q} \|s\|_p}{\|\nabla f(\theta)\|_q + \frac{L_0}{L_1}} \qquad \text{(By definition of the matrix norm)}$$

$$\implies \|v\|_q \leq \frac{\left\|\nabla^2 f(\theta)\right\|_{p \to q}}{\|\nabla f(\theta)\|_q + \frac{L_0}{L_1}} \qquad \text{(If } s \in \partial(\|\cdot\|_q)\text{, then, } \|s\|_p \leq 1\text{)}$$

Using Assn. 4 to simplify the numerator, and noting that lower-bounding the denominator,

$$\forall v \in \partial h(\theta), \quad \|v\|_q \leq \frac{L_0 + L_1 \|\nabla f(\theta)\|_q}{\|\nabla f(\theta)\|_q + \frac{L_0}{L_1}} \leq L_1$$

By Lebourg's mean value theorem ((Clarke et al., 1998), Thm. 2.4), since $h$ is Lipschitz in an open set containing $y$ and $x$, then there exists a point $u = tx + (1-t)y$, for some $t \in [0,1]$ such that

$$\exists g \in \partial h(u), \qquad h(y) - h(x) = \langle g, y - x \rangle.$$

Therefore,

$$h(y) - h(x) \leq \max_{t \in [0,1]} \langle g, y - x \rangle \quad where \quad g \in \partial h(t\,x + (1-t)\,y)$$

$$\leq \|y - x\|_p \max_{t \in [0,1]} \|g\|_q \qquad \text{(Using Holder's inequality)}$$

$$\leq L_1 \|y - x\|_p \qquad \text{(Using the above bound on the subgradient)}$$

$$\implies \ln(\|\nabla f(y)\|_q + \frac{L_0}{L_1}) - \ln(\|\nabla f(x)\|_q + \frac{L_0}{L_1}) \leq L_1 \|y - x\|_p$$

$$\implies \|\nabla f(y)\|_q + \frac{L_0}{L_1} \leq \left( \|\nabla f(x)\|_q + \frac{L_0}{L_1} \right) \exp\left( L_1 \|y - x\|_p \right)$$

$\qquad\qquad\qquad\qquad\qquad\qquad\qquad\qquad\qquad\qquad\qquad\qquad\qquad\qquad\qquad\qquad\qquad\qquad$ $\square$

**Lemma 22.** *Under Assn. 1, Assn. 4 with $L_0 \geq 0, L_1 > 0$, if for all iterations $t$, (i) $\|\theta_t - \theta_{t+1}\|_p \leq \frac{a}{L_1}$ for some constant $a$ such that $\sqrt{\beta} \exp(a) < 1$, then,*

$$\frac{\|\nabla_t\|_1}{\sqrt{1 - \beta} \sum_{j=0}^{t-1} (\sqrt{\beta})^j \|\nabla_{t-j}\|_1} \geq \frac{\|\nabla_t\|_1}{\|\nabla_t\|_1 + \frac{L_0}{L_1}} \frac{1 - \sqrt{\beta} \exp(a)}{\sqrt{1 - \beta}}$$

*Proof.* Using part (1) of Lemma. 21 with $p = \infty, q = 1$ and $y = \theta_{t-1}, x = \theta_t, c > 0$,

$$\|\nabla_{t-1}\|_1 + \frac{L_0}{L_1} \leq (\|\nabla_t\|_1 + \frac{L_0}{L_1}) \exp\left( L_1 \|\theta_{t-1} - \theta_t\|_\infty \right)$$

$$\leq (\|\nabla_t\|_1 + \frac{L_0}{L_1}) \exp(a) \qquad \text{(Using assumption (i))}$$

$$\implies \|\nabla_{t-j}\|_1 \leq (\|\nabla_t\|_1 + \frac{L_0}{L_1}) \exp(a\,j) \qquad \text{(Recursing and since } L_0/L_1 > 0\text{)}$$

Using the above relation to lower-bound $\sqrt{1 - \beta} \sum_{j=0}^{t-1} (\sqrt{\beta})^j \|\nabla_{t-j}\|_1$, first note that,

$$\sqrt{1 - \beta} \sum_{j=0}^{t-1} (\sqrt{\beta})^j \|\nabla_{t-j}\|_1 \leq \sqrt{1 - \beta} \sum_{j=0}^{t-1} (\sqrt{\beta})^j (\|\nabla_t\|_1 + \frac{L_0}{L_1}) \exp(a\,j)$$

$$\leq \frac{\sqrt{1-\beta}\,(\|\nabla_t\|_1 + \frac{L_0}{L_1})}{1 - \sqrt{\beta}\,\exp(a)} \qquad \text{(Since } \sqrt{\beta}\,\exp(a) < 1\text{)}$$

$$\implies \frac{\|\nabla_t\|_1}{\sqrt{1-\beta}\,\sum_{j=0}^{t-1}(\sqrt{\beta})^j\,\|\nabla_{t-j}\|_1} \geq \frac{\|\nabla_t\|_1}{\|\nabla_t\|_1 + \frac{L_0}{L_1}}\,\frac{1 - \sqrt{\beta}\,\exp(a)}{\sqrt{1-\beta}}$$

$\square$

**Lemma 23.** *If Assn. 1 and 4 hold, for all $y, x$ s.t. $\|y - x\|_p \leq \frac{1}{L_1}$,*

$$\|\nabla f(y) - \nabla f(x)\|_q \leq [L_0 + L_1\,\|\nabla f(x)\|_q]\,e\,\|y - x\|_p\,.$$

*Proof.* By the fundamental theorem of calculus,

$$\nabla f(y) - \nabla f(x) = \int_{t=0}^1 \nabla^2 f((1-t)\,x + t\,y)\,(y - x)\,dt$$

$$\implies \|\nabla f(y) - \nabla f(x)\|_q = \left\|\int_{t=0}^1 \nabla^2 f((1-t)\,x + t\,y)\,(y - x)\,dt\right\|_q$$

$$\leq \int_{t=0}^1 \left\|\nabla^2 f((1-t)\,x + t\,y)\,(y - x)\right\|_q\,dt \qquad \text{(Triangle inequality)}$$

$$\leq \int_{t=0}^1 \left\|\nabla^2 f((1-t)\,x + t\,y)\right\|_{p\to q}\,\|y - x\|_p\,dt \qquad \text{(By definition of matrix norm)}$$

$$\leq \|y - x\|_p\,\left[\int_{t=0}^1 L_0 + L_1\,\|\nabla f((1-t)\,x + t\,y)\|_q\,dt\right] \qquad \text{(Using Assn. 4)}$$

$$= \|y - x\|_p\,\left[L_0 + L_1\,\int_{t=0}^1 \|\nabla f((1-t)\,x + t\,y)\|_q\,dt\right]$$

$$\leq \|y - x\|_p\,\left[L_1\,\int_{t=0}^1 \left(\|\nabla f(x)\|_q + \frac{L_0}{L_1}\right)\,\exp(L_1\,t\,\|y - x\|)\,dt\right]$$
$$\text{(Using Lemma. 21 with } \theta = ty + (1-t)x \text{ and } \theta' = x\text{)}$$

$$= \|y - x\|_p\,\left[L_1\,\left(\|\nabla f(x)\|_q + \frac{L_0}{L_1}\right)\,\int_{t=0}^1 \exp(L_1\,t\,\|y - x\|)\,dt\right]$$

$$\leq \|y - x\|_p\,\left[L_1\,\left(\|\nabla f(x)\|_q + \frac{L_0}{L_1}\right)\,\int_{t=0}^1 \exp(t)\,dt\right] \qquad \text{(Since } \|y - x\|_p \leq \frac{1}{L_1}\text{)}$$

$$= \left[L_1\,\left(\|\nabla f(x)\|_q + \frac{L_0}{L_1}\right)\,(e - 1)\right]\,\|y - x\|_p$$

$$\leq [L_0 + L_1\,\|\nabla f(x)\|_q]\,e\,\|y - x\|_p$$

$\square$

**Lemma 24.** *If Assn. 1 and 4 hold, for all $y, x$ s.t. $\|y - x\| \leq \frac{1}{L_1}$,*

$$f(y) \leq f(x) + \langle \nabla f(x), y - x \rangle + \left(L_0 + L_1\,\|\nabla f(x)\|_q\right)\,\|y - x\|_p^2$$

*Proof.* Define $u(t) = (1-t)x + ty$ and $g(t) = f(u(t))$. Use Taylor's theorem for $g$,

$$g(b) = g(a) + (b - a)\,g'(a) + \int_{t=a}^b (b - t)\,g''(t)\,dt$$

$$\implies g(1) = g(0) + g'(0) + \int_{t=0}^{1} (1-t)\, g''(t)\, dt \qquad \text{(Substituting } a = 0 \text{ and } b = 1)$$

We know that,

$$g'(t) = \frac{\partial f(u(t))}{\partial t} = \langle \nabla f(u(t)), y - x \rangle \quad ; \quad g''(t) = \frac{\partial^2 f(u(t))}{\partial t^2} = (y-x)^T \nabla^2 f(u(t))(y-x)$$

Combining the above relations, and using that $g(1) = f(y)$, $g(0) = f(x)$, $g'(0) = \langle \nabla f(x), y - x \rangle$.

$$f(y) = f(x) + \langle \nabla f(x), y - x \rangle + (y-x)^T \left[ \int_{t=0}^{1} (1-t)\, \nabla^2 f(t\, y + (1-t)\, x)\, dt \right] (y - x)$$

Simplifying the last term,

$$(y-x)^T \left[ \int_{t=0}^{1} (1-t)\, \nabla^2 f(t\, y + (1-t)\, x)\, dt \right] (y - x)$$

$$\leq \|y - x\|_p \left\| \left[ \int_{t=0}^{1} (1-t)\, \nabla^2 f(t\, y + (1-t)\, x)\, dt \right] (y - x) \right\|_q \qquad \text{(Holder's inequality)}$$

$$\leq \|y - x\|_p \left[ \int_{t=0}^{1} (1-t)\, \left\| \nabla^2 f(t\, y + (1-t)\, x)\, (y - x) \right\|_q dt \right] \qquad \text{(Triangle inequality)}$$

$$\leq \|y - x\|_p^2 \int_{t=0}^{1} (1-t)\, \left\| \nabla^2 f(t\, y + (1-t)\, x) \right\|_{p \to q} dt \qquad \text{(By definition of matrix norm)}$$

$$\leq \|y - x\|_p^2 \int_{t=0}^{1} (1-t)\, [L_0 + L_1 \|\nabla f(t\, y + (1-t)\, x)\|_q]\, dt \qquad \text{(Using Assn. 4)}$$

$$= \|y - x\|_p^2 \int_{t=0}^{1} (1-t)\, L_1 \left[ \frac{L_0}{L_1} + \|\nabla f(t\, y + (1-t)\, x)\|_q \right] dt$$

$$\leq \|y - x\|_p^2 \int_{t=0}^{1} (1-t)\, L_1 \left[ \left( \|\nabla f(x)\|_q + \frac{L_0}{L_1} \right) \exp(L_1\, t\, \|y - x\|_p) \right] dt$$
$$\text{(Using Lemma. 21 with } \theta = ty + (1-t)x \text{ and } \theta' = x)$$

$$= \|y - x\|_p^2 \int_{t=0}^{1} (1-t)\, \left( (L_0 + L_1 \|\nabla f(x)\|_q)\, \exp(L_1\, t\, \|y - x\|_p) \right) dt$$

$$= \|y - x\|_p^2 \, (L_0 + L_1 \|\nabla f(x)\|_q) \int_{t=0}^{1} (1-t)\, \left( \exp(L_1\, t\, \|y - x\|_p) \right) dt$$

$$\leq \|y - x\|_p^2 \, (L_0 + L_1 \|\nabla f(x)\|_q) \int_{t=0}^{1} (1-t)\, \left( \exp(t) \right) dt \qquad \text{(Since } \|y - x\|_p \leq \frac{1}{\sqrt{L_1}})$$

$$\leq \|y - x\|_p^2 \, (L_0 + L_1 \|\nabla f(x)\|_q) \left[ \int_{t=0}^{1} \exp(t)\, dt - \int_{t=0}^{1} t\, \exp(t)\, dt \right]$$

$$\leq \|y - x\|_p^2 \, (L_0 + L_1 \|\nabla f(x)\|_q)\, (e - 2)$$
$$\leq \|y - x\|_p^2 \, (L_0 + L_1 \|\nabla f(x)\|_q)$$

Putting everything together,

$$f(y) \leq f(x) + \langle \nabla f(x), y - x \rangle + \left( L_0 + L_1 \|\nabla f(x)\|_q \right) \|y - x\|_p^2$$

$\square$

**Lemma 25.** *Under Assn. 1, Assn. 2, Assn. 4 with $L_0 \geq 0$ and $L_1 \geq 0$, for the coordinate-wise Adam update in Eq. (18), with $\eta \leq \frac{1}{B\, L_1}$ and $\beta_1 < \beta_2$, where $B := \sqrt{\frac{1-\beta_1}{1-\beta_2}}$,*

$$f(\theta_{t+1}) \leq f(\theta_t) - \eta \langle \nabla_t, d_t \rangle + (L_0 + L_1 \|\nabla f(\theta_t)\|_1)\, \eta^2 B^2 \quad ; \quad \|d_t\|_\infty \leq B$$

*Proof.* If $\|\theta_{t+1} - \theta_t\|_\infty \leq \frac{1}{L_1}$, using Lemma. 24 with $p = \infty$ and $q = 1$, and denoting $\nabla_t := \nabla f(\theta_t)$ for convenience,

$$f(\theta_{t+1}) \leq f(\theta_t) - \eta \langle \nabla_t, d_t \rangle + (L_0 + L_1 \|\nabla f(\theta_t)\|_1) \, \eta^2 \, \|d_t\|_\infty^2$$

Simplifying the third term on the RHS, $\|d_t\|_\infty = \max_i \frac{m_{t,i}}{\sqrt{v_{t,i}}}$.

$$m_{t,i} = (1 - \beta_1) \sum_{s=1}^{t} \beta_1^{t-s} g_{s,i} \tag{84}$$

$$\implies m_{t,i}^2 \leq (1 - \beta_1)^2 \left( \sum_{s=1}^{t} \beta_1^{t-s} \right) \left( \sum_{s=1}^{t} \beta_1^{t-s} g_{s,i}^2 \right) \qquad \text{(By Cauchy–Schwarz)}$$

$$\leq (1 - \beta_1) \sum_{s=1}^{t} \beta_1^{t-s} g_{s,i}^2 \qquad \text{(By geometric series)}$$

$$\leq \frac{1 - \beta_1}{1 - \beta_2} \left[ (1 - \beta_2) \sum_{s=1}^{t} \beta_2^{t-s} g_{s,i}^2 \right] \qquad \text{(Since } \beta_1 \leq \beta_2 \text{)}$$

$$= \frac{1 - \beta_1}{1 - \beta_2} v_{t,i} \qquad \text{(By definition of } v_{t,i} \text{)}$$

$$\implies \frac{m_{t,i}^2}{v_{t,i}} \leq \frac{1 - \beta_1}{1 - \beta_2} \implies \frac{m_{t,i}}{\sqrt{v_{t,i}}} \leq B := \sqrt{\frac{1 - \beta_1}{1 - \beta_2}} \tag{85}$$

$$\implies \|\theta_{t+1} - \theta_t\|_\infty \leq \eta \, \|d_t\|_\infty \leq \eta \, B \tag{86}$$

Ensuring that $\eta \leq \frac{1}{B L_1}$ guarantees that $\|\theta_{t+1} - \theta_t\|_\infty \leq \frac{1}{L_1}$. Combining the above inequalities,

$$f(\theta_{t+1}) \leq f(\theta_t) - \eta \langle \nabla_t, d_t \rangle + (L_0 + L_1 \|\nabla f(\theta_t)\|_1) \, \eta^2 \, B^2 \tag{87}$$

$\square$

**Lemma 26.** *Under Assn. 1 and Assn. 4, if $B = \sqrt{\frac{1-\beta_1}{1-\beta_2}}$ with $L_0 \geq 0$ and $L_1 > 0$, for the Adam update in Eq. (18) with $\eta \leq \frac{1}{B L_1}$ for all $t$,*

$$\sum_i \frac{|g_{t,i}| \, |m_{t,i} - g_{t,i}|}{\sqrt{v_{t,i}}} \leq C \left( L_0 + L_1 \|\nabla f(\theta_t)\|_1 \right) \quad \text{where,} \quad C = \frac{\beta_1 \, e \, B \sum_{s=1}^{t-1} (\sqrt{\beta_1})^{t-1-s} \eta_s}{\sqrt{1 - \beta_2}}.$$

*Proof.*

$$\sum_i \frac{|g_{t,i}| \, |m_{t,i} - g_{t,i}|}{\sqrt{v_{t,i}}} \leq \frac{1}{\sqrt{1 - \beta_2}} \sum_i |m_{t,i} - g_{t,i}| \qquad \text{(Since } v_{t,i} \geq (1 - \beta_2) g_{t,i}^2 \text{)}$$

$$\leq \frac{\|m_t - g_t\|_1}{\sqrt{1 - \beta_2}} \tag{88}$$

In order to bound $\|m_t - g_t\|_1$, note that,

$$|m_{t,i} - g_{t,i}| = |\beta_1 m_{t-1,i} + (1 - \beta_1) g_{t,i} - g_{t,i}| = \beta_1 |m_{t-1,i} - g_{t,i}| \tag{89}$$

$$\leq \beta_1 |m_{t-1,i} - g_{t-1,i}| + \beta_1 |g_{t-1,i} - g_{t,i}|$$

$$\implies \sum_i |m_{t,i} - g_{t,i}| \leq \beta_1 \sum_i |m_{t-1,i} - g_{t-1,i}| + \beta_1 \sum_i |g_{t-1,i} - g_{t,i}|$$

$$\implies \|m_t - g_t\|_1 \leq \beta_1 \|m_{t-1} - \nabla_{t-1}\|_1 + \beta_1 \|\nabla_t - \nabla_{t-1}\|_1$$

Using Lemma. 23 with $p = \infty$, $q = 1$, $y = \theta_{t-1}$ and $x = \theta_t$ while ensuring that $\eta_{t-1} \leq \frac{1}{B L_1}$ guarantees that $\|\theta_t - \theta_{t+1}\|_\infty \leq \frac{1}{L_1}$

$$\leq \beta_1 \|m_{t-1} - \nabla_{t-1}\|_1 + \beta_1 (L_0 + L_1 \|\nabla f(\theta_t)\|_1) e \|\theta_t - \theta_{t-1}\|_\infty$$
$$\leq \beta_1 \|m_{t-1} - \nabla_{t-1}\|_1 + \beta_1 \underbrace{(L_0 + L_1 \|\nabla f(\theta_t)\|_1)}_{:=\bar{L}_t} e \eta_{t-1} B$$

$$\implies \frac{\|m_t - g_t\|_1}{\bar{L}_t} \leq \beta_1 \frac{\|m_{t-1} - \nabla_{t-1}\|_1}{\bar{L}_t} + \beta_1 e \eta_{t-1} B \tag{90}$$

We now write $\bar{L}_t$ in terms of $\bar{L}_{t-1}$. In particular, using Lemma. 21 with $p = \infty$, $q = 1$ and $y = \theta_{t-1}$, $x = \theta_t$,

$$\bar{L}_t = L_0 + L_1 \|\nabla f(\theta_t)\|_1 \geq (L_0 + L_1 \|\nabla f(\theta_{t-1})\|_1) \exp(-L_1 \|\theta_t - \theta_{t-1}\|_\infty)$$
$$= \bar{L}_{t-1} \exp(-L_1 \|\theta_t - \theta_{t-1}\|_\infty)$$
$$\frac{1}{\bar{L}_t} \leq \frac{1}{\bar{L}_{t-1}} \exp(L_1 \|\theta_t - \theta_{t-1}\|_\infty) \leq \frac{1}{\bar{L}_{t-1}} \frac{1}{\sqrt{\beta_1}}$$

(Using Eq. (86) and ensuring that $\eta_{t-1} \leq \frac{1}{B L_1} \ln(1/\sqrt{\beta_1})$)

Combining the above inequality with Eq. (90),

$$\frac{\|m_t - g_t\|_1}{\bar{L}_t} \leq \sqrt{\beta_1} \frac{\|m_{t-1} - \nabla_{t-1}\|_1}{\bar{L}_{t-1}} + \beta_1 e \eta_{t-1} B \tag{91}$$

$$\leq \beta_1 e B \sum_{s=1}^{t-1} (\sqrt{\beta_1})^{t-1-s} \eta_s \qquad \text{(Recursing and using that } m_{0,i} = g_{1,i})$$

$$\implies \|m_t - g_t\|_1 \leq (L_0 + L_1 \|f(\theta_t)\|_1) \beta_1 e B \sum_{s=1}^{t-1} (\sqrt{\beta_1})^{t-1-s} \eta_s \tag{92}$$

Combining the above inequality with Eq. (88),

$$\sum_i \frac{|g_{t,i}| |m_{t,i} - g_{t,i}|}{\sqrt{v_{t,i}}} \leq (L_0 + L_1 \|f(\theta_t)\|_1) \underbrace{\frac{\beta_1 e B \sum_{s=1}^{t-1} (\sqrt{\beta_1})^{t-1-s} \eta_s}{\sqrt{1 - \beta_2}}}_{:=C} = C (L_0 + L_1 \|f(\theta_t)\|_1)$$

$\square$

# I. Simplifying and Generalizing the result in Vaswani & Harikandeh (2025)

The update for steepest descent with Armijo line-search can be written as:

$$\theta_{t+1} = \theta_t - \eta_t \|\nabla_t\|_q d_t \quad \text{where,} \quad d_t := \arg\max_{\|d\|_p \leq 1} \langle d, \nabla_t \rangle. \tag{93}$$

Given $c \in (0, 1)$ and $\eta_{\max} > 0$, $\eta_t$ is the largest step-size that satisfies the Armijo condition at iteration $t$, i.e.,

$$f(\theta_t - \eta d_t) \leq f(\theta_t) - c \eta \|\nabla_t\|_q^2 \quad ; \quad \eta \leq \eta_{\max} \tag{94}$$

We will now prove the following lemma that lower-bounds the step-size returned by the Armijo line-search.

**Lemma 27.** *If $f$ satisfies Assn. 1 and 2, at iteration t, the update in Eqs. (93) and (94) returns a step-size $\eta_t \geq \min\left\{\eta_{\max}, \frac{1-c}{H_0 + H_1 f(\theta_t)}\right\}$.*

*Proof.* We will show that any $\eta \leq \frac{1-c}{H_0 + H_1 f(\theta_t)}$ will satisfy the Armijo condition, and hence the back-tracking Armijo line-search will return a step-size larger than $\min\left\{\eta_{\max}, \frac{1-c}{H_0 + H_1 f(\theta_t)}\right\}$.

First, note that for $\eta \leq \frac{1}{H_0 + H_1 \, f(\theta_t)}$,

$$\eta \, \|\nabla_t\|_q \leq \frac{\|\nabla_t\|_q}{H_0 + H_1 \, f(\theta_t)} \leq \frac{\sqrt{2 H_0 \, f(\theta_t) + H_1 \, [f(\theta_t)]^2}}{H_0 + H_1 \, f(\theta_t)} \qquad \text{(Using Lemma. 3)}$$

$$\leq \frac{1}{\sqrt{H_1}} \frac{\sqrt{2 H_0 \, H_1 \, f(\theta_t) + H_1^2 \, [f(\theta_t)]^2 + H_0^2}}{H_0 + H_1 \, f(\theta_t)} \qquad \text{(Since } H_0 + H_1 \, f(\theta_t) > 0)$$

$$\implies \eta \, \|\nabla_t\|_q \leq \frac{1}{\sqrt{H_1}}$$

Hence, for any $\eta \leq \frac{1-c}{H_0 + H_1 \, f(\theta_t)} < \frac{1}{H_0 + H_1 \, f(\theta_t)}$, the condition required for Lemma. 7 is satisfied for $y = \theta_{t+1}$, $x = \theta_t$. Using Lemma. 7,

$$f(\theta_{t+1}) \leq f(\theta_t) - \eta \, \|\nabla_t\|_q \, \langle \nabla_t, d_t \rangle + \eta^2 \, (H_0 + H_1 \, f(\theta_t)) \, \|\nabla_t\|_q^2 \qquad \text{(Since } \|d_t\|_p = 1)$$

$$= f(\theta_t) - \eta \, \|\nabla_t\|_q^2 + \eta^2 \, (H_0 + H_1 \, f(\theta_t)) \, \|\nabla_t\|_q^2 \qquad \text{(By definition of the dual norm)}$$

$$\leq f(\theta_t) - c \, \eta \, \|\nabla_t\|_q^2 \qquad \text{(By choice of } \eta)$$

Hence, the Armijo condition is satisfied for any $\eta \leq \frac{1-c}{H_0 + H_1 \, f(\theta_t)}$. Since the back-tracking line-search returns the largest step-size (smaller than $\eta_{\max}$) that satisfies the Armijo condition, the returned step-size $\eta_t \geq \max \left\{ \eta_{\max}, \frac{1-c}{H_0 + H_1 \, f(\theta_t)} \right\}$. □

The above lower-bound on $\eta_t$ holds for steepest descent and is tighter than the one derived in Vaswani & Harikandeh (2025). Given this lower-bound, the subsequent results in Vaswani & Harikandeh (2025) can be derived analogously.

