# OpenReview forum: "Convergence of Steepest Descent and Adam under Non-Uniform Smoothness"
_ICML.cc/2026/Conference — ICML 2026 regular_

### Official Review · Reviewer_zxcW · 2026-03-06

**Soundness:** 3
**Presentation:** 4
**Significance:** 4
**Originality:** 3
**Overall Recommendation:** 5
**Confidence:** 3

**Summary:**

The paper analyzes the convergence of first-order optimization methods including normalized steepest descent (NSD) and adaptive gradient methods (deterministic, diagonal variants of RMSProp and Adam) under a generalized non-uniform smoothness (NS) assumption.

**Compliance With Llm Reviewing Policy:**

Affirmed.

**Final Justification:**

After taking into account all factors, I recommend an acceptance due to the significance contribution as well as the clean presentation of the papers.

**Key Questions For Authors:**

Key Questions: With respect to the second point in the weaknesses, can the authors comment on the slowness of the aforementioned algorithms if those assumptions are not satisfied, intuitively?

**Strengths And Weaknesses:**

Strengths:

Strong Motivation from Practice: The paper properly justifies its theoretical assumptions by showing that $(L_0, L_1)$-NS is naturally satisfied by widely used models, including logistic regression, softmax policy gradients in RL, and certain two-layer neural networks. This makes the theoretical results highly relevant to actual practitioner experience.

Provable Advantage of Adaptive Methods: One of the most compelling aspects is the formal theoretical separation between adaptive methods (Adam, RMSProp) and standard methods (GD, AdaGrad, AMSGrad). Providing a lower bound that shows why Adam and RMSProp are provably faster on specific $(0, L_1)$-NS functions addresses a long-standing question in optimization theory.

Linear Convergence Results: Establishing linear convergence for Sign GD and Adam/RMSProp with constant step-sizes on separable data is a strong result. It provides a rigorous explanation for why these methods perform so well in classification tasks where the loss can be driven to zero.

Weaknesses:

Complexity of the Two-Phase Convergence: The transition between the linear "fast phase" and the subsequent "slow phase" is mathematically dense. For a reader focused on implementation, it may be difficult to discern exactly when this transition occurs or how to prevent the slowdown in a real-world training loop.

Significance of Theorem 4: To claim that Gradient Descent (GD), Heavy-ball momentum, AdaGrad, and AMSGrad are slow ($\Omega(1/T)$), the paper requires some conditions with respect to stepsizes are met.

---

> ### Author Rebuttal · Authors · 2026-03-30
>
> We thank the reviewer for their positive review and helpful feedback. We address their concerns below.
>
> > *Complexity of the Two-Phase Convergence: The transition between the linear "fast phase" and the subsequent "slow phase" is mathematically dense. For a reader focused on implementation, it may be difficult to discern exactly when this transition occurs or how to prevent the slowdown in a real-world training loop.*
>
> From a practical perspective, our result shows the importance of using a smaller step-size when using normalized steepest descent, RMSProp or Adam for minimizing functions with $L_0 \neq 0$.  Importantly, this is an inherent limitation of using these methods when minimizing functions with a finite optimum. For example, when minimizing a quadratic using sign GD, if we use an $O(1)$ step-size (independent of $\epsilon$), the iterates will start oscillating closer to the minimum. Consequently, oscillations in the training loss indicate that the iterates are in the vicinity of the solution and are ``overshooting'' the minimizer, meaning that the step-size is too large. This can be used as a practical diagnostic to discern that the algorithm has transitioned into a slow phase where it must use a smaller step-size to converge to the minimizer. In this slow phase, when minimizing a quadratic, decreasing the step-size as $O(1/\sqrt{t})$ or using a constant step-size of $O(\epsilon^2)$ is sufficient to guarantee convergence to an $\epsilon$-optimal solution. Unfortunately, this slowdown is a real phenomenon when using sign GD (or other related variants), and cannot be prevented.
>
> We hope that this answers the reviewer's question.
>
> > *Significance of Theorem 4: To claim that Gradient Descent (GD), Heavy-ball momentum, AdaGrad, and AMSGrad are slow, the paper requires some conditions with respect to stepsizes are met. Can the authors comment on the slowness of the aforementioned algorithms if those assumptions are not satisfied, intuitively?*
>
> The restriction on the step-sizes for these algorithms is quite natural. In particular, in both theory and practice, it is standard to use these algorithms with a constant, non-increasing step-size independent of $T$ (for example, see the PyTorch documentation of these algorithms for practical choices). If these assumptions are not satisfied, the behaviour of these algorithms will depend on the specific choice of step-sizes.
>
> For example, for GD with an increasing step-size (violating the non-increasing assumption), if the step-size is increased at a rate inversely proportional to the rate of decrease in the gradient norm (an adaptive scheme), the resulting algorithm resembles normalized gradient descent. Consequently, from Theorem 1, we know that it will attain a linear convergence rate when minimizing the exponential/logistic loss on separable data. On the other hand, suppose we use GD with a large constant $O(T)$ step-size ( violating the independent of $T$ assumption), existing theory proves an $\Theta(1/T^2)$ rate when minimizing the exponential/logistic loss on separable data [1]. However, note that these large step-sizes will cause the algorithm to diverge when minimizing a quadratic. Hence, the behaviour of GD depends on how the large step-sizes are chosen and the properties of the function (for example, see [1,2,3] and references therein). The behaviour of other algorithms, such as HB, AdaGrad, and AMSGrad, with increasing, large step-sizes is more complex [4], and to the best of our knowledge, has not been studied theoretically.
>
> **Additional References**
>
> [1] Wu et al, Large Stepsize Gradient Descent for Logistic Loss: Non-Monotonicity of the Loss Improves Optimization Efficiency
>
> [2] Wu et al, Large Stepsizes Accelerate Gradient Descent for Regularized Logistic Regression
>
> [3] Cohen et al, Gradient Descent on Neural Networks Typically Occurs at the Edge of Stability
>
> [4] Cohen et al, Adaptive Gradient Methods at the Edge of Stability

---

> > ### Author Rebuttal · Reviewer_zxcW · 2026-04-02
> >
> > I thank the authors for the detailed answers. I would like to keep my positive score.

---

### Official Review · Reviewer_opr9 · 2026-03-13

**Soundness:** 3
**Presentation:** 3
**Significance:** 3
**Originality:** 2
**Overall Recommendation:** 4
**Confidence:** 4

**Summary:**

This paper proves linear convergence of various algorithms (SignGD, NormGD, RMSProp, Adam) under two different assumptions - the $(L_0, L_1)$ - nonuniform smoothness upper bound and the $\tau$ - nonuniform Lojasiewicz condition. Many problems of interest, such as  logistic regression, reinforcement learning, and certain neural networks satisfy the conditions. With the two assumptions even though the function is non-convex we can deal with convex optimization - like inequalities with descent lemmas. Also in the nonuniform smooth upper bound we can deal with different norms, which leads to natural convergence guarantees for different optimizers. Extending the analysis of SignGD we can obtain the convergence guarantee for RMSProp, and moreover Adam. This fast convergence is in distinction with AdaGrad, which has a lower bound worse than the previous methods.

**Compliance With Llm Reviewing Policy:**

Affirmed.

**Final Justification:**

I have mixed feelings towards this paper. I understand the contribution of convergence proofs for different optimization algorithms under the $(L_0, L_1)$ constraint is a meaningful contribution, and the authors provide some distinctions between different optimization algorithms (e.g. Adagrad versus Adam). And I understand that certain lemmas are nontrivial and novel. However the two main concerns on technical novelty and application to different "norm based" algorithms (e.g. Adam vs normalized GD) yet remains. I remain my score.

**Key Questions For Authors:**

1. Is there a way to estimate the smoothness coefficients in Assumption 2?
2. Can this theory give insights to practical training, especially on hyperparameter tuning?

**Limitations:**

yes

**Strengths And Weaknesses:**

Soundness: there is no experiment, a purely theoretical paper. It is great that authors extensively explain why their assumptions on Hessian and nonuniform Lojasiewicz makes sense. It is impressive to have a general result that can be applicable to a broad class of optimization algorithms. The proof of convergence for RMSProp and Adam is also nice, and I get the high level picture (that entries of RMSProp lie in a l- infinity ball) which seems correct. I went through the proof of descent lemmas, RMSProp and Adam, and they seem correct.

Presentation: Even though this is a purely theoretic paper, the authors present various implications of their theory and successfully proposes that their work is strong enough to explain many different things. I was personally surprised that reinforcement learning had that benign property that loss can decay exponentially, which it can never happen for fixed learning rate GD. Also extensive background research is presented so that it is easy to accept the premises of the authors and their impact.

Significance: Proving convergence guarantees for RMSProp and Adam is a meaningful work for optimization. Relaxing conditions so that we can prove something in the nonconvex landscape, and refining which conditions are meaningful is also the right direction for research.

Originality: I have mixed feelings in the originality. I think the proofs are all original, and they contain meaningful novel content about the convergence of various optimization algorithms. However, two things make me think that the improvements in the paper are "somewhat" incremental. First, when we assume the $(L_0, L_1)$ - NS and nonuniform Lojasiewicz, essentially we have a descent lemma and we can show linear convergence, which feels like a variant of proof in convex optimization. Hence the proof technique feels a bit replicated to me. Also, we have all these algorithms with different norms (Sign - GD, Norm - GD, Sign - CD GS), but it seems to me that we all have the same qualitative property that for a while it converges linearly then it converges $O(1/\epsilon^\tau)$. Hence the proof is meaningful enough for acceptance, but I can't help but think that there might be more going on that the current theory cannot grasp.

---

> ### Author Rebuttal · Authors · 2026-03-30
>
> We thank the reviewer for their positive review and helpful feedback. We address their concerns below.
>
> > *First, when we assume the - NS and nonuniform Lojasiewicz, essentially we have a descent lemma, and we can show linear convergence, which feels like a variant of a proof in convex optimization. Hence, the proof technique feels a bit replicated to me.*
>
> Indeed, establishing the convergence rates for many optimization methods follows this recipe:  first show one-step descent, then show global convergence at a good rate. A key limitation of the standard theoretical framework is its inability to establish meaningful guarantees for minimizing standard loss functions, such as logistic regression on separable data. From a technical perspective, such limitations arise because the standard descent lemma only holds for uniformly smooth functions with Euclidean norms.
>
> Consequently, in Section 3, we extend the descent lemma and other standard results to $(L_0, L_1)$ non-uniform smooth functions with general $p, q$ norms. Once these results are established, the proofs for normalized steepest descent (Theorem 1) follow the standard optimization template. We believe that generalizing the standard properties and following the same proof structure that the community is used to is an important contribution of our work. We note that the proofs for RMSProp and Adam require establishing additional results beyond the standard optimization machinery. In particular, for RMSProp, we prove Lemmas 17-19 that allow us to bound the effect of the RMSProp preconditioner. The proof for Adam further needs to quantify the effect of the momentum, and we do that by using Lemma 21.
>
> We hope this clarification addresses the reviewer's concern and makes our contributions clearer.
>
> > *Is there a way to estimate the smoothness coefficients in Assumption 2?*
>
> The smoothness coefficients can be estimated for some common losses (for example, see Prop. 1-3). In general, there are techniques to obtain the same convergence rates without requiring the explicit knowledge of the smoothness. For example, for gradient descent, Vaswani & Harikandeh (2025) set the step-size by using Armijo line-search. This allows them to obtain similar guarantees as in our Theorem 1, without requiring the explicit knowledge of the constants (L0, L1). However, their results inherently assume the Euclidean setting and do not hold for general steepest descent. We believe that by using the properties derived in Section 3, we can extend these results to steepest descent with Armijo line-search.
>
> However, note that it is not clear how to achieve such adaptivity for RMSProp or Adam. For this, we plan to investigate recent works [3,4] that combine adaptive methods with a line-search or Polyak step-size, and adapt to the smoothness for standard uniformly-smooth functions. Other possibilities include quantifying the effect of misspecifying the step-size (in terms of L0, L1) and analyzing the convergence [5]. We believe that this is an important direction for future work.
>
> > *Can this theory give insights into practical training, especially on hyperparameter tuning?*
>
> Our theoretical results can give us some insight about practice -- for example, it establishes the separation between Adam and RMSprop vs AdaGrad and AMSGrad, and justifies the practical use of Adam with a constant step-size and momentum. As we mentioned in the paper, existing theoretical results show that such adaptive methods have the same slower convergence rate. Ours is the first such separation result that can distinguish between the performance of these adaptive gradient methods. Furthermore, our theory provides a loss-specific justification for preferring Adam/RMSProp over gradient descent, a question that has been extensively studied recently [1,2].
>
> With respect to hyper-parameter tuning, existing theory (including ours) can only provide weak justification for choosing specific values of $\beta_1, \beta_2$ for Adam, and there is still a significant gap between theory and practice. However, we believe that developing the properties of practical objectives (like we did in Section 3) is a step towards analyzing the stochastic setting and understanding the dependence between the batch-size and other algorithm hyperparameters. We plan to build on our existing results and investigate this important direction in future work.
>
> **Additional References**
>
> [1] Kunstner et al, Noise Is Not the Main Factor Behind the Gap Between SGD and Adam on Transformers, but Sign Descent Might Be
>
> [2] Tomihari et al, Understanding Why Adam Outperforms SGD: Gradient Heterogeneity in Transformers
>
> [3] Vaswani et al, Adaptive Gradient Methods Converge Faster with Over-Parameterization (but you should do a line-search)
>
> [4] Jiang et al, Adaptive SGD with Polyak stepsize and Line-search: Robust Convergence and Variance Reduction
>
> [5] Hubler et al, Parameter-Agnostic Optimization under Relaxed Smoothness

---

> > ### Author Rebuttal · Reviewer_opr9 · 2026-04-02
> >
> > My concerns have been mostly adequately addressed by the authors.
> >
> > I believe this could be an issue with flavor: I agree with the authors that providing a template that many loss functions can fall into is a meaningful contribution, and I understand that to prove properties for RMSGrad and Adam, further bounds on the momentum is needed. But the point is, the paper is extending the $(L_0, L_1)$ - NS assumption that was proposed in  to general $(p,q)$ and proving theorems - hence to me the concept that is introduced in this paper seems slightly incremental (though enough for publication).
> >
> > Also a weakness that I see is that we cannot explain why certain norm-based methods work better than others with this framework. This part was not really addressed by the authors. For instance, can this framework be used to show that Adam works better than SGD in Language model tasks or tasks with anisotropic gradient, but works similar with SGD when gradient is well-posed? The rates are all identical so it is hard to see.
> >
> > At last the framework has interesting theoretical implications but the practical benefit is not clear. For a theory paper it does not necessarily have to be practical, but with the above weaknesses I personally think I will keep my score.
> >
> > [1] Alimisis, F., Islamov, R., and Lucchi, A. Why do we need warm-up? a theoretical perspective. arXiv preprint arXiv:2510.03164, 2025

---

> > > ### Author Response · Authors · 2026-04-06
> > >
> > > Thank you for your continued engagement and for maintaining a positive view of our work. We would like to make two quick clarifications.
> > >
> > > >the paper is extending the $(L_0,L_1)$- NS assumption to general $(p,q)$ and proving theorems - hence to me the concept that is introduced in this paper seems slightly incremental (though enough for publication).
> > >
> > > *We note that we provide the first convergence guarantees for normalized steepest descent and RMSProp/Adam for $(L_0, L_1)$-NS functions* (please see our response to Rev. jjRJ for a detailed literature review).
> > >
> > > In order to prove dimension-free rates for normalized steepest descent and the practical coordinate-wise variants of Adam and RMSProp, it is necessary to handle general $(p,q)$ norms with the $(L_0, L_1)$-NS assumption. For example, $p = \infty$, $q = 1$ for the RMSProp/Adam analysis.
> > >
> > > In addition to this generalization, our paper develops novel properties of non-uniform smooth functions. For example, to the best of our knowledge, Lemmas 5 and 10 are new, even in the standard $p = q = 2$ setting. The multiplicative Lipschitz bounds derived in these lemmas are crucial in the subsequent analyses.
> > >
> > > We will clarify this in the final version of the paper, and hope that our response alleviates the reviewer's concerns about the novelty of our work.
> > >
> > > > can this framework be used to show that Adam works better than SGD in Language model tasks or tasks with anisotropic gradient, but works similar with SGD when gradient is well-posed? The rates are all identical, so it is hard to see.
> > >
> > > We believe that there is a misunderstanding here. We consider the deterministic full-batch setting (there is no stochasticity in the updates). Please see our response to Rev. YrJy for a discussion about extensions to the stochastic setting.
> > >
> > > Having said that, our paper already shows a separation between Adam and GD, i.e., the rates of these methods are not identical on $(0, L_1)$-NS functions (e.g., logistic regression on separable data). In particular, we prove that, while Adam (and RMSProp) can attain a fast $O(\exp(-T))$ rate of convergence for $(0, L_1)$ functions (Section 5.2), GD (as well as AdaGrad, AMSGrad and HB momentum) can only attain a slow $\Omega(1/T)$ rate on such functions (Section 6).
> > >
> > > We hope that this addresses the reviewer's concerns about all rates being identical.

---

### Official Review · Reviewer_YrJy · 2026-03-13

**Soundness:** 3
**Presentation:** 4
**Significance:** 3
**Originality:** 3
**Overall Recommendation:** 5
**Confidence:** 4

**Summary:**

This paper studies the convergence of adaptive first-order methods under a generalized non-uniform smoothness condition, $\\|\nabla^2 f(\theta)\\|_{p\to q}\le L_0 + L_1 f(\theta)$, which generalizes conditions studied in several prior works. Under this condition, it is shown that SignSGD and Normalized Steepest Descent obtain better dimension-free rates than fixed step-size gradient descent in logistic regression and for soft-max policy gradient objectives. Convergence is proven for Adam and RMSProp under this condition for two-layer neural networks with separable data, without relying on Lipschitz or convexity assumptions. Finally, a lower bound is shown demonstrating a separation between the performance of AMSGrad/AdaGrad and RMSProp/Adam.

**Compliance With Llm Reviewing Policy:**

Affirmed.

**Key Questions For Authors:**

- The problem setting here considers only optimizing a fixed function; to what extent do the results here generalize to stochastic problem settings?
- Setting the step-size optimally seems to require explicit knowledge of the constants $L_0$ and $L_1$; is there any hope to adapt to these on-the-fly in some way?

**Limitations:**

yes

**Strengths And Weaknesses:**

## Originality / Significance

Strengths:
- The lower bound seems very significant to me; AMSGrad and Adagrad are well-known to perform poorly in practice compared to Adam/RMSProp, despite technically having much stronger guarantees in theory. This is the first result I'm aware of that shows a real separation between the performance guarantees of AMSGrad/Adagrad vs Adam/RMSProp.

- The extension of the NS condition to handle p-norms seems valuable as it allows them to derive dimension-free rates in some important problem settings (logistic regression, policy optimization).


Weaknesses:
- The significance of the result regarding the converging of Adam is unclear to me beyond showing "Adam also works". The result only shows that Adam has a similar guarantee to RMSProp but does not really provide new insights as to why it tends to be better.

## Presentation

I found the paper very clear. The approach, challenges, and results, are all explained well explained.

A few minor typos:
 - Eq. 14 should have $\theta_{t+1}$ rather than $\theta_{T+1}$
 - The matrix norm $\\|\cdot\\|_{p\to q}$ is never defined, and I'm not entirely sure what it means.

## Soundness

I did not find any issues with soundess of the results.

---

> ### Author Rebuttal · Authors · 2026-03-30
>
> We thank the reviewer for their positive review and helpful feedback. We will fix the typos in the final version of the paper and address the reviewer's concerns below.
>
> > *The result only shows that Adam has a similar guarantee to RMSProp, but does not really provide new insights as to why it tends to be better*
>
> Indeed, the purpose of Theorem 3 was to prove that Adam also enjoys the same desirable convergence guarantees as RMSProp. In the deterministic setting that we consider, it is unlikely that Adam has a theoretical benefit over RMSprop. In fact, even without preconditioning, when comparing heavy-ball momentum and gradient descent beyond quadratic functions [1], the convergence rates are similar in the deterministic setting. That being said, it is likely that the behaviour and convergence of Adam vs RMSProp is different in the stochastic setting. For example, in [2], the authors show that adding momentum to normalized SGD removes the need for large batch sizes on nonconvex objectives. We plan to investigate this important direction in future work.
>
> > *to what extent do the results here generalize to stochastic problem settings?*
>
> Thank you for the good question. We plan to use the structural properties derived in Section 3 of our paper and investigate the effect of stochasticity on the convergence of these algorithms for $(L_0, L_1)$ non-uniform smooth functions.
>
> Some recent works have explored the convergence of specific algorithms in the stochastic setting, for example, Vaswani & Harikandeh (2025); Gorbunov et al. (2024) studied the convergence of SGD on non-uniform smooth but convex functions. Hubler et al, (2024) have studied normalized SGD with momentum for $(L_c, L_g)$ non-uniform smooth, general non-convex functions (not necessarily satisfying Assumption 3) and have derived sub-linear rates for stationary point convergence.
>
> We believe that the stochastic extension of our results is non-trivial and will require careful reasoning about the dependence between the algorithm parameters, batch size, and problem properties. We plan to investigate this direction in future work.
>
> > "Setting the step-size optimally seems to require explicit knowledge of the constants, and is there any hope to adapt to these on-the-fly in some way?"
>
> We agree that this is an important direction of future research, and there are some related recent works. For example, for gradient descent, Vaswani & Harikandeh (2025) set the step-size by using Armijo line-search. This allows them to obtain similar guarantees as in our Theorem 1, without requiring the explicit knowledge of the constants. However, their results inherently assume the Euclidean setting and do not hold for general steepest descent. We believe that by using the non-uniform smoothness properties derived in Section 3, we can extend these results to steepest descent with Armijo line-search.
>
> However, we note that it is not clear how to achieve such adaptivity for RMSProp or Adam. For this, we plan to investigate recent works [3,4] that combine adaptive methods with a line-search or Polyak step-size and aim to adapt to L for standard uniformly smooth functions. Other possibilities include quantifying the effect of misspecifying the step-size (in terms of L0, L1) and analyzing the convergence [5].
>
> > The matrix norm $|| \cdot ||_{p \to q}$ is never defined.
>
> $|| A ||_{p \to q}$  is a matrix norm. It is the $\max ||Ax||_q$ over vectors $x$ such that $|| x ||_p \leq 1$ . In the special case where $p = q = 2$, it is equal to the standard spectral norm. We will explicitly define it in the final version of the paper.
>
> **Additional References**
>
> [1] Ghadimi et al, Global convergence of the Heavy-ball method for convex optimization
>
> [2] Cutkosky et al, Momentum Improves Normalized SGD
>
> [3] Vaswani et al, Adaptive Gradient Methods Converge Faster with Over-Parameterization (but you should do a line-search)
>
> [4] Jiang et al, Adaptive SGD with Polyak stepsize and Line-search: Robust Convergence and Variance Reduction
>
> [5] Hubler et al, Parameter-Agnostic Optimization under Relaxed Smoothness

---

> > ### Author Rebuttal · Reviewer_YrJy · 2026-03-31
> >
> > Thanks for the detailed and informative response! I will keep my positive score

---

### Official Review · Reviewer_jjRJ · 2026-03-22

**Soundness:** 4
**Presentation:** 3
**Significance:** 3
**Originality:** 3
**Overall Recommendation:** 5
**Confidence:** 2

**Summary:**

This paper introduces two novel non-uniform smoothness conditions for objective functions in minimization problems (Assumptions 2 and 3) and studies the convergence of three classes of first-order optimization algorithms: steepest descent methods (under the $l_1$, $l_2$, and $l_\infty$ norms), as well as RMSProp and Adam, when the objective satisfies these conditions.
As a first step, the paper demonstrates that a broad class of problems satisfies the proposed non-uniform smoothness conditions, including logistic regression, softmax policy gradients in multi-armed bandits, and certain classes of two-layer neural networks. The main results rely on inequalities (11), (12), and (13), which are derived from Assumptions 2 and 3. The key theoretical contributions are presented in Theorem 1 (for steepest descent), Theorem 3 (for RMSProp), and a corresponding result for Adam. These results can be interpreted as establishing a two-phase convergence rate, depending on whether $\varepsilon > \frac{L_0}{L_1}$ or $\varepsilon \le \frac{L_0}{L_1}$. The paper also provides an extensive discussion of the implications of these results and offers a comprehensive comparison with existing classical results in the literature. Finally, in Section 6, the authors establish a lower bound on the convergence rate for a class of algorithms (including AdaGrad and AMSGrad) and show that RMSProp and Adam do not satisfy these lower-bound conditions.

**Compliance With Llm Reviewing Policy:**

Affirmed.

**Final Justification:**

The authors have carefully addressed my main concern regarding the positioning of their contribution within the existing literature. As I understand, these clarifications can be readily incorporated into the manuscript. Accordingly, I have decided to increase my score.

**Key Questions For Authors:**

- It is not clear from the current presentation how the main results (Theorem 1, Theorem 3, and the corresponding result for Adam) compare to existing results in the literature. Could the authors provide a clear and concise statement highlighting the precise improvements or differences relative to known convergence guarantees?

- It would also be helpful to better understand the breadth of the function class characterized by Assumptions 2 and 3. In particular, how does this class compare to those defined by standard smoothness conditions in the literature? A more explicit discussion or formal comparison would significantly improve clarity.

**Limitations:**

yes

**Strengths And Weaknesses:**

**strengths**

- *Originality and significance*: The non-uniform smoothness conditions introduced in this paper, along with the techniques developed to incorporate them into gradient-based algorithms, are both original and innovative. The wide range of examples provided demonstrates that these conditions hold for models such as logistic regression, softmax policy gradients, and certain neural network and highlights the broad applicability of these non-uniform smoothness conditions. Moreover, the comparisons with existing literature suggest that the proposed results improve upon a substantial body of work on convergence rates of gradient-based methods. However, it is not clear (at least for me) from the current presentation of the paper how significant are these results compared to the existing results in the literature?

- *Soundness*: To the best of my knowledge, the proof sketches for the main results appear to be correct. Although I did not verify the technical details of all lemmas in the appendix, they seem consistent and sound to me.

**Weaknesses**

- *Presentation*: My only concern with the paper is its dense presentation, which at times makes it difficult to follow. In particular, the comparisons with existing literature appear somewhat unstructured. The paper lists several related works and claims improvements over them, but these comparisons are not systematically organized, making it challenging for the reader to clearly understand the precise contributions and distinctions.

---

> ### Author Rebuttal · Authors · 2026-03-30
>
> We thank the reviewer for their positive review and helpful feedback. We address their concerns below.
>
> > *It is not clear from the current presentation how the main results (Theorem 1, Theorem 3, and the corresponding result for Adam) compare to existing results in the literature*
>
> To the best of our knowledge, there is no single paper that unifies and presents the results for normalized steepest descent, RMSProp, and Adam for $(L_0, L_1)$-NS functions. Previous work can be characterized into three broad categories:
>
> **Convergence under $(L_0, L_1)$ non-uniform smoothness:** The most relevant papers are Vaswani & Harikandeh (2025); Alimisis et al. (2025) that consider the same non-uniform assumption as we do. Vaswani & Harikandeh (2025) use this assumption to justify the use of Armijo line-search, while Alimisis et al. (2025) use this assumption to justify the importance of learning-rate warm-up.
>
> In terms of convergence guarantees, Vaswani & Harikandeh (2025) analyze GD with Armijo line-search and Alimisis et al. (2025) analyze normalized GD. Both papers derive similar convergence guarantees as Theorem 1 in our paper. However, both these papers are inherently in the Euclidean setting, and do not derive dimension-free guarantees for general normalized steepest descent, nor do
> they analyze RMSProp or Adam.
>
> **Convergence under $(L_c, L_g)$ non-uniform smoothness**: Zhang et al., 2019; Gorbunov et al., 2024; Vankov et al., 2024; Li et al., 2023; Wang et al., 2024a;b consider a different non-uniform smoothness assumption (see Eq (2)). This assumption only considers Euclidean norms and is, in general, stronger than our (L0, L1) assumption. Importantly, this assumption cannot model even the finite-sum exponential or logistic regression loss.
>
> Under this assumption, Gorbunov et al. 2024 and Vankov et al. 2024 have analyzed the convergence of variants of normalized gradient descent for general non-convex functions (that do not satisfy Assumption 3) and convex functions. However, these papers do not derive dimension-free guarantees for general normalized steepest descent, nor do they consider RMSProp or Adam. For the special case of normalized gradient descent, similar convergence rates can also be obtained under our (L0, L1) assumption (see Vaswani & Harikandeh (2025); Alimisis et al. (2025)).
>
> Under this assumption, Li et al. (2023); Wang et al. (2024a;b) analyze the scalar or norm version of Adam and derive an $O(1/\epsilon^2)$ stationary-point convergence for general non-convex functions. In Appendix G of our paper, we specialize our proof technique to the (Lc, Lg ) assumption, and can derive a faster $O(1/\epsilon)$ convergence rate for the standard, diagonal, coordinate-wise variant of Adam (see Lines 413-416 and 2755-2758).
>
> **Other work:** Apart from these main bodies of work, some papers focus on specific examples of (L0, L1) functions and analyze the convergence of specific normalized steepest descent methods and derive similar rates as in our Theorem 1. For example, Mei et al. (2021) use normalized gradient descent for the softmax policy gradient objective, Taheri & Thrampoulidis (2023) use it for 2-layer neural networks, and Axiotis & Sviridenko (2023) use greedy coordinate descent (corresponding to p = 1, q = ∞ in our Theorem 1) and analyze its convergence on logistic regression.
>
> We will better organize these details and write a more comprehensive literature review in the final version of the paper.
>
> > *..how does this class compare to those defined by standard smoothness conditions in the literature..*
>
> The class of (L0, L1) non-uniform smooth functions that we consider includes the (Lc, Lg ) non-uniform smooth functions satisfying Eq 2 in our paper – see Vaswani & Harikandeh (2025, Prop. 3) and Alimisis et al. (2025, Prop B.1, Appendix D) for a justification. Both these function classes are strictly more expressive than the class of standard uniformly-smooth functions (See lines 111-121 in our paper). For example, the exponential loss is not uniformly smooth, but does satisfy Assumption 2.
>
> On the other hand, Assumption 3 captures gradient-dominated functions. For example, if $\tau = \frac{1}{2}$, Assumption 3 corresponds to the PL condition that can be satisfied by non-convex functions (for example, matrix factorization). Additionally, if the function is also convex, then Assumption 3 generalizes strong-convexity (see Karimi et al. (2016)). For example, since the squared loss is both uniformly smooth and strongly convex, it satisfies Assumptions 2 and 3. We hope this clarifies the connection to the more standard assumptions in the literature.

---

> > ### Author Rebuttal · Reviewer_jjRJ · 2026-04-03
> >
> > I appreciate the authors for their careful and detailed response. I believe their rebuttal has addressed my main concerns regarding the paper. I would encourage the authors to incorporate these comparisons with the literature into the final version, as they would further strengthen the presentation. I will increase my score based on this rebuttal.

---

### Decision · Program_Chairs · 2026-04-30

**Decision:**

Accept (regular)

**Comment:**

This work analyzes first-order methods under non-uniform smoothness assumptions for a broad class of machine learning objectives. The reviewers unanimously agree that this is an important work that provides further understanding of the convergence of popular algorithms, such as SignSGD and Adam, under $(L_0, L_1)$-type smoothness assumptions, and that it does so clearly in the context of prior work. It is therefore the case that acceptance of the paper is recommended.